# Combined dendritic cell and anti-TIGIT immunotherapy potentiates adaptive NK cells against HIV-1

Ildefonso Sánchez-Cerrillo[1,2], María Agudo-Lera[1,3], Olga Popova[1,3], Ilya Tsukalov[1,3], Marta Calvet-Mirabent[1,3], Ignacio de los Santos[2,4], Lucio García-Fraile[2,4], Patricia Fuentes [5], Cristina Delgado-Arévalo[1,3], Juan Alcain[5], Nerea Sánchez-Gaona[6], Judith Grau-Expósito[6], María Lázaro-Díez[7], Cecilia Muñoz-Calleja[1,2,3], Arantzazu Alfranca[1,3,8], Meritxell Genescà [6], Julia G Prado[2,7,9], Vladimir Vrbanac[10], Alejandro Balazs [10], María José Buzón [6], María L Toribio[5], María A Muñoz-Fernández[11], Francisco Sánchez-Madrid[1,8] & Enrique Martín-Gayo [1,2,5]✉

## Abstract

Natural Killer (NK) cells are promising candidates for targeting persistently infected CD4 + T cells in people with HIV-1 (PWH). However, chronicity of HIV-1 infection impairs NK cell functionality, requiring additional strategies to potentiate their cytotoxic activity. This study demonstrates that dendritic cells primed with nanoparticles containing Poly I:C (Nano-PIC-MDDC) enhance the natural cytotoxic function of NK cells from effective responder PWH. These NK cells exhibit increased proportions of NKG2C+ cell subsets capable of eliminating HIV-1 infected CD4 + T cells through the TRAIL receptor. In contrast, in non-responder PWH, elevated expression of the inhibitory receptor TIGIT is associated with reduced frequencies of NKG2C + NK cells and diminished TRAIL expression. TIGIT blockade restores cytotoxicity of NK cells from non-responder PWH against HIV-1-infected cells by upregulating TRAIL. Furthermore, combining Nano-PIC-MDDC-primed NK cells with anti-TIGIT immunotherapy in humanized NSG mice reduces the expansion of HIV-1 infected cells, preserves NKG2C + NK cell precursors and increases TRAIL expression in tissue. Collectively, these findings support the combined use of Nano-PIC-MDDC and TIGIT blockade as a promising immunotherapeutic strategy toward an HIV-1 cure.

**Keywords** Dendritic Cell; HIV; Natural Killer; TIGIT; TRAIL
**Subject Categories** Immunology; Microbiology, Virology & Host Pathogen Interaction

## Introduction

Antiretroviral therapy (ART) is very effective inducing almost complete viral suppression and reduction of plasma viral load to undetectable levels in people with HIV-1 (PWH). However, this treatment does not eliminate persistent HIV-1 reservoirs in latently infected CD4 + T cells (Bachmann et al, 2019; Wan et al, 2020; Mohammadzadeh et al, 2023), which are present in different tissues such as lymphoid organs and gastrointestinal mucosa (Mohammadzadeh et al, 2023; Pieren et al, 2024; Ikeogu et al, 2023; Banga et al, 2021; Thompson et al, 2017). A fraction of these latently infected T cells contains complete viral sequences and can produce replication-competent HIV-1 particles (Siliciano and Siliciano, 2021; Wang et al, 2018). It has been proposed that the proliferation of latently infected cells allows the maintenance of persistent HIV-1 reservoirs (Mullins and Frenkel, 2017; Liu et al, 2020; Virgilio and Collins, 2020). In addition, the chronicity of the persistent HIV-1 infection and the sporadic reactivation of latently infected cells in PWH leads to residual inflammation and continuous stimulation of immune cells with antiviral capacities such as CD8 + T cells and Natural Killer (NK) cells. Such hyperstimulation eventually leads to an exhausted state in these immune cell types that is characterized by the expression of multiple inhibitory checkpoint receptors, reduced functional properties and impaired ability to eliminate infected cells (Guo et al, 2023; Lan et al, 2023; Blanch-Lombarte et al, 2023). In some cases, ART does not seem to completely restore such dysfunctional and exhausted state in immune cells and longer periods of treatment may be required to recover responsiveness in these cells (Calvet-Mirabent and Martín-Gayo, 2022; Dos Santos Guedes et al, 2023; Crowell et al, 2024; Fenwick et al, 2019). Thus, new approaches are needed to develop a more effective

[1]Immunology Unit from Hospital Universitario La Princesa, Instituto Investigación Sanitaria-Princesa IIS-IP, Madrid, Spain. [2]CIBER Infectious Diseases (CIBERINFEC) from Instituto de Salud Carlos III, Madrid, Spain. [3]Medicine Department, Universidad Autónoma de Madrid, Madrid, Spain. [4]Infectious Diseases Unit Hospital Universitario La Princesa, Madrid, Spain. [5]Centro de Biología Molecular Severo-Ochoa, CSIC-UAM, Madrid, Spain. [6]Infectious Diseases Department, Hospital Universitari Vall d'Hebron, Institut de Recerca (VHIR), Barcelona, Spain. [7]IrsiCaixa Research Institute, Barcelona, Spain. [8]CIBER Cardiovascular, Instituto de Salud Carlos III, Madrid, Spain. [9]Germans Trias i Pujol Research Institute (IGTP), Badalona, Spain. [10]Ragon Institute of MGH, MIT and Harvard, Boston, MA, USA. [11]Hospital General Universitario Gregorio Marañón, Madrid, Spain. ✉E-mail: emgayo@cbm.csic.es

antiviral immune response against HIV-1, allowing to achieve a reduction of persistent viral reservoirs and, potentially, a cure. Several studies have previously focused on enhancing HIV-1-specific T cell responses (Calvet-Mirabent et al, 2022; Miner et al, 2024; Malyshkina et al, 2023; Kopycinski et al, 2023; van Duijn et al, 2023; Copertino et al, 2023; Hanke, 2019) from PWH on ART, including the stimulation with activated dendritic cells (DCs) loaded with HIV-1 peptides (Calvet-Mirabent et al, 2022; Mohamed et al, 2020; Kristoff et al, 2019; García et al, 2011; Lu et al, 2004; Lévy et al, 2014) and/or the blockade of checkpoint receptors such as T-cell immunoglobulin and ITIM domain (TIGIT) and T-cell immunoglobulin and mucin-domain containing protein-3 (TIM3) (Benito et al, 2023; Gonzalez-Cao et al, 2019). While these approaches have improved T-cell mediated targeting of HIV-1 infected cells in vitro and in vivo, they may not be sufficient to reduce viral reservoirs to a level capable of promoting spontaneous control or remission of infection.

NK cells are innate immune cells with antiviral capacities that have been shown to be essential to control HIV-1 replication in vivo in animal models (Seay et al, 2015; Marras et al, 2017; Bardhi et al, 2017; Sungur et al, 2022; McBrien et al, 2020; Papasavvas et al, 2019; Pohlmeyer et al, 2019) and in spontaneous and post-treatment controllers (Calvet-Mirabent and Martín-Gayo, 2022; Moyano et al, 2023; Zhang et al, 2021b; Etemad et al, 2023). Although some studies point to their promising therapeutic potential against HIV-1, modulation of NK cells has been less explored (Papasavvas et al, 2019; Board et al, 2024; Schober et al, 2023; Astorga-Gamaza et al, 2023; Grasberger et al, 2024; Anderko and Mailliard, 2023; Alrubayyi et al, 2022). Therefore, enhancing the activity of these cells has currently become an area of active research. NK cells are capable of detecting and eliminating virus-infected cells either by directly recognizing ligands for activating receptors or antibodies bound to viral proteins expressed by target cells. These two killing mechanisms are known as natural and antibody-dependent cellular cytotoxicity (ADCC) (Chung et al, 2008; Abuharfeil et al, 2019; Lucar et al, 2019; Shin et al, 2023; Maia et al, 2024; Prager and Watzl, 2019), and are dependent on the expression of a repertoire of activating receptors such as Natural Killer group 2 member D (NKG2D) or C (NKG2C) and CD16 expressed by NK cells at different stages of maturation, respectively. In this regard, NKG2C+ cells are a subset included in CD56dim CD16+ and CD56lo/− CD16 + NK cells initially identified as an expanded population in cytomegalovirus (CMV) infected individuals that is characterized by a primed state, including higher cytotoxic potential and enriched ADCC activity, and therefore, this subset has been considered to have memory-like properties (Stary and Stary, 2020; Hammer and Romagnani, 2017). These cells originate from NKG2C-NKG2D+ precursors giving rise to immature NKG2C+ CD57− precursors and finally, mature into NKG2C+ CD57+ adaptive cells (Kared et al, 2016; van der Ploeg et al, 2023; Alsulami et al, 2023; Palamarchuk et al, 2021). In the case of HIV-1 infection, adaptive NKG2C+ NK cells have been shown to be capable of recognizing non-classical human leukocyte antigen E (HLA-E) expressing infected CD4 + T cells loaded with viral peptides (Jost et al, 2023; Romero-Martín et al, 2022). In addition, NKG2C+ adaptive NK cells were also associated with remission and functional cure of HIV infection (Climent et al, 2023). Thus, inducing and potentiating adaptive NKG2C+ NK cells may be beneficial to improve

immunotherapy against HIV-1, but this possibility has not been sufficiently explored.

Among different tools useful to improve antiviral immune responses, dendritic cells (DC) are myeloid innate immune cells that are able not only to present antigens and mediate polarization of T cells (O'Keeffe et al, 2015; Rhodes et al, 2019), but also they have the ability to support maturation and activation of NK cells (Zitvogel et al, 2006; Peterson and Barry, 2020; Thomas and Yang, 2016). Recently, we observed that the activation of intracellular innate RNA sensors such as RIG-I increases expression of ligands for NKG2D and the ability of DC to activate cytotoxic NK cells in the autoimmune disorder Sjogren´s Syndrome (Sánchez-Cerrillo et al, 2023). Moreover, monocyte-derived DC (MDDC) modified by agonists of the STING and TLR3 pathways can be used as an immunotherapy to restore HIV-1-specific CD8+ T cell responses from PWH (Calvet-Mirabent et al, 2022). However, the potential of MDDC to enhance cytotoxic function of NK cells from PWH and those clinical and immunological parameters associated with increased NK functional response have not been studied in detail. In this regard, the use of nanoparticles is an attractive approach to facilitate the selective targeting of intracellular sensors such as RIG-I, which may be beneficial to enhance NK cell-activating function in MDDC and to improve immunotherapies against HIV-1.

Here, we show that nanoparticles loaded with Poly I:C (Nano-PIC) increase the ability of MDDC from PWH to activate and restore cytotoxic activity of autologous NK cells against HIV-1 infected CD4 + T cells in a group of as effective responder PWH. Such enhancement of NK cell activity was associated with increased proportions of adaptive NKG2C+ cells expressing tumor necrosis factor-related apoptosis-inducing ligand (TRAIL) in NK cells from effective responder PWH. Moreover, functional restoration and proportions of NKG2C + NK were negatively associated with TIGIT expression in non-responder PWH and positively correlated with TIM3. Finally, blockade of TIGIT increased expression of TRAIL and restored the cytotoxic activity of NK cells from non-responder PWH against p24+ cells in vitro following Nano-PIC-MDDC stimulation. TIGIT blockade was also associated with reduced expansion of infected human CD4+ T cells and with an enrichment in NKG2C+ TRAIL+ NK cells in lymphoid tissue in vivo in humanized mice treated with Nano-PIC-MDDC and NK therapy. Together, our results indicate that combined blockade of TIGIT and stimulation with Nano-PIC-MDDC immunotherapy can potentially be used as a combined strategy to improve NK cell cytotoxic functionality and to reduce persistent HIV-1 replication in PWH.

# Results

## Stimulation of intracellular RNA sensors by Poly I:C-loaded nanoparticles increases ability of MDDC to activate cytotoxic CD16+ NK cells

To determine whether stimulation of CD11chi CD1a+ CD14lo/− MDDCs (Fig. EV1A) with soluble (Sol-PIC) or nanoparticle-encapsulated Poly I:C (Nano-PIC) may differentially regulate functional maturation in these cells and their ability to stimulate NK cells, we first analyzed cell viability, intracellular expression of different cytokines interleukin-12 (IL-12), interferon beta (IFNβ)

and induction of maturation markers such as CD86 and CD40 after in vitro stimulation using cells from n = 23 recruited HIV negative donor (HD) and n = 21 PWH (Figs. 1A–C and EV1B,C; Appendix Table S1). Treatment of MDDC with Nano-PIC induced a mild but significant reduction of cell viability compared to cells treated with Sol-PIC, although it remained over 60% in the case of HD, however viability remained over 90% in PWH (Fig. EV1B). As shown in Fig. 1B, Nano-PIC induced a more significant and consistent increase in intracellular expression of IL-12 in MDDC from both HD and PWH at 6 h compared to cells treated with Sol-PIC, while intracellular expression of IFNβ was induced by Nano-PIC in MDDC from HD without further increasing in MDDC from PWH at this time point (Fig. EV1C, right). Such increase in IL-12 secretion by MDDC from PWH treated with Nano-PIC was confirmed in culture supernatants at 16h, along with significant increased levels of other secreted proinflammatory and immuno-modulatory cytokines such as interleukin 6 (IL-6) and tumor necrosis factor alpha (TNFα), more efficiently induced after Nano-PIC stimulation in MDDC from PWH in contrast to cells from HD (Fig. EV1D). A similar trend was observed for interleukin 15 (IL-15) (Fig. EV1D). While a significant defect in IFNβ detection was more significantly confirmed for MDDC from PWH treated with Nano-PIC, we observed significantly higher secretion of interferon gamma (IFNγ) in cultures from these cells but not in the case of HD (Fig. EV1D). In contrast, interleukin 2 (IL-2) was similarly induced by Nano-PIC in MDDC from HD and from PWH (Fig. EV1D). Although surface levels of the maturation marker CD86 tended to be higher in MDDCs from PWH compared to cells from HD (Fig. EV1E), mean fluorescence intensity (MFI) of this costimulatory molecule was also more efficiently increased at 6 and 16 h in MDDC primed with Nano-PIC from HD and PWH, suggesting an earlier and more effective maturation under these conditions (Figs. 1C and EV1F). Similar effect was observed in surface levels of other maturation markers such as CD40 in MDDC from HD treated with Nano-PIC at 6 h after treatment but was not sustained over time. In contrast, CD40 was significantly increased at baseline in MDDC from PWH and after 16 h in the presence of Nano-PIC compared to Sol-PIC in these cells (Fig. EV1E–H). Upregulation of CD40 was also induced in cells from PWH treated with empty Nano at 16 h, although to a lesser extent (Fig. EV1H). Opposite to cells exposed to Sol-PIC, MDDC from PWH and HD primed with Nano-PIC for 16 h more efficiently induced significant higher expression of major histocompatibility complex class I-related chain A and B (MICa/b) and UL16 binding protein 1 (ULBP-1) (ligands for the activating receptor NKG2D) as well as HLA-E (ligand for both the activating and inhibitory receptors NKG2C or NKG2A) (Appendix Fig. S1A; Fig. 1D). Furthermore, expression of these NK receptor ligands was more pronounced on activated CD86+ compared to CD86- MDDCs from PWH after Nano-PIC stimulation (Appendix Fig. S1A,B). Therefore, higher expression of IL-12 and immunomodulatory cytokines such as IFNγ, as well as ligands for NK cell receptors indicate that activated CD86+ Nano-PIC-MDDCs may potentially, display increased abilities to activate NK cells. To address this possibility, we compared the ability of MDDCs treated with empty nanoparticles or Nano-PIC to increase maturation and activation of autologous NK cells in vitro. To this end, expression of IFNγ and CD107a on CD56dim CD16+ cells was analyzed (Appendix Fig. S2A). An increase in the expression of the degranulation marker CD107a

(Fig. 2A, top panel) and proportions of CD107a+ IFNγ+ (Fig. 2A, bottom panel) cells within the CD56dim CD16 + NK subset from HD (left) and in PWH (right) were observed in response to Nano-PIC-MDDCs compared to NK cells cultured individually. Proportions of CD107a+ IFNγ + NK cells from HD and PWH were also significantly increased under these conditions in contrast to individual NK cells, although statistical significance was higher in the case of PWH (Fig. 2A, bottom panel). The expression of the inhibitory receptor natural killer group 2, member A (NKG2A) and the activating receptor NKG2D was decreased in NK cells from PWH after culture with Nano-PIC-MDDC (Appendix Fig. S2B,C). In the case of NKG2C receptor, no significant differences were observed in NK cells from HD and the initially tested cohort of PWH treated with empty Nano versus Nano-PIC-MDDC (Appendix Fig. S2B,C). Notably, the induction of a cytotoxic cell phenotype was dependent on cell-to-cell contact of NK cells with Nano-PIC-MDDCs in a transwell both in the case of cells from HD (left panel) and PWH (right panel) (Fig. 2B).

To address whether augmented CD107a expression reflected an increase in the natural cytotoxic capacities of NK cells stimulated with Nano-PIC-MDDCs, we performed killing assays in the presence of the target cell line K562 transfected with GFP (Appendix Fig. S2D). As shown in Fig. 2C, significantly higher proportions of dead K562-GFP cells were observed in the presence of NK cells from both HD (left panel) and PWH (right panel) donors primed with Nano-PIC-MDDCs compared to target cells cultured with untreated NK or cells exposed to Nano-MDDCs. In addition, a tendency to a higher ADCC activity against a HIV1-gp120-expressing target cell line precoated with a mix of HIV-1 specific broadly neutralizing antibodies (bNAbs) was observed in the presence of NK cells stimulated with Nano-PIC-MDDCs (Appendix Fig. S2E). Together, the data suggest that nanoparticle-loaded Poly I:C efficiently improves the ability of MDDCs to support the activation of functional cytotoxic NK cells.

## Restoration of cytotoxic activity of NK cells from PWH after Nano-PIC-MDDC stimulation associates with the induction of adaptive NKG2C+ cells

We next assessed whether NK cells from PWH could more efficiently eliminate autologous HIV-1 infected CD4 + T cells after stimulation with Nano-PIC-MDDC. To this end, we tested the ability of NK cells from our cohort of aviremic PWH for at least 1 year on ART (Appendix Table S2) to eliminate autologous HIV-1 infected CD4 + T cells defined by intracellular p24 capsid protein (p24) expression in vitro in the presence of the latency reversal agent (LRA) romidepsin (RMD) and the antiretroviral drug raltegravir (Ral) (Fig. 3A,B). HIV-1 p24 expression was also analyzed in CD4 + T cells treated exclusively with Ral as a control of baseline viral reactivation (Fig. 3B). Upon assessment of functional results considering cells from all participants tested we did not observe any significant differences in proportions of p24 + CD4 + T cells in culture in the presence of NK cells treated with Nano-PIC-MDDCs when considering the total n = 33 PWH tested in these functional assays, which in some cases increased or decreased under these conditions compared to levels present in autologous LRA-treated CD4 + T cells as a reference (fold change p24 + % >1 and <1, respectively) (Fig. EV2A). Next, we assessed whether such heterogeneity observed in changes in the frequencies of p24+ may be associated with specific patterns of NK cell

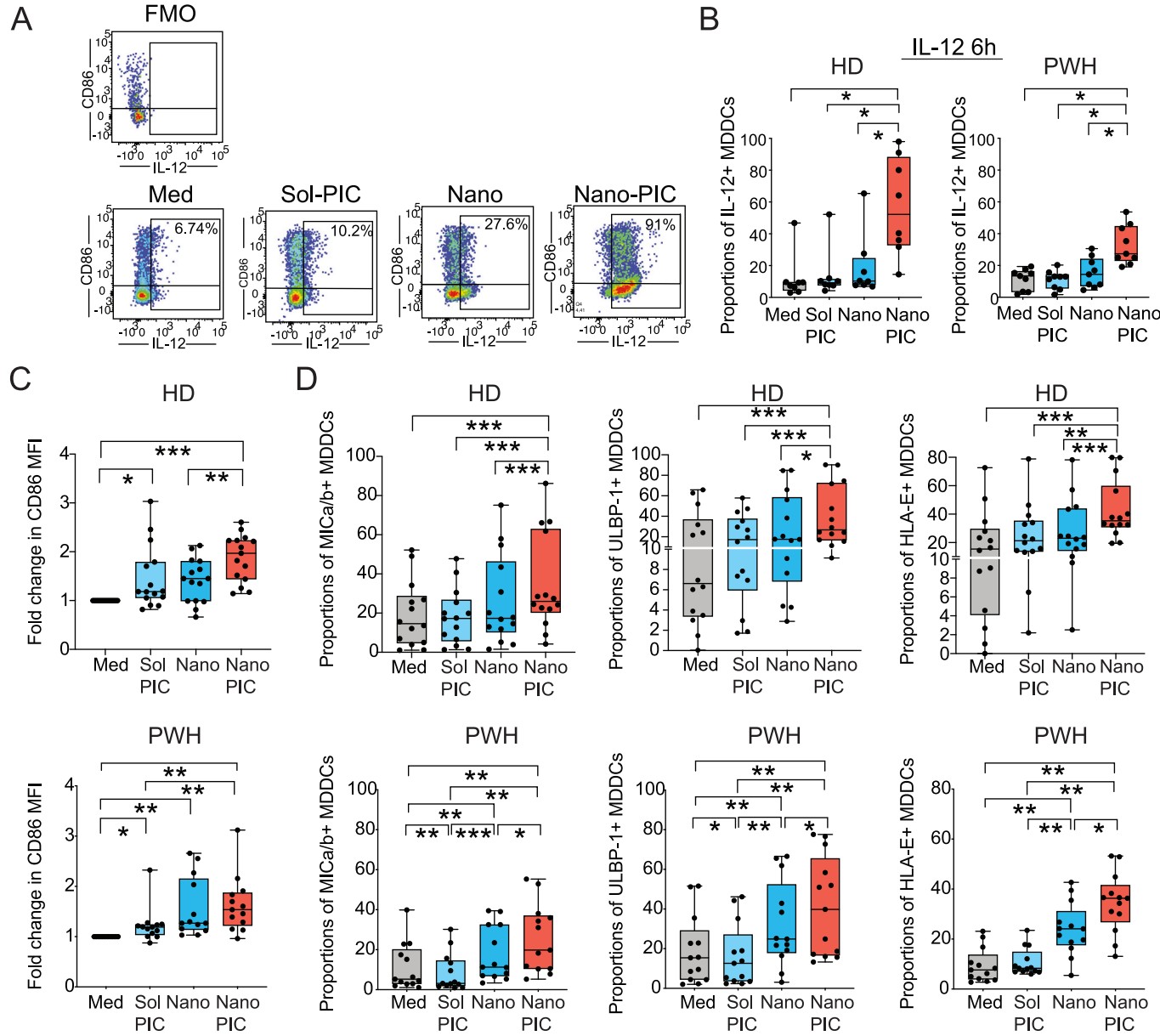

**Figure 1. Analysis of cytokine production, maturation and expression of ligands for NK receptors in MDDC from HD and PWH stimulated with nanoparticle-encapsulated Poly I:C.**

(A) Representative flow cytometry gating strategy defining expression of CD86 in combination with intracellular staining of IL-12 in MDDCs cultured in media (Med) or in the presence of soluble Poly I:C (Sol PIC) or empty (Nano) or poly:IC loaded nanoparticles (Nano PIC). FMO for IL-12 is shown. (B) Intracellular expression of IL-12 at 6 h in $n = 8$ HD (left panel) and $n = 9$ PWH (right panel). Data are represented in Box and Whiskers plots showing median values and maximum and minimum error bars. (C) Mean of fluorescence intensity (MFI) levels of CD86 in $n = 15$ from HD (upper plot) and from $n = 13$ (lower plot) PWH. Data are represented in Box and Whiskers plots showing median values and maximum and minimum error bars. (D) Proportions of NK receptor ligands MICa/b, ULBP-1 and HLA-E in $n = 14$ HD (upper plot) and $n = 13$ PWH (lower plot) donors cultured in these conditions. Data are represented in Box and Whiskers plots showing median values and maximum and minimum error bars. Statistical significance was calculated using a two-tailed Wilcoxon pair matched test and Bonferroni correction for multiple comparisons. *$P < 0.05$; **$P < 0.01$; ***$P < 0.001$. Source data are available online for this figure.

activation, such as differences in proportions of adaptive NKG2C + NK cells from PWH after Nano-PIC-MDDC treatment, which have been linked to the control of HIV-1 infection (Ma et al, 2017). Taking into consideration these populations, we observed a significant negative correlation between proportions of NKG2C+ cells in CD56 dim and CD56lo/− CD16+ NK stimulated with

Nano-PIC-MDDC and reduction of percentages of co-cultured infected p24+ cells from all PWH tested (Fig. 3C,D). Based on these associations, we then asked if levels of NKG2C could predict different PWH groups whose NK differentially responded to Nano-PIC-MDDC stimulation inducing a decrease (fold change <1) or an increase (fold change >1) in proportions of p24+ cells from

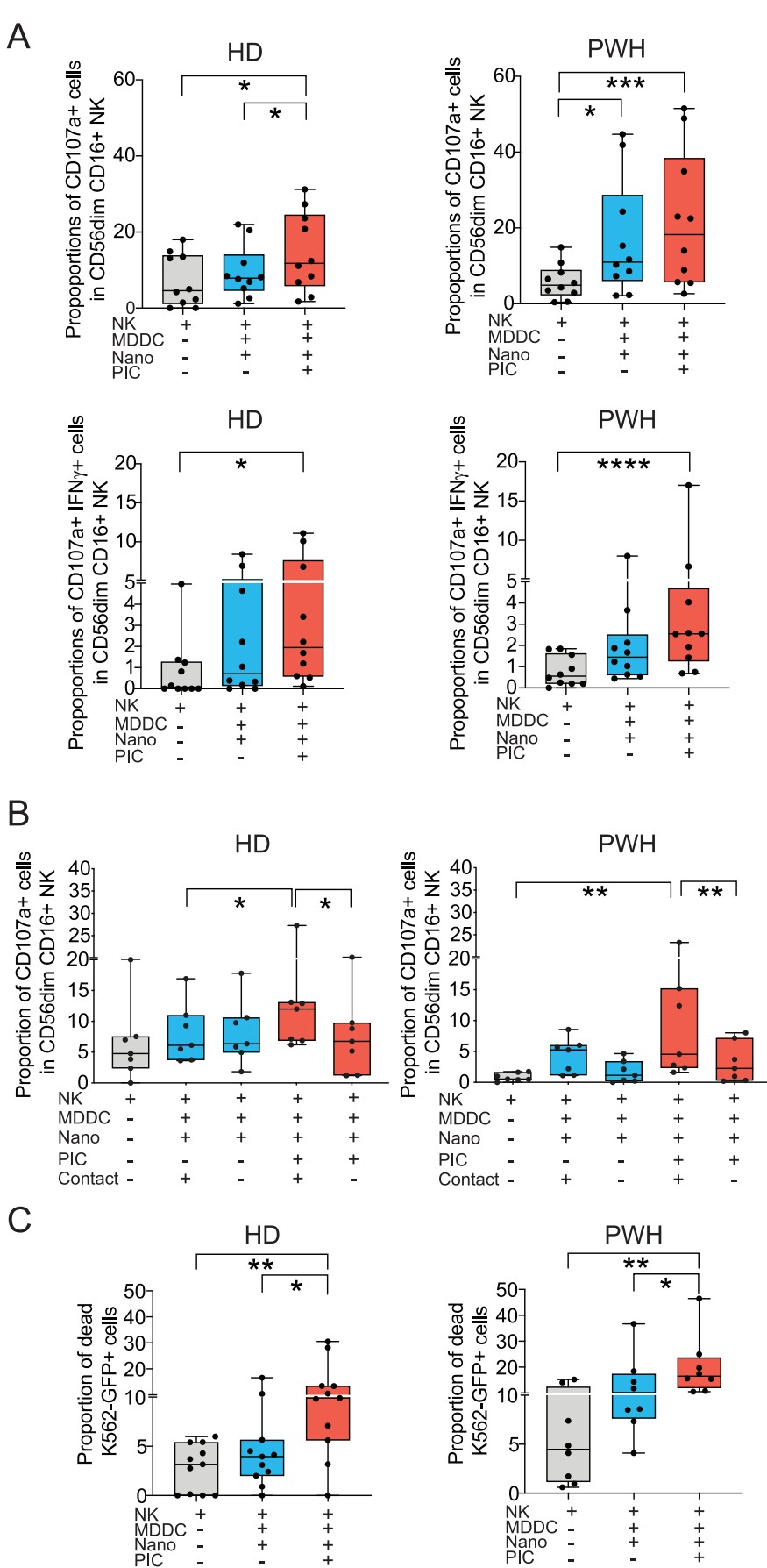

**Figure 2. Nano-PIC-MDDCs promote the activation of cytotoxic NK cells from HD and PWH.**

(A) Analysis of proportions of total CD107a+ (upper panels) and CD107a+ IFNγ+ cells (lower panels) on CD56dim CD16+ NK cells cultured in the absence or the presence of MDDC stimulated either with empty or PIC-loaded nanoparticles in $n = 10$ HD (left panel) and $n = 10$ PWH (right panel) donors. Data are represented in Box and Whiskers plots showing median values and maximum and minimum error bars. (B) Test of proportions of total CD107a+ CD56dim CD16 + NK cells with or without cell contact in a transwell assay in the same conditions using $n = 7$ HD (left panel) and $n = 7$ PWH (right panel) donors. Data are represented in Box and Whiskers plots showing median values and maximum and minimum error bars. (C) Analysis of NK-cell mediated natural cytotoxicity function assessed by proportions of dead target K562-GFP cell line after culture in the presence of NK cells in $n = 11$ HD and $n = 8$ PWH donors alone or co-cultured with MDDC treated with empty or PIC-loaded nanoparticles. Data are represented in Box and Whiskers plots showing median values and maximum and minimum error bars. Statistical significance was calculated using a Friedman test for multiple comparisons. *$P < 0.05$; **$P < 0.01$; ***$P < 0.001$; ****$P < 0.0001$. Source data are available online for this figure.

baseline. A ROC-curve analysis predicted with 80% specificity and 77.8% sensitivity two groups of effective responder (defined by >16.6% NKG2C+ cells and a fold change of p24+ proportions from baseline <1) and non-responder (<16.6% NKG2C+ cells and a fold change of p24+ proportions from baseline >1) subgroups from our PWH cohort that accounted for 80% of individuals (Fig. 3E–G). A 20% PWH remained in a third unclassified group that was characterized by an intermediate phenotype between the two extreme PWH groups and did not meet the mentioned criteria (Fig. 3E–G). Interestingly, effective responders expressed low basal levels of p24 protein (Median proportion of positive cells 0.035%) and increased levels after LRA treatment (Median proportion of positive cells 0.076%) (Fig. EV2B). Individual treatment of CD4+ T cells with NK cells from effective responders did not significantly change frequencies of infected cells, but a significant reduction (median fold change 0.81) in proportions of p24+ cells was observed in the presence of NK cells and Nano-PIC-MDDC but not empty Nano MDDC (Fig. 3G). In contrast, non-responders seemed to basally express higher levels of p24 protein (Median proportion of positive cells 0.043%) but did not further increase percentages of infected cells after RMD treatment (Median proportion of positive cells 0.045%) (Fig. EV2B). Moreover, individual NK cells were also unable to reduce infected cells and a median 2.6-fold increase in percentages of p24+ cells were observed in the presence of NK and Nano-PIC-MDDC compared to LRA-treated CD4 + T lymphocytes in the non-responder PWH group (Fig. 3G). We did not consistently detect p24+ cells in co-cultured Nano-PIC-MDDCs, suggesting that HIV-1 replication was restricted to CD4 + T cells in these assays (Fig. EV2C). Notably, we observed a consistent trend in reduction of proportions of intact proviral HIV-1 DNA sequences compared to defective viruses in selected samples from effective responders from our functional assays, suggesting a reduction in cells producing replication competent-virus, in contrast to non-responder PWH that followed the opposite trend (Fig. EV2D; Appendix Table S3). We next analyzed co-expression of NKG2C with CD57, since combination of these markers can be used as a maturation criteria (Fig. 3C), allowing to distinguish between NKG2C + CD57− precursor cells, fully mature NKG2C + CD57+ adaptive NK cells and a non-adaptive effector NKG2C− CD57 + NK subset (Kobyzeva et al, 2020). Using this stratification criteria, we observed that the significant enrichment of total NKG2C+ cells in effective compared to non-responder PWH (Fig. 3H) was due to an increase in proportions of both NKG2C+ CD57− precursors (Fig. 3I) and mature NKG2C+ CD57+ (Fig. 3J) adaptive NK cells within CD56dim CD16+ and also in CD56lo/− CD16+ cell subsets from these individuals. In contrast, proportions of effector NKG2C− CD57 + NK cells tended to increase in NK from the non-responder

PWH group compared to effective responder subjects, especially in CD56dim CD16+ NK cells (Fig. EV2E). Once again, unclassified patients displayed an intermediate phenotype for these NK populations between effective and non-responder PWH (Figs. 3H±J and EV2E). Of note, from all clinical parameters available from the extreme two effective and non-responder PWH groups, we only detected significant differences in age and CD4+ /CD8 + T cell ratios (Appendix Table S2). In this regard, effective PWH were younger (median 45 vs 56 years) and characterized by higher CD4+ /CD8 + T cell ratios (median 1.1 vs 0.88) than non-responder individuals (Appendix Table S2). Finally, at a functional level, NKG2C+ CD57- adaptive NK cell precursors from selected effective responder PWH were intrinsically characterized by significant higher expression levels of the degranulation marker CD107a (Fig. EV2F, left) and higher co-expression of this molecule with IFNγ (Fig. 3K, left) and granzyme B (Fig. 3K, right), compared to effector NKG2C− CD57 + NK cells in this PWH group. Mature NKG2C+ CD57+ memory NK were characterized by an intermediate and similar functional phenotype and tended to contain higher frequencies of CD107a+ cells and significantly higher proportions of CD107a+ Granzyme B+ cells compared to NKG2C− CD57+ cells (Figs. 3K and EV2F). Likewise, the same tendencies in total IFNγ+ cells were also found in NKG2C + CD57− and NKG2C + CD57 + NK cell subsets (Fig. EV2F, upper right plot). In contrast, levels of TNFα tended to be lower on NKG2C + CD57− NK cells and higher on NKG2C+ CD57+ memory mature NK from effective responder PWH (Fig. EV2F, bottom plot). Together, these findings suggest that effective response of NK cells from PWH to Nano-PIC-MDDC immunotherapy is associated with the presence of precursor and mature memory NKG2C+ NK cells that may be characterized by increased cytotoxic and cytokine secretion phenotypical profiles, and which may contribute to improved targeting of HIV-infected CD4+ T cells in vitro.

## Expression of TIGIT limits induction of functional adaptive NKG2C+ CD57- NK and cytotoxic function against HIV-1-infected cells

To determine whether different levels of immune exhaustion could be associated with the presence of adaptive NKG2C+ NK cells and the differential functional response observed in the effective and non-responder PWH groups, we analyzed the expression of the checkpoint receptors TIGIT, TIM3, and programmed cell death protein 1 molecule (PD-1) in CD56dim and CD56lo/− CD16 + NK at baseline and after stimulation with Nano-PIC-MDDC (Appendix Fig. S3A). Expression of TIGIT and PD-1 was intrinsically higher in cells from PWH compared to HD at baseline, while similar levels of TIM3 were observed (Appendix Fig. S3B). However, no obvious

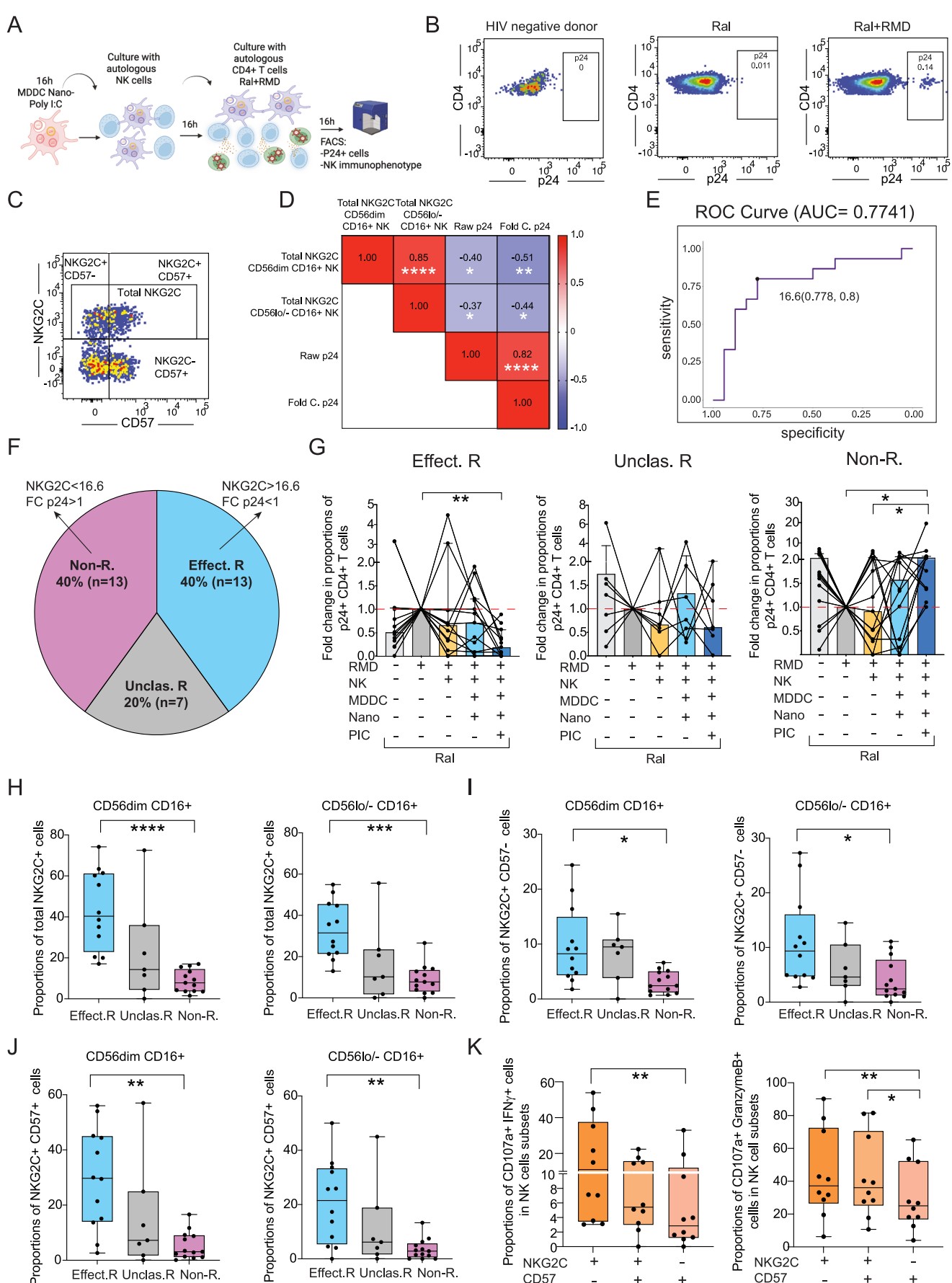

**Figure 3.   Impact of Nano-PIC-MDDC on cytotoxic function of NK cells from PWH against HIV-1 infected CD4 + T cells.**

(A) Schematic representation experiment of functional assays evaluating the elimination p24 + CD4 + T cells after culture with NK cells primed by Nano-PIC-MDDC. (B) Representative flow cytometry dot plots showing intracellular expression of HIV-1 p24 on CD4 + T cells from a PWH in the presence of Raltegravir alone or in combination with Romidepsin. A staining background control from an HIV negative donor is also shown. (C) Representative flow cytometry dot plot representing NKG2C and CD57 expression in gated CD56dim CD16 + NK cells. (D) Heatmaps representing Spearman correlation matrix between memory NK subsets within CD56dim or CD56lo/- CD16 + NK cells and proportions of p24+ cells after co-culture with NK primed by Nano-PIC-MDDC immunotherapy. Fold changes in p24+ frequencies are also included. Levels of positive and negative associations are highlighted in different intensities of red and blue, respectively. Significant associations have also been highlighted. *$P < 0.05$; **$P < 0.01$; ****$P < 0.0001$. (E) ROC curve analysis, defining the cut-off based on NKG2C expression defining effective and non-responder PWH groups in our cohort. (F) Pie chart representing proportion of PWH from our cohort predicted as effective and non-responder PWH by ROC curve analysis and those PWH that were not classified by this model. (G) Fold change of intracellular expression of HIV-1 p24 on CD4 + T cells from aviremic ART PWH treated with Raltegravir and Romidepsin, in the absence or the presence of NK cells alone or stimulated with MDDCs treated with empty Nano or Nano-PIC. Data from three separate groups with effective-responders (Effect. R.; $n = 13$), unclassified (Unclas. R.; $n = 7$) and non-responders (Non-R.; $n = 13$) PWH to Nano-PIC-MDDC. (H–J) Proportions of total (H) NKG2C+ cells and different memory NK subsets based on combination of NKG2C and CD57 (I, J) on CD56dim or CD56lo/− CD16 + NK from effective responder ($n = 13$, Effect. R; blue), unclassified ($n = 7$, Unclas. R; gray) and non-responder ($n = 13$, Non-R.; pink) PWH after activation with Nano-PIC-MDDCs. Data are represented in Box and Whiskers plots showing median and maximum and minimum values. (K) Proportions of CD107a+ IFNγ+ (left) and CD107a+ Granzyme B+ (right) cells included in gated NK subsets defined by differential NKG2C and CD57 expression from ($n = 10$) selected effective responder PWH after PMA and Ionomycin stimulation (see methods). Data are represented in Box and Whiskers plots showing median and maximum and minimum values. Statistical significance was calculated using a Friedman or a Kruskal–Wallis tests for multiple comparisons. *$P < 0.05$; **$P < 0.01$; ***$P < 0.001$; ****$P < 0.0001$. Source data are available online for this figure.

differences in basal expression of these checkpoint receptors were observed in NK cells from effective and non-effective responder PWH or unclassified individuals (Appendix Fig. S3C). Interestingly, no significant changes in PD-1 expression were observed, but this molecule tended to be downregulated in CD56dim CD16 + NK from effective responder PWH in response to Nano-PIC-MDDC (Appendix Fig. S3D). In contrast, expression of TIGIT significantly increased in CD56dim CD16+ and CD56lo/− CD16 + NK cells from non-responder PWH from baseline after Nano-PIC-MDDC stimulation compared to effective responders PWH (Fig. 4A). Induction of TIGIT expression in CD56dim CD16+ NK cells positively correlated with proportions of p24+ CD4+ T cells observed in our functional assays after culture with Nano-PIC-MDDC either considering all PWH or selecting the most extreme effective and non-effective responder groups excluding unclassified intermediate PWH groups (Fig. 4B, left). Similar results were obtained for the CD56lo/− CD16+ NK subset (Fig. 4B, right). Moreover, TIGIT expression was associated with lower percentages of NKG2C+ CD57− (Fig. 4C, left) adaptive precursor NK cells and with higher proportions of effector NKG2C− CD57+ (Fig. 4C, right) cells within the CD56dim CD16+ population from all PWH including unclassified individuals or only considering effective and non-effective responder PWH. Once again, these tendencies were maintained in CD56lo/− CD16+ subsets, but these correlations were not significant (Appendix Fig. S3E). In contrast, TIM3 expression was significantly induced in NK cells from effective responder group compared to non-responder PWH after Nano-PIC-MDDC treatment in CD56lo/− CD16 + NK cell subset. A similar trend was observed for CD56dim CD16 + NK cells (Appendix Fig. S3F). Moreover, levels of TIM3 expression in NK cells appeared to be associated with lower proportions of infected p24+ CD4+ T cells in our functional assays (Appendix Fig. S3G) and with higher frequencies of NKG2C+ CD57− adaptive precursor NK cells (Appendix Fig. S3H, right) and lower percentages of NKG2C− CD57+ cells in the CD56lo/- CD16+ subpopulation (Appendix Fig. S3I, right), although a similar tendency for these populations seemed to be present in in the CD56dim CD16+ NK subset (Appendix Fig. S3H–I, left). Therefore, TIGIT and TIM3 may differentially associate with the presence of adaptive NKG2C + NK cells and functional restoration of NK in response to Nano-PIC-MDDC.

Next, we determined whether blockade of TIGIT could restore the cytotoxic properties of NK cells from $n = 20$ PWH from our cohort including $n = 11$ non responder and $n = 9$ effective individuals (Appendix Table S4) in response to Nano-PIC-MDDC. To this end, we performed additional functional assays using NK cells from PWH stimulated with Nano-PIC-MDDC and analyzed their ability to eliminate autologous reactivated p24+ CD4+ T cells in the presence of anti-TIGIT or the corresponding isotype control Abs. As shown in Fig. 4D (left), blockade of TIGIT led to a marked significant reduction in the frequencies of p24+ CD4+ T cells in the presence of NK cells from PWH primed with Nano-PIC-MDDC compared to cultures treated with isotype control. The positive functional effect of the anti-TIGIT mAb in these assays was more marked in non-responder PWH unable to reduce proportions of infected cells in in the presence of Nano-PIC-MDDC and isotype Abs (Fig. 4D, right). No statistical differences were observed in the reduction of infected p24+ cell proportions in assays using NK cells from effective responder PWH in the presence of Nano-PIC-MDDC and isotype Ab compared to cultures performed with anti-TIGIT Abs (Fig. EV3A). Also, the functional impact of the anti-TIGIT ab was specific of NK from PWH cultured in the presence of Nano-PIC-MDDCs, since no significant changes in elimination of p24+ cells were observed in cultures performed with unstimulated NK cells treated with this blocking Ab (Fig. EV3B). In contrast, blockade of TIM3 did not significantly enhance functional restoration of NK cells from PWH considering all individuals tested or by separately analyzing effective and non-responders PWH (Fig. EV3C, left and right). Supporting an increase of cytotoxic function of NK cells from PWH in the presence of anti-TIGIT mAbs, a significantly higher proportion of cells co-expressing CD107a and IFNγ were found in these conditions but not in the presence of isotype control Abs (Fig. 4E). In contrast, no significant changes on proportions of CD107a+ IFNγ+ NK cells were observed in the presence of the blocking anti-TIM3 Ab compared to isotype controls (Fig. EV3D). In addition, we ruled out that the anti-TIGIT mAbs had an impact on frequencies of p24+ cells when added to cultures of isolated CD4+ T cells from PWH cultured individually in the presence of media or after reactivating with PHA and IL-2, suggesting that they did not significantly revert viral latency or mediate elimination of infected cells (Fig. EV3E). Thus, our data demonstrate that

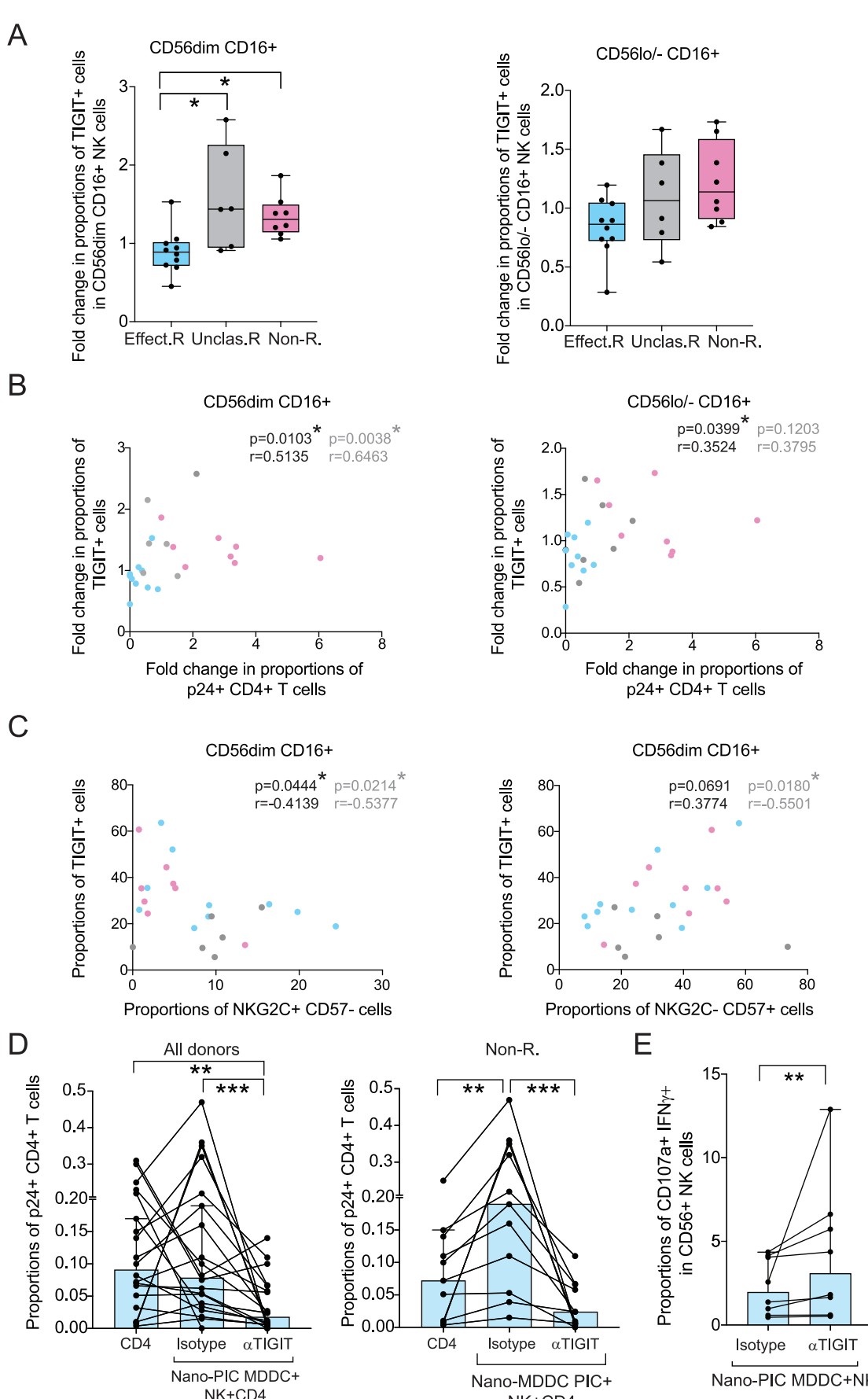

**Figure 4. Association of expression of TIGIT with memory NK cell subset proportions in response to Nano-PIC-MDDC and functional restoration after checkpoint receptor blockade.**

(A) Fold change in proportions of TIGIT+ cells compared to basal in CD56dim (left) or CD56lo/− (right) CD16+ NK subsets after Nano-PIC-MDDC from effective responder ($n = 10$; Effect. R; blue), unclassified ($n = 6$; Unclas. R.; gray) and non-responder ($n = 8$; Non-R.; pink) responder PWH. Data are represented in Box and Whiskers plots showing median values and maximum and minimum error bars. (B) Spearman correlations between fold change proportions of TIGIT+ cells within CD56dim (left) or CD56lo/− (right) CD16+ NK and fold change proportions of p24+ CD4+ T cells after Nano-PIC-MDDC. (C) Spearman correlations between proportions of TIGIT+ cells and proportions of NKG2C+ CD57- memory NK subset (left) and NKG2C− CD57+ effector subset (right) after Nano-PIC-MDDC within CD56dim CD16+ NK. Data from effective responder, non-responders and unclassified PWH were highlighted in blue, pink and grey, respectively. $P$ and $R$ values in (B, C) considering all PWH or selected effective responder and non-responder groups are highlighted in black and grey, respectively. (D) Proportions of CD4 + T cells expressing HIV p24+ cells cultured alone or in combination with NK cultured with Nano-PIC-MDDC from $n = 20$ total PWH (left panel) and in $n = 11$ non responder individuals (right panel) in the presence of isotypic control mAb or individual treatment with anti-TIGIT. (E) Proportions of CD107a+ IFNγ + NK cells from non-responder PWH in the co-culture experimental conditions described in (D). Data are represented in Box and Whiskers plots showing median values and maximum and minimum error bars. Statistical significance was calculated using a Friedman or a Kruskal–Wallis tests for multiple comparisons or Spearman associations. Statistical $P$ and $R$ values considering all data (black) and without unclassified group (gray) are shown (B, C). *$P < 0.05$; **$P < 0.01$; ***$P < 0.001$. Source data are available online for this figure.

blockade of TIGIT can improve functional restoration of NK cells from non-responder PWH in combination with Nano-PIC-MDDCs.

## Cytotoxic function of adaptive NK cells from PWH primed with Nano-PIC-MDDC is mediated through TRAIL

We next investigated the mechanism of enhanced cytotoxicity observed in NK primed with Nano-PIC-MDDC. To this end, we analyzed the expression of different ligands for NKG2D, NKG2C and TRAIL receptors on HIV-1 infected CD4+ T cells from PWH (Fig. EV4A). As shown in Fig. 5A, significant higher levels of the NKG2D ligand MICa/b, the NKG2C ligand HLA-E and the TRAIL ligand (TRAIL-inducing ligand receptor 1/DR4) were observed in reactivated p24+ compared to p24- CD4 + T cells from PWH. Notably, median proportions of DR4+ cells (40.4% min 25%-max 72%) were higher than those of MICa/b+ (Median 18%; min 14%-max 27%) and HLA-E+ cells (median 27% min 20% max 50%) in HIV-1-infected CD4+ T lymphocytes (Fig. 5A). Interestingly, TRAIL was induced in both NKG2C+ cells (including significant increase in CD57- precursors and a similar trend in CD57+ mature cells) and in NKG2C- after exposure to Nano-PIC-MDDC in CD56dim CD16 + NK cells from effective responders (Fig. 5B) and similar trends were present in the CD56lo/− CD16 + NK subpopulation (Fig. EV4C). In contrast, none of the NK cell subsets from non-responder PWH were able to further increase expression of this molecule in the subsets analyzed in response to Nano-PIC-MDDC (Figs. 5B and EV4C). On the other hand, unclassified responders appeared to still be able to induce TRAIL in the most mature NKG2C+ CD57+ population included in CD56lo/− CD16+ NK subset compared with these cells in the non-responder PWH group but not with the NKG2C + CD57− precursor population (Fig. EV4C). Of note, resulting raw levels of TRAIL were significantly higher in mature NKG2C+ CD57+ adaptive NK cells from effective responders after Nano-PIC-MDDC stimulation compared to the low amount of these cells present in non-responder PWH in the most mature CD56lo/− CD16+ NK population but not in the CD56dim CD16+ NK cells subset (Figs. 5C and EV4D). In addition, proportions of TRAIL+ cells were significantly correlated with lower proportions of TIGIT+ cells in CD56dim CD16+ NK cells considering all tested PWH (black and green p values excluding the outlier), or focusing on effective vs non-effective responder individuals (gray p values) (Fig. 5D). Moreover, blockade of TIGIT led to an increase

in surface TRAIL expression on mature NKG2C+ CD56+ adaptive NK cells from non-responder PWH exposed to Nano-PIC-MDDC (Fig. 5E). In addition, higher ratios of cells expressing TIGIT versus TRAIL were present in the NKG2C− CD57+ population enriched in non-responder PWH as well as the few adaptive NKG2C+ CD57− precursor and NKG2C+ CD57+ present in this group (Figs. 5F and EV4E,F). Therefore, our results indicate a generalized defect in upregulation of TRAIL in both NKG2C+ and NKG2C− NK from non-responder PWH in response to Nano-PIC-MDDC that is associated to TIGIT upregulation and may be dependent on this receptor. To determine whether killing of NK cells primed with Nano-PIC-MDDC in the effective responder PWH group was dependent on TRAIL or other potential activating receptor such as NKG2C, we performed functional co-culture assays of NK primed with Nano-PIC-MDDC and autologous CD4+ T cells from effective responder PWH in the presence of blocking mAbs directed to each of these molecules or control isotype mAb. Notably, blockade of TRAIL selectively impaired the reduction of p24+ CD4+ T cells by total autologous NK cells and Nano-PIC-MDDC (Fig. 5G). No significant effect of anti-TRAIL mAb in proportions of p24+ cells were observed in functional assays performed with unstimulated NK cells from effective responder PWH (Appendix Table S4) in contrast to NK co-cultured with Nano-PIC-MDDC (Fig. EV4G). On the other hand, a non-significant partial inhibition of the reduction of p24+ cells was observed after the blockade of NKG2C (Fig. EV4H). In contrast, a strong association was found between higher proportions of TIM3+ cells and proportions of TRAIL+ cells both in CD56dim and CD56lo/− CD16+ NK subsets (Fig. EV4I). However, no significant effect on TRAIL expression was observed after blockade of TIM3 in NK cells from effective responder PWH stimulated with Nano-PIC-MDDC (Fig. EV4J). Together, these data indicate that enhanced functionality of NK cells from effective responder PWH is dependent on more effective upregulation of TRAIL expression in NKG2C+ cells, which is negatively associated with TIGIT induction and may be regulated by this checkpoint receptor.

## TIGIT expression increases in adaptive NKG2C + NK cells in vivo in a humanized BLT mouse model of HIV-1 infection

Based on our previous results, we next sought to determine whether maturation of adaptive NKG2C+ NK cells and TIGIT expression

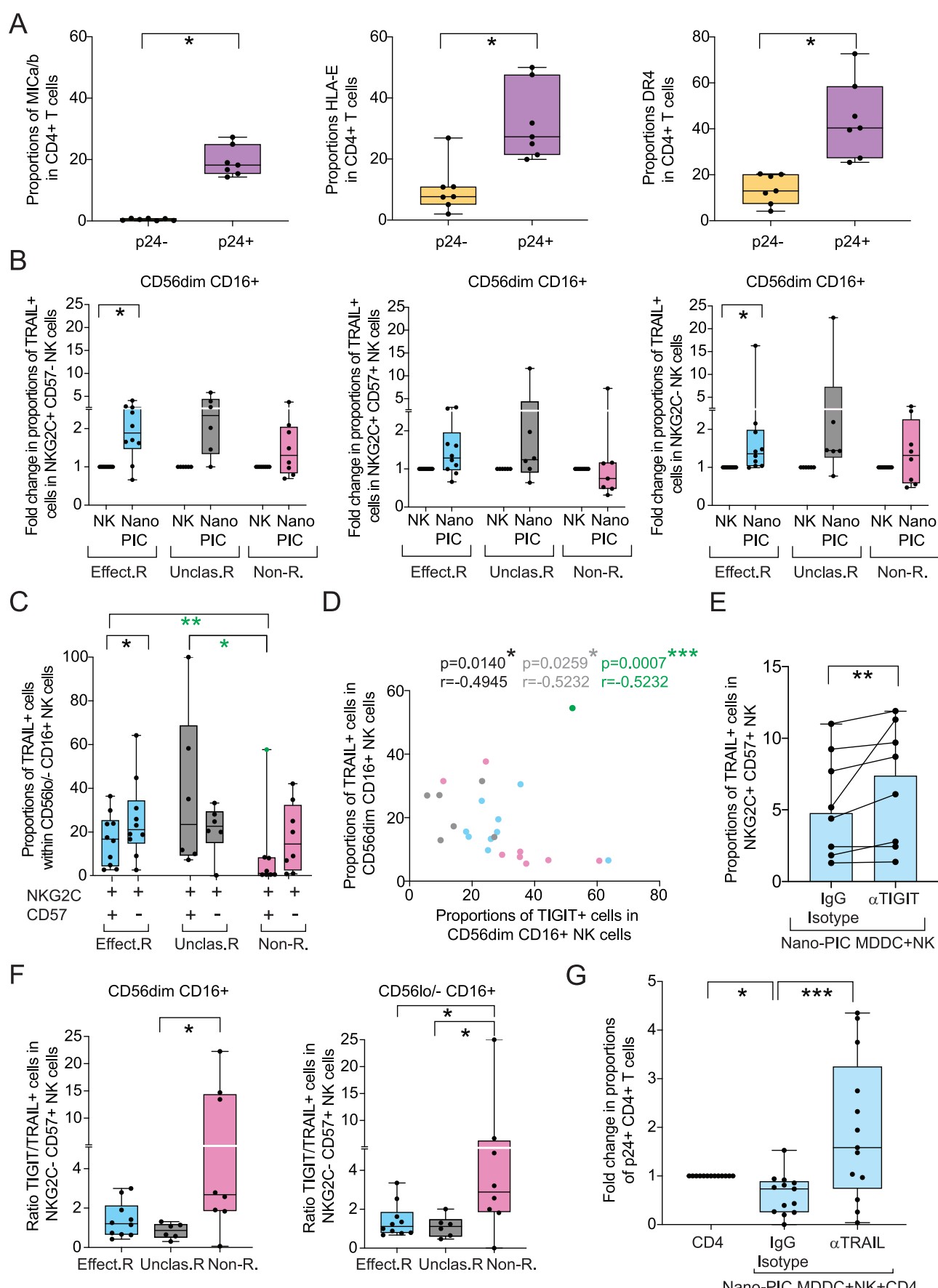

**Figure 5. Analysis of expression of ligands for NK receptors on HIV-1-infected cells and mechanisms of cytotoxic function of NK cells primed with Nano-PIC-MDDC.**

(A) Proportions of CD4+ T cells from PWH expressing the NK receptor ligands MICa/b (left), HLA-E (middle) and DR4 (right) within HIV-1 p24+ (purple) and p24- (yellow) populations in $n = 7$ PWH. Data are represented in Box and Whiskers plots showing median and maximum and minimum values in error bars. (B) Fold change proportions of TRAIL+ cells within adaptive NKG2C+ CD57− precursors (left) and mature NKG2C+ CD57+ (middle) and NKG2C- (right) NK subsets after Nano-PIC-MDDC from effective responder ($n = 10$; Effect. R; blue), unclassified responder ($n = 6$; Unclas. R; gray) and non-responder ($n = 8$; Non-R.; pink) PWH included in CD56dim CD16 +. Data are represented in Box and Whiskers plots showing median and maximum and minimum values in error bars. (C) Proportions of TRAIL+ cells within adaptive NKG2C + CD57+ and NKG2C+ CD57- NK subsets from effective responder ($n = 10$; Effect. R.; blue), unclassified responder ($n = 6$; Unclas. R; gray) and non-responder ($n = 8$; Non-R; pink) PWH included in CD56lo/- CD16 + NK subpopulation. Statistical significance was calculated using a two-tailed Wilcoxon matched pairs or a Mann–Whitney tests Statistical and Bonferroni correction was applied for multiple comparisons. Significance after removed outliers recognized by a Grubbs´test were highlighted in green. *$P < 0.05$; **$P < 0.01$. Data are represented in Box and Whiskers plots showing median and maximum and minimum values in error bars. (D) Spearman correlation between proportions of cells expressing TIGIT versus TRAIL in CD56dim CD16+ NK cells from PWH. Data from effective and non or unclassified responders PWH were highlighted in blue, pink and gray, respectively. Statistical $P$ and $R$ values considering all data (black), without unclassified group (gray) or removing outlier data recognized by a Grubb´s test (green) are shown. (E) Proportions of TRAIL in NKG2C+ CD57 + NK cells after Nano-PIC-MDDC from $n = 8$ PWH, in the presence of either IgG Isotypic control or anti-TIGIT antibodies. Statistical significance was calculated using a two-tailed Wilcoxon matched pairs test. (F) Ratio of TIGIT+ /TRAIL+ cells in NKG2C- CD57+ effector NK expressing from effective ($n = 10$; Effect. R; blue), unclassified ($n = 6$; Unclas.R; gray) and non ($n = 8$; Non-R.; pink) responders PWH in CD56dim (left) and CD56lo/− (right) CD16 + NK subsets. Statistical significance was calculated using a two-tailed Mann–Whitney test and Bonferroni correction for multiple comparisons. Data are represented in Box and Whiskers plots showing median and maximum and minimum values in error bars. (G) Fold change of p24+ CD4+ T cells from $n = 13$ PWH previously identified as effective responders treated with Raltegravir and Romidepsin and cultured in the absence or the presence of autologous NK cells treated with Nano-PIC-MDDCs and either IgG Isotypic control or with anti-TRAIL antibodies. Data are represented in Box and Whiskers plots showing median and maximum and minimum values in error bars. Statistical significance was calculated using a Friedman anova and Dunn´s post-hoc test for multiple comparisons. *$P < 0.05$; **$P < 0.01$; ***$P < 0.001$. Source data are available online for this figure.

were affected during HIV-1 infection in vivo. To this end, we analyzed the dynamics of adaptive NK cells in vivo in a humanized NOD/SCID IL2R-y−/− (NSG) mice transplanted with human BM, fetal liver, and thymus (hBLT) mouse model, which allows the reconstitution of human immune cells including NK cells and their maturation into CD16+ cells upon injection with rhIL-15 (Appendix Fig. S4A). No significant changes in body weight were observed after HIV infection during the course of the experiment in these hBLT mice (Appendix Fig. S4B). When we analyzed the reconstitution of these animals (Appendix Fig. S4C), a significant expansion of human cells and mature CD16+ CD56 + NK cells was observed upon injection with rhIL-15 (Appendix Fig. S4D,E). As expected, infected hBLT mice displayed detectable plasma HIV-1 viral loads (pVL) peaking at 2 weeks post-infection (p.i.) (Appendix Fig. S4F, left) and a subsequent reduction in proportions of CD4+ T cells within total CD3+ cells after 3 weeks was observed in infected mice (Appendix Fig. S4F, right), in agreement with kinetics reported in previous studies with this model (Brainard et al, 2009). We then studied the evolution and maturation of adaptive NKG2C+ NK cells analyzing co-expression of CD57 in this HIV-1-infected hBLT mice (Appendix Fig. S4C). An early expansion of NKG2C+ CD57- adaptive NK precursors within CD56dim CD16+ population tended to occur at 1 week post-infection and this subset was subsequently reduced at 2 weeks in HIV-1 infected mice (Fig. 6A). Interestingly, proportions of fully mature NKG2C+ CD57+ NK within circulating CD56dim CD16+ NK cells followed opposite dynamics and were progressively and significantly increased after HIV-1 infection specially at 2 weeks p.i. compared with uninfected mice (Fig. 6B). These data suggested induced maturation of NKG2C+ NK cells after HIV-1 infection in humanized mice. We next analyzed expression of immune-checkpoint receptors TIM3 and TIGIT in NKG2C+ CD57- precursors in this model. We observed an increase of TIGIT expression on adaptive NK precursors at 2 weeks p.i. that tended to correlate with the highest peak in pVL in HIV-1 infected BLT mice (Fig. 6C; Appendix Fig. S4F,G). In addition, reduced ratios of TIM3+ /TIGIT+ cells were observed from 1 to 2 weeks p.i. suggesting a loss in TIM3 expression and progressive enrichment of TIGIT in NKG2C+ CD57− adaptive NK cell precursors from

HIV-1-infected mice (Fig. 6D; Appendix Fig. S4G). Thus, TIGIT expression is increased during maturation of adaptive NK cell precursors in vivo in the HIV infected hBLT mice model.

## Combination of Nano-PIC-MDDC/NK immunotherapy and TIGIT blockade reduces expansion of HIV-1-infected cells and associates with increased cytotoxic markers on NK in humanized mice

We next assessed whether combined TIGIT blockade and Nano-PIC-MDDC immunotherapy could potentiate the ability of primary NK cells from PWH to eliminate autologous HIV-1-infected CD4+ T cells in vivo in the absence of CD8+ T cells. To this end, we used a different humanized mouse model based on immunodeficient NSG mice transplanted with pre-isolated CD4 + T cells from PWH by intravenous injection. Mice were split in three different groups of either animals left untreated ($n = 12$) to allow expansion of human infected CD4+ T cells or injected intravenously with autologous Nano-PIC-MDDC and NK cells and receiving intraperitoneal injections of either anti-TIGIT ($n = 14$) or Isotype ($n = 14$) mAbs (Fig. 7A). Three independent in vivo experiments using cells from three different PWH donors were performed (Appendix Table S5). In agreement with these previous studies (Flerin et al, 2019), human CD45+ cells from PWH were detected in the blood and in the spleen of xenotransplanted NSG mice (Fig. EV5A,B). Interestingly, a preferential distribution of human cells in the spleen (Fig. EV5B, right) versus the peripheral blood (Fig. EV5B, left) was observed in mice treated with anti-TIGIT mAbs in contrast to those receiving isotype controls. Consistently, numbers of hCD4+ T cells tended to be higher in the spleen in animals treated with Nano-PIC-MDDCs and NK cells in combination with anti-TIGIT mAbs (Fig. EV5B, right plot). Detection of viremia in the NSG mouse viral outgrowth assay (mVOA) model is quite variable and may rely on the initial frequencies of infected CD4+ T cells transplanted, the level of expansion of these cells mediated by mechanisms such as graft versus host disease (GvHD). As expected, we were not able to detect HIV-1 viral loads over the limit of detection in the plasma in transplanted NSG mice characterized by low reconstitution of human cells (Fig. EV5C). Alternatively, to assess potential effects on the selective elimination of HIV-1 infected CD4+ T cells in different body compartments, we analyzed the

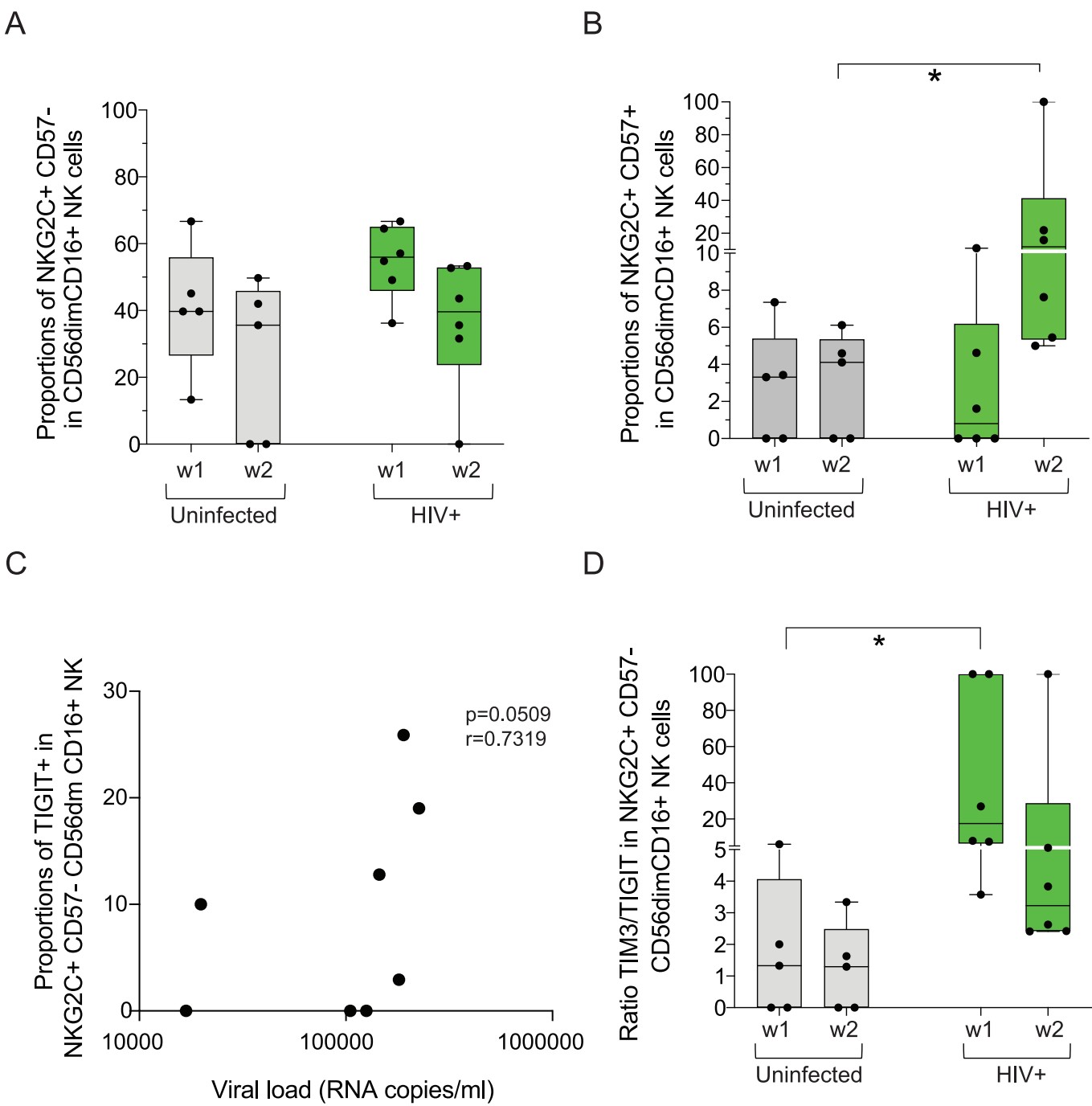

**Figure 6. Dynamics of adaptive NKG2C + NK cell subsets and immune-checkpoint expression in a humanized BLT mice during HIV-1 infection.**

(A, B) Proportions of precursor adaptive NKG2C+ CD57− (A), mature adaptive NKG2C+ CD57+ (B) subsets within circulating CD56dim CD16+ NK populations in $n = 6$ hBLT mice after 1 and 2 weeks of infection with HIV-1 JRCSF (green). Data from $n = 5$ uninfected hBLT mice (gray) analyzed in parallel at the same time points are also shown. Data are represented in Box and Whiskers plots showing median and maximum and minimum values in error bars. (C) Spearman correlation between expression of TIGIT in circulating NKG2C+ CD57- precursor within CD56dim CD16+ NK and plasma viral load in $n = 8$ infected hBLT mice after 1 week. (D) Ratio between proportions of TIM3+ vs TIGIT+ in adaptive NKG2C+ CD57- precursors in hBLT model at 1 and 2 weeks post-HIV-1 JRCSF infection, as previously defined in (A, B). Data are represented in Box and Whiskers plots showing median and maximum and minimum values in error bars. Statistical significance was calculated using a Mann–Whitney test and Bonferroni correction for multiple comparisons. *$P < 0.05$. Source data are available online for this figure.

intracellular expression of p24 in in human CD4 + T cells form peripheral blood and spleen cells in infected NSG mice by flow cytometry (Fig. EV5D). In this regard, we detected a notable increase up to a median of 23.35% of p24+ cells in human circulating CD4 + T cells from the transplanted NSG mice that did not receive any immunotherapy, suggesting an expansion of infected cells in peripheral blood (Fig. 7B, left). Moreover, the treatment of mice with Nano-PIC-MDDC and NK cells in the presence of isotypic controls tended to reduce proportions of p24+ cells within circulating CD4+ T cells, suggesting that immunotherapy by itself may be able to reduce expansion of infected cells in peripheral blood, although differences did not reach statistical significance (Fig. 7B, left panel). However, a higher proportion (42.85%) of animals exhibiting undetectable p24 expression within CD4+ T cells from peripheral blood was present in the anti-TIGIT group reaching statistical significance compared to the control mice transplanted only with CD4+ T cells (Fig. 7B, left panel). When we analyzed levels of intracellular p24+ cells in human CD4+ T cells present in tissue, we observed again a trend of lower levels (median percentage 2.21%) of p24+ cells the NSG treated with the combination of Nano-PIC-MDDC and NK cells with the anti-TIGIT mAb immunotherapies compared to mice treated with isotypic Abs (median 4.93%) or animals transplanted with individual CD4+ T cells (median percentage 6.6%) (Fig. 7B, right). These effects in peripheral blood and spleen were observed in the separate experiments using different donors in these in vivo experiments (Fig. EV5E).

The mVOA model allows a good visualization of cluster on human HIV-1-infected cells in the lymphoid tissue of transplanted mice (Flerin et al, 2019). To corroborate whether differences in frequencies of p24+ cells in mice treated with the combination of Nano-PIC-MDDC-NK anti-TIGIT mAbs could be associated to control of spread of infected cells in tissue, we assessed the size of p24+ cell clusters in the spleen by histology of the different transplanted NSG mice groups (Figs. 7C, left and EV5F). Notably, we observed a significant reduction in the size of infected cell clusters in the NSG mice group treated with combined Nano-PIC-MDDC-NK and the anti-TIGIT mAb combined immunotherapy compared to mice treated with just Nano-PIC-MDDC-NK and the Isotype control Ab (Fig. 7C, right). These effects were consistently observed in experiments performed with the three different PWH donors (Fig. EV5G). In this regard, we analyzed whether there was an association between the presence of cytotoxic NK cells and the reduction of p24+ cells in the spleen of transplanted NSG mice treated with anti-TIGIT mAbs. We detected a significant enrichment of granzyme B+ p24− cells around areas of high p24 expression in the spleen of mice treated with anti-TIGIT compared to isotype antibody treated mice (Figs. 7D and EV5F), suggesting a higher recruitment of cytotoxic NK cells in areas of high viral replication. Together, these results indicate that the combination of Nano-PIC-MDDC and NK cells and anti-TIGIT mAb immunotherapy more efficiently reduces the spread of HIV-1 CD4+ T cells in peripheral blood and tissue and induction of cytotoxic NK cells in humanized mice.

### Combined Nano PIC-MDDC/NK and anti-TIGIT immunotherapy preserves adaptive NK cell precursors expressing higher levels of TRAIL in lymphoid tissue from humanized mice

Finally, we analyzed proportions of NKG2C+ adaptive NK cells in the tissues from NSG mice transplanted with CD4+ T cells from

PWH and the combinations of Nano-PIC-MDDC-NK with anti-TIGIT or Isotype mAbs (Appendix Fig. S5A). Interestingly, we also detected a significant expansion in frequencies of total adaptive NKG2C+ cells in tissue from transplanted mice compared to baseline levels present prior to injection in most effective responder PWH donors used in these experiments (36.4% donor 1; 70% donor 2; 30.2% donor 3) (Appendix Fig. S5B). Moreover, we observed a significant enrichment of NKG2C+ CD57- precursors within the total pool of adaptive NK cells in the spleen after anti-TIGIT mAb treatment in detriment of fully mature NKG2C+ CD57 + NK cells that were enriched in the animals treated with immunotherapy and isotype mAb control and in pre-mVOA samples (Fig. 8A; Appendix Fig. S5A,B). Last, a higher proportion of NKG2C+ cells co-expressing TRAIL were detected in the spleen from mice treated with the combination of Nano-PIC-MDDC/NK anti-TIGIT mAb immunotherapy which tended to negatively associate with HIV-1 p24+ cluster size in these animals (Fig. 8B,C). Thus, reduction of the spread of HIV-1-infected CD4 + T cells in vivo associates with preserved frequencies of NKG2C+ CD57- adaptive NK precursors and TRAIL expression in the tissue.

## Discussion

The achievement of a remission of HIV-1 infection requires either the complete elimination or significant reduction of persistent HIV-1 reservoirs leading to either a sterilizing or functional cure of the infection, respectively. To this end, multiple strategies are being explored including trapping remaining viral reservoirs and preventing new replication (block and lock) (Acchioni et al, 2021; Li et al, 2021) or to potentiate the immune response against latently infected cells (Shock and Kill, Kick and kill) (Abner and Jordan, 2019; Campbell and Spector, 2022; Acchioni et al, 2021). Previous studies have shown that modulation of the immune responses against HIV-1 in PWH even on ART has, in many cases, not been successful due to an exhausted hyporesponsive state of immune cells in PWH. This is characterized by increased expression of checkpoint receptors and the lack of response upon antigenic or cell-to-cell stimulation (Mu et al, 2024; Bhattacharyya et al, 2023; Fenwick et al, 2019). The present study addresses the potential of MDDC immunotherapies increasing cytotoxic activity of NK cells from PWH against autologous HIV-1 infected CD4+ T cells. It is relevant to mention that improvement of MDDC generated from circulating monocytes (Mo) for immunotherapy is relevant for the field since it may address technical issues relative to using of less frequent but highly functional primary cDC2 and cDC1 cells which are generally depleted in PWH (Barron et al, 2003). Our data show that modulation of MDDC with Nano-PIC increased their ability to promote the expression of cytotoxic markers and natural cytotoxicity function on NK cells in vitro, and that the signals provided by Nano-PIC-MDDC to NK cells require cell-to-cell contact. Moreover, despite higher basal activation is present in MDDC from PWH consistent with previous reports in primary DC from these individuals (Smed-Sörensen et al, 2004), treatment of these cells with Nano-PIC further increased surface levels of CD86/ CD40 and induced expression of the NKG2D ligands MICa/b and UBLP1, thereby supporting a potential interaction mediated via NKG2D-ligand. In a different physio-pathological scenario, previous data showed that activation of the intracellular sensor RIG-I

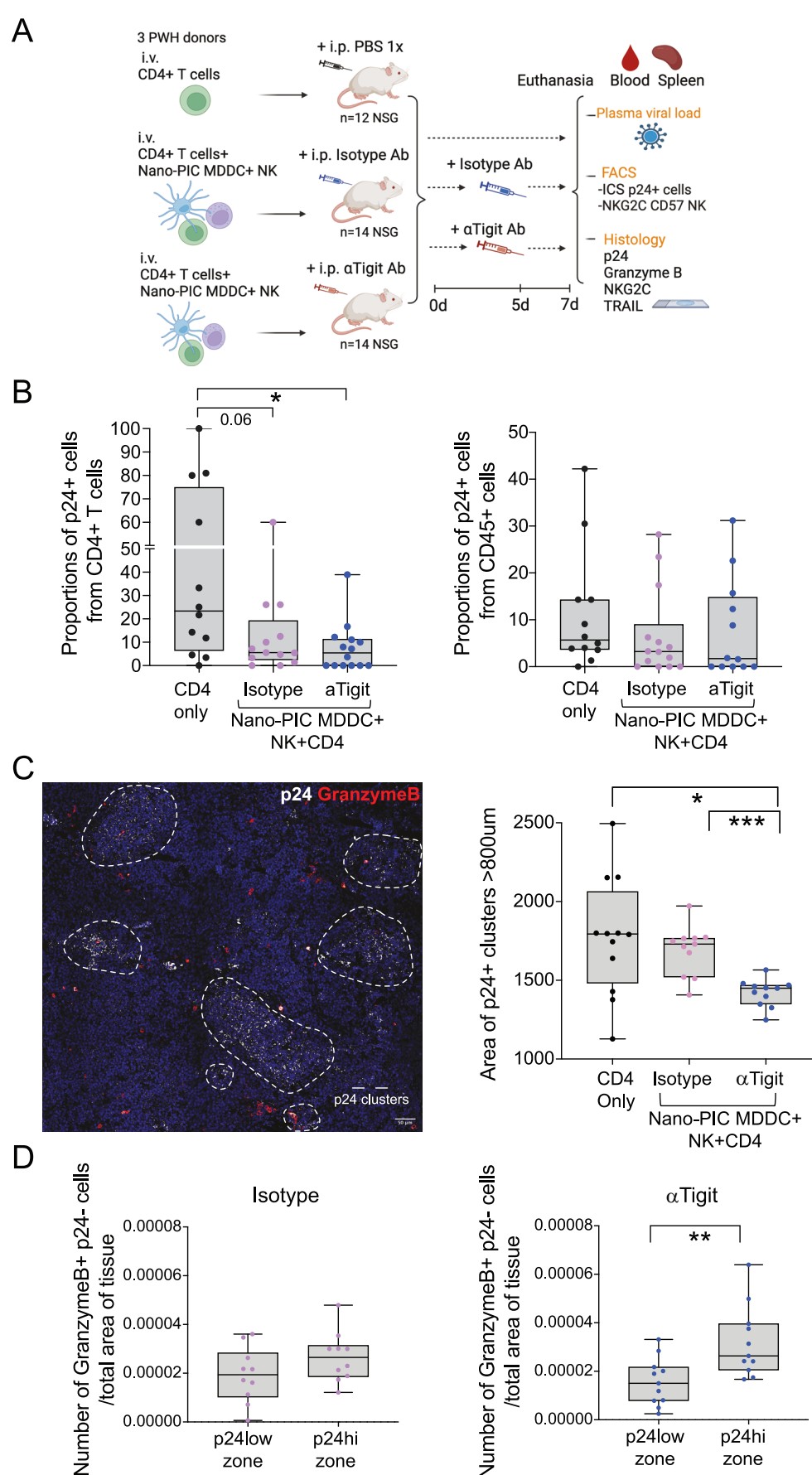

**Figure 7. Blockade of TIGIT enhances NK-NanoPIC-MDDC immunotherapy against HIV-1-infected CD4+ T cells from PWH in vivo in transplanted NSG mice.**

(A) Schematic representation of experimental humanized NSG mouse model and the different NanoPIC-MDDC-NK immunotherapy, and treatment with either Isotypic control and anti-TIGIT mAbs. Number of total combined mice from $n = 3$ independent experiments. (B) Proportions of HIV-1 p24+ cells within CD4 + T cells in the peripheral blood ($n = 12$, left plot) and spleen ($n = 12$, right plot) of NSG mice transplanted with CD4 + T cells from PWH alone (black) or in combination with autologous NK cells and Nano-PIC-MDDCs and injected with either Isotypic ($n = 13$ blood and $n = 14$ spleen; violet) or anti-TIGIT ($n = 14$ blood and $n = 12$ spleen; blue,) mAbs. Data are represented in Box and Whiskers plots showing median and maximum and minimum values in error bars. (C) Analysis of size of clusters of p24+ cells present in the spleen of transplanted NSG mice with CD4+ T cells from PWH alone ($n = 12$; black) or in combination with autologous NK cells and Nano-PIC-MDDC and injected with either isotypic ($n = 11$; violet) or anti-TIGIT ($n = 11$; blue) mAbs. A representative confocal microscopy image showing expression of p24 (white) and granzyme B (red) is shown in the left. p24 cluster areas in tissue are highlighted with white circles. Scale bar: 50 μm. (D) Quantification of granzyme B+ cells around and within areas of low and high p24+ clusters in spleen from transplanted NSG mice receiving NK-Nano-PIC-MDDC and isotype ($n = 10$; violet) or anti-TIGIT ($n = 11$; blue) immunotherapy corrected by the total area of tissue analyzed. Data are represented in Box and Whiskers plots showing median and maximum and minimum values in error bars. Statistical significance between different experimental groups was calculated using a two-tailed Mann–Whitney test and Bonferroni correction for multiple comparisons. *$P < 0.05$; **$P < 0.01$; ***$P < 0.001$. Source data are available online for this figure.

induces expression of the NKG2D ligand MICa/b on cDC2 and this interaction is needed to induce aberrant activation of NK cells in the autoimmune Sjogren´s Syndrome (Sánchez-Cerrillo et al, 2023). Notably, Nano-PIC treatment did not alter cell viability of MDDC from PWH while a trend to lower proportions of live cells was observed using cells from HD, potentially due to different sources of Mo using fresh versus buffy coat peripheral blood samples, respectively. Upregulation of HLA-E in Nano-PIC-MDDC may also be important for activation of preexistent NKG2C+ adaptive NK from PWH, consistent with previous studies on HIV-1 and CMV (Jost et al, 2023; Phan et al, 2022). In addition, although cytokines secreted by MDDC alone were not sufficient to induce activation of NK cells, it is known that they play a role in NK cell maturation and education (Bödder et al, 2020; Jacobs et al, 2021). In fact, IL-15 secreted by MDDC has been involved in the activation of NK cells (Uyangaa et al, 2023), and in our model we observed a trend to higher secretion of this cytokine in culture supernatants, which could also contribute to reinvigorating NK cell activation in PWH (Moreno-Cubero et al, 2024). Instead, Nano-PIC-MDDC secreted higher intracellular levels of IL-12 which also contribute to modulate NK cell survival and activation (Cooper et al, 2004). In the case of MDDC from PWH, we confirmed that Nano-PIC treatment increased IL-12 expression as well as secretion of other inflammatory cytokines such as TNFα, IL-2 and IL-6. Interestingly, IL-12 has been shown to be required for expansion of adaptive NK cells in response to murine cytomegalovirus infection (Sun et al, 2012), while a stimulatory effect of TNFα (Khan et al, 2023) and potential inhibition of cytotoxic activity of NK cells by IL-6 (Wu et al, 2019; Lee et al, 2023) has been reported. Moreover, we have observed no further induction of IFNβ in MDDC from PWH in response to Nano-PIC treatment, which is in accordance to previous studies reporting a defect in IFN I induction in MDDC from HIV+ individuals (Tan et al, 2005). However, Nano-PIC treated MDDC from PWH induce a significant increase in IFNγ secretion, which is key for NK cell activation and maturation into cytotoxic cells (Lin et al, 2021) and promotes the expression of ligands for activating NK receptors in target cells (Aquino-López et al, 2017). Therefore, our data support that Nano-PIC promotes secretion of immunomodulatory cytokines in MDDC which could help to reverse dysfunctional NK cells in PWH. Nevertheless, we have shown that the cell-to-cell contact between MDDC and NK cells is required for activation of NK cells. Thus, further investigation of the direct effect of each of these individual cytokines secreted by Nano-PIC-MDDC or the impact on NK receptor ligand expression induced in these cells modulating NK

cells from PWH is needed. It is important to acknowledge some potential limitations of the Nano-PIC-MDDC immunotherapy. A first limitation to consider is the potential effect of Nano-PIC-MDDC on uninfected cell populations in our in vitro studies which may also have contributed to the changes observed in the low frequencies of p24+ cells detected in the in vitro functional assays. In addition, although we did not observe an evidence for infection of MDDC defined by consistent detection of p24+ cells in these cells in our in vitro assays, we cannot completely exclude that MDDC may also become infected and favor viral spread in the mentioned assays, as previously suggested (Rhodes et al, 2021; Sanders et al, 2002; Laguía et al, 2024). In addition, our criteria of effective and non-responder responder PWH was based on ability of NK cells to reduce frequencies of p24+ cells and levels of NKG2C after treatment with Nano-PIC-MDDC compared to proportions of infected cells after LRA treatment in our in vitro assays. Such criteria did not specifically consider donors in which individual NK treatment was effective in the absence of stimulation and became dysfunctional after Nano-PIC-MDDC treatment, bringing the question of whether in some cases Nano-PIC-MDDC could also over-activate or cause exhaustion of NK cells. Also, we cannot exclude that Nano-PIC-MDDC treatment also affected non-infected cell populations. Therefore, additional studies addressing these possibilities and potentially using ex vivo infection systems will help to better understand specificity of the Nano-PIC-MDDC treatment. Nevertheless, we have been able to observe consistent decrease of intact HIV-1 proviruses in those tested cases of in vitro assays where a decrease in frequencies of intracellular p24+ cells was present, suggesting the selective targeting of infected cells capable of producing replication-competent viruses. However, IPDA has been linked to viral rebound in previous studies (Lu et al, 2018), this type of analysis could underestimate the size of the persistent HIV-1 reservoir (Kinloch et al, 2021). To better understand the impact of our in HIV-1 reservoirs, additional studies using additional in vitro and in vivo viral outgrowth techniques and an extended analysis of intact viral sequences using isolated CD4 + T cells from a larger cohort of PWH are needed.

Our data also provide novel information about enhancement of adaptive NKG2C+ NK cells in response to Nano-PIC-MDDC immunotherapy. While some studies support a protective role of these cells during HIV-1 infection in HIV-1 post-treatment controllers (Climent et al, 2023) and in the simian immunodeficiency virus (SIV) primate model (LaBonte et al, 2006), there was no previous information regarding the modulation of this NK cell subset in response to DC immunotherapy. In this regard, our data

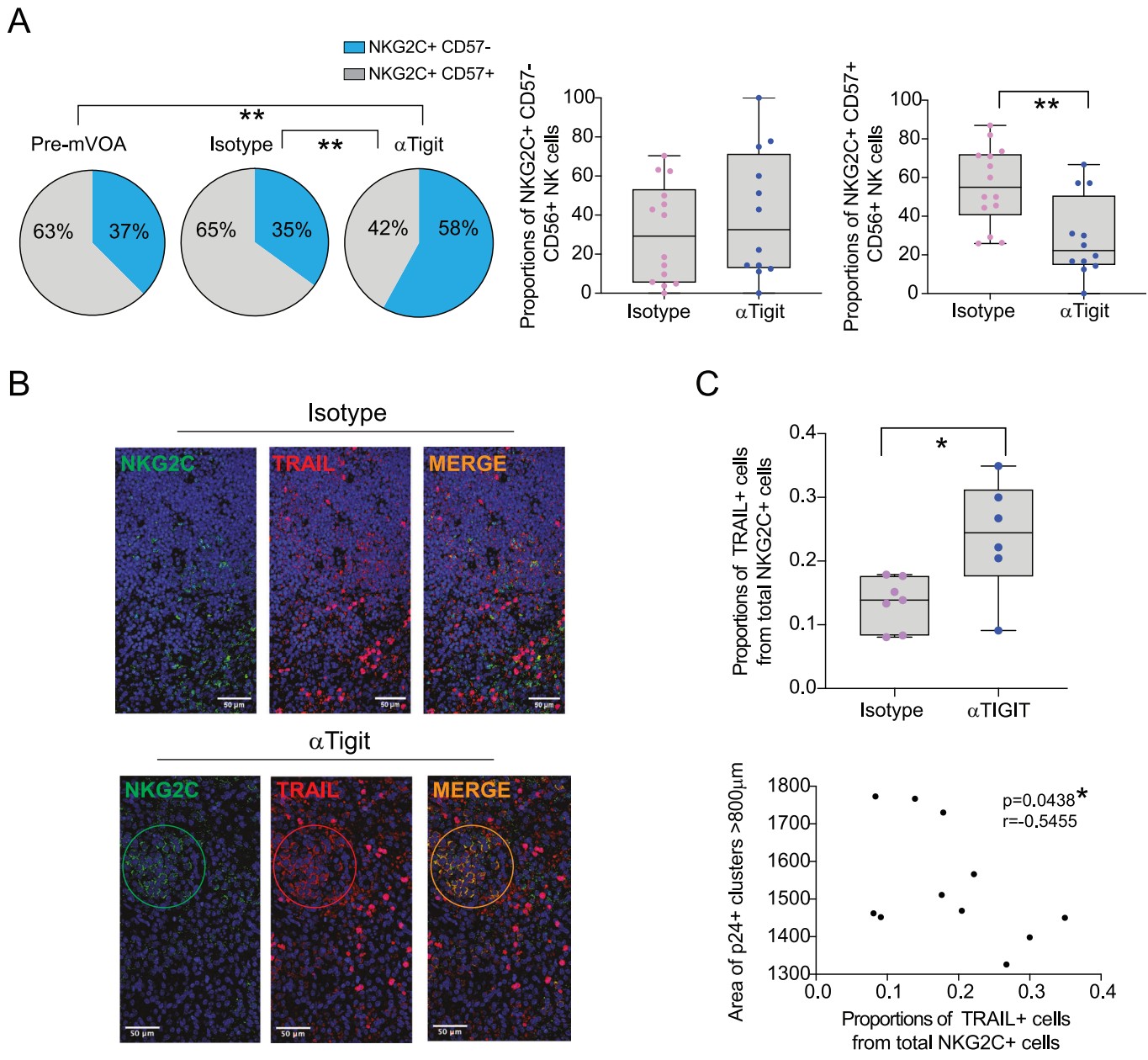

**Figure 8. Enrichment of NKG2C adaptive NK precursors and expression of TRAIL involved in HIV-1-infected CD4+ T cells elimination after NK-NanoPIC-MDDC immunotherapy and blockade of TIGIT in a in vivo PWH transplanted NSG mice.**

(A) Analysis of proportions of NKG2C+ CD57- precursors and NKG2C+ CD57+ mature memory NK cells from hCD45+ CD56+ cells present in the spleen of humanized NSG mice treated with isotype ($n = 14$; violet) or anti-TIGIT ($n = 12$; blue) antibodies. Data are represented in Box and Whiskers plots showing median and maximum and minimum values in error bars. Pie chart representing quantification of proportions of each population within the total NKG2C+ population is also shown on the left. Median proportions of adaptive NK subsets present on PWH donors prior to mVOA (Pre-mVOA) was included. (B) Representative ×40 magnification confocal microscopy images showing co-expression of NKG2C (green) and TRAIL (red) individually and merged in the spleen of NSG mice receiving anti-TIGIT and isotypic control Abs. Scale bar: 50 μm. (C) Quantification of NKG2C and TRAIL colocalization in the spleen of humanized NSG mice treated with isotype ($n = 7$; violet) or anti-TIGIT ($n = 6$; blue) antibodies (upper plot) and spearman correlation of proportions of NKG2C+ TRAIL+ cells and p24+ cluster size in the spleen of both Isotype and anti-TIGIT mice combined (lower plot). Data are represented in Box and Whiskers plots showing median and maximum and minimum values in error bars. Statistical significance between different experimental groups was calculated using either a two-tailed Mann–Whitney test or a Chi-square test. *$P < 0.05$; **$P < 0.01$. Source data are available online for this figure.

provide new evidence regarding the association of NKG2C+ adaptive NK cells and the effective elimination of HIV-1 infected CD4+ T cells in functional assays with these cells. Previous reports associated increased proportions of adaptive NKG2C+ NK cells to

CMV infection (Muntasell et al, 2013). Consistently, we observed an almost significant tendency to higher co-infection with CMV in PWH capable of more efficiently inducing adaptive NKG2C+ cells and improving functionality after exposure to Nano-PIC-MDDC.

These findings are consistent with previous studies that reported an expansion of NKG2C+ adaptive NK cells during acute CMV and HIV infections that is mainly driven by viremia (Ram et al, 2020; Shah et al, 2018). Therefore, CMV co-infection could represent a biomarker of individuals that received a priming on NK cells that could be used as a criterion to select PWH candidates for NK immunotherapy against HIV-1. However, this possibility should be investigated in more detail in preclinical studies.

Several studies have already described that adaptive NKG2C+ NK cells display higher ADCC function (Cruz-Muñoz et al, 2019). Our results also support increased ADCC activity in the presence of bNAbs directed to conserved HIV-1 envelope sites such as VCR01; 3BNC117 (CD4 binding site); PGT121 (V3 glycan site) that have been previously shown to be effective neutralizing most HIV-1 strains in combination (Thavarajah et al, 2024; Edupuganti et al, 2025; Frattari et al, 2023) which could be relevant in potentiating this function in adaptive NK cells (Sánchez-Gaona et al, 2024). However, we did not specifically address changes in total or adaptive NK cells from PWH in our study and this possibility deserves further exploration. our study identifies a TRAIL-dependent mechanism of natural cytotoxicity potentially also operating in NK from effective PWH, including adaptive NKG2C+ NK cells. Moreover, our data may also suggest that TRAIL is also efficiently induced in NKG2C- NKs from effective responder PWH after Nano-PIC-MDDC (although less efficiently than NKG2C + NK) and could play a role in preserving the cytotoxic function. Therefore, TRAIL contribution to function of these two populations should be further explored by separately analyzing the impact of blockade in isolated NKG2C+ and NKG2C- cells in these good responder PWH. Interestingly, TRAIL has previously been involved in the molecular mechanism allowing elimination of HIV-1 infected CD4+ T cells by NK cells ex vivo and controlling the reservoir size in humanized mice (Höfle et al, 2022; Cheng et al, 2020; Chandrasekar et al, 2022) and increased expression of this molecule had been reported in IL-15 expanded NK cells from PWH (Macedo et al, 2022). Also, TRAIL has been proposed as a mechanism of killing utilized by tissue-resident NK cells in salivary glands in autoimmune Sjogren´s syndrome (Schuster et al, 2023) and autoimmune diabetes (Saeki et al, 2019). However, the role of TRAIL had not been previously investigated in adaptive NKG2C+ NK cells or in the context of immunotherapy against HIV-1.

Another important aspect of this study is the identification of TIGIT and TIM3 as two different checkpoint inhibitory receptors whose inductions are differentially associated with the presence of NKG2C+ NK and the differential functional response of NK observed in the two groups PWH after Nano-PIC-MDDC stimulation. These results agree with previous studies reporting accumulation of TIGIT+ NK cells during chronic HIV-1 infection (Zhang et al, 2021a). Moreover, TIGIT expression is accumulated in mature adaptive NKG2C+ CD57 + NK cells from women with chronic HIV-1 infection (Vendrame et al, 2020). Our present results show that TIGIT is associated with dysfunctional state and with the presence of adaptive NKG2C+ NK cells. In addition, we found that blockade of TIGIT improves response of NK cells to Nano-PIC-MDDC and targeting of HIV-1 infected T cells in vitro and in vivo, in agreement with previous in vitro studies (Holder et al, 2021). Moreover, our data suggest that TIGIT may negatively regulate expression of TRAIL on

adaptive NKG2C+ NK from PWH, which has not previously reported. On the other hand, although TIM3 was associated with TRAIL expression and improved cytotoxic response of NK from PWH to Nano-PIC-MDDC, we did not observe a significant effect by blocking this receptor, which may suggest that expression of this molecule may be a biomarker of NK cells that retain certain level of functionality. However, the molecular mechanisms associated to the selective expression of TIGIT and TIM3 with the functional state of NK cells, or whether the blockade of TIGIT exclusively affects the generation or expansion of adaptive NKG2C+ NK cells were not analyzed in this study and deserve further studies.

Finally, our work provides in vivo data on the NSG-mVOA model demonstrating that combined blockade of Nano-PIC-MDDC and NK immunotherapy is capable of limiting expansion of HIV-1 infected CD4+ T cells. In these assays, although we did observe an expansion of p24+ infected cells within circulating human CD4+ T cells from PWH, we were not able to detect HIV-1 viral loads in transplanted NSG mice, maybe due to the low frequency of human CD45+ cells in the blood. Supporting this possibility, several studies have previously reported that the efficiency of graft of human cells in these animals combined with the via used to infuse the human cells may greatly affect the detection of plasma viral loads (Schmitt and Akkina, 2018; Metcalf Pate et al, 2015; Su et al, 2020). Our study also focuses on investigating the impact of immunotherapy at earlier time points to avoid the impact of development of GvHD mediated by NK cells and T cells from PWH. However, several studies suggest that viral loads become more detectable beyond 1-week post-transplantation (Flerin et al, 2019) and therefore, future studies should be conducted to determine whether this parameter could be applicable to our assay. Instead, our study analyzes the clusters of HIV-1 infected cells in lymphoid tissue, which had already been described in the mVOA model (Flerin et al, 2019) as a more reliable parameter to determine the level of expansion of clones of infected cells in the spleen of transplanted mice. Using this approach, we have shown that co-administration of anti-TIGIT antibodies and Nano-PIC MDDC-NK cell immunotherapy limits the size of clusters of HIV-1 infected cells in the spleen, suggesting that we may be able to limit the spread of virus in this tissue. Additional in vivo data of HIV-1 infection using an alternative hBLT mouse model show an increased relative expression of TIGIT over TIM3 taking place during maturation of adaptive NKG2C+ NK cells in humanized BLT mice infected with HIV-1 that is accompanied by increase of plasma viral load and reduced depletion of CD4+ T cells. However, in this model NK cells were not the only immune cell subsets present in this model in vivo, and animals were treated with hIL-15 to increase NK cell reconstitution (Kim and Zack, 2022), which may have also impact the generation of adaptive NKG2C+ NK cells and also T cells, as suggested in primate models (Okoye et al, 2019). Moreover, we have provided new evidence that this phenotype is associated with a higher expression of TRAIL and granzyme B on NKG2C+ cells also present in the spleen of NSG mice. Thus, taking into account these data combined with a higher enrichment in less differentiated NKG2C+ adaptive precursors, it is tempting to speculate that we were capable of preserving functional memory NK cells in the tissue in response to anti-TIGIT Abs. For instance, it has been described that tissue resident HIV-1 infected CD4+ T cells express high levels of PD-1 and TIGIT (Adams et al, 2021; Damouche et al, 2017) and therefore, may be susceptible to elimination or viral reactivation in the presence of anti-TIGIT mAb.

Although we did not observe a consistent trend to viral reactivation of CD4 + T cells from PWH in vitro, future analyses in NSG mice transplanted exclusively with HIV-1 infected CD4+ T cells should be carried out in order to completely discard this possibility. We have also observed that CD155 tends to be more expressed in reactivated p24+ infected CD4+ T cells from PWH, which may represent an additional potential mechanism by which anti-TIGIT may facilitate targeting of HIV-1 infected cells that are more resistant to elimination. However, conflicting results from previous studies (Vendrame et al, 2020; Vassena et al, 2013; Matusali et al, 2012) indicate that the upregulation of CD155 may not be homogeneous. Therefore, whether this is a significant evasion mechanism used by CD4 + T cells should be studied in more detail. In addition, in future clinical trials, the impact of anti-TIGIT antibodies on CD8 + T cells should also be considered. We previously showed that TIGIT also marks CD8 + T cells unable to respond to MDDC stimulation (Calvet-Mirabent et al, 2022) and the fate of these cells in vivo during HIV-1 infection and in response to anti-TIGIT antibodies should be assessed in the future. Interestingly, TIM3 seemed to associate with the generation of adaptive NKG2C+ NK cells in our study, but the blockade of this receptor did not result in improvement of their cytotoxic function. Therefore, future studies should focus on the impact of TIM3 on NK cell immunotherapy against HIV-1 as this molecule may display different roles in immune cell regulation (Tang et al, 2019). Despite these limitations, our study identified the expression of TIGIT on NK cells in PWH on ART as a new biomarker relevant to identify candidates with potential limited response to DC-based immunotherapy and lower induction of adaptive NKG2C+ cells and to potentially recommend a blockade of TIGIT in a personalized fashion in this particular group of individuals. Thus, TIGIT may represent an attractive candidate for immune modulation against HIV-1 (Holder and Grant, 2020). Future studies should examine in more detail the efficacy of the presented immunotherapies as well as the selective targeting of these different TIGIT, TIM3 as well as other checkpoint receptors in the humanized mouse models, using new systems allowing better NK and DC reconstitution such as human FMS-like tyrosine kinase 3 ligand (FLT3L) overexpression or administration (Pham et al, 2019; Kim et al, 2023; Iwabuchi et al, 2018). Collectively, our study has identified MDDC as an useful tool to modulate adaptive NKG2C+ NK cells as cell population with beneficial therapeutic properties and the blockade of selective checkpoint receptors as a modulator of these therapies, and therefore we have expanded the knowledge about the development of new tools to potentiate NK immunotherapy, which will be relevant for the field to develop new strategies to cure HIV-1 infection.

## Methods

### Reagents and tools table

| Reagent/resource | Reference or source | Identifier or catalog number |
| --- | --- | --- |
| **Experimental models** | | |
| EGFP-K562 | NIH Reagent Program | 116799 |
| HIV-1 HXB2 gp120 Expressing CHO cells (CHO-WT) | NIH Reagent Program | 2239 |
| HEK-293 | NIH Reagent Program | ATCC® CRL-1573 |

| Reagent/resource | Reference or source | Identifier or catalog number |
| --- | --- | --- |
| **Antibodies (dilution)** | | |
| Mouse Anti-TRAIL (5 µg/mL) | RyD Systems | MAB375 |
| Mouse Anti-human NKG2C (5 µg/mL) | RyD Systems | MAB1381 |
| IgG1 Isotype Control (1 µg/mL); (5 µg/mL) | BioLegend | 401402 |
| Mouse Anti-human TIGIT (1 µg/mL) | RyD Systems | MAB7898 |
| Goat Anti-human TIM-3 (1 µg/mL) | RyD Systems | AF2365 |
| IgG2b Isotype Control (1 µg/mL); (5 µg/mL) | BioLegend | 402202 |
| Purified Goat IgG (1 µg/mL) | SIGMA | I2136-1ML |
| Tiragolumab (anti-TIGIT) (100 µg/mouse) | Selleck Chemicals | A2028 |
| In vivo Human IgG1 isotype (100 µg/mouse) | BioXCell | BE0297 |
| Anti-human IFNb (1:100) | Pbl assay science | PBL-21400-3 #7225 |
| Anti-IL-12 (1:100) | RyD Systems | IC2191C |
| Anti-human CD86 (1:50) | Biolegend | 374210 |
| Anti-human CD40 (1:50) | Biologend | 334306 |
| Anti-human ULBP-1 (1:50) | RyD systems | FAB1380P |
| Anti-human MICA/B (1:50) | Biolegend | 320912 |
| Anti-human HLA-E (1:50) | Biolegend | 342606 |
| Anti-human IFNg (1:100) | Biolegend | 554551 |
| Anti-human CD107a (1:200) | Biolegend | 328620 |
| Anti-human TNFa (1:100) | Biolegend | 502923 |
| Ghost Dye™ Red 780 (1:100) | TONBO biosciences | 13-0865-T100 |
| Anti-human CD16 (1:50) | Biolegend | 302045 |
| Anti-human CD56 (1:50) | Biolegend | 362505 |
| Anti-human HLA-DR (1:50) | Biolegend | 307630 |
| Anti-human PD-1 (1:50) | Biolegend | 329938 |
| Anti-human CD253 (TRAIL) (1:50) | Biolegend | 308210 |
| Anti-human DR4 (TRAIL-R1) (1:50) | Biolegend | 307207 |
| Anti-human TIGIT (1:50) | Biolegend | 372734 |
| Anti-human CD45 (1:50) | BD Horizon | 560777 |
| Anti-human CD57 (1:50) | Biolegend | 359624 |
| Anti-human NKG2C (1:50) | RyD Systems | FAB138G |
| Anti-human CD3 (1:50) | Biolegend | 300306 |
| Anti-human CD4 (1:50) | Biolegend | 317429 |
| Anti-human P24 (3:100) | Beckman Coulter | 6604667 |
| Anti-human Granzyme B (1:100) | Biolegend | 563389 |
| Anti-human TIM3 (1:50) | Biolegend | 345022 |
| Anti-human CD14 (1:50) | Biolegend | 367110 |
| Anti-human CD1a (1:50) | BD | 333167 |

| Reagent/resource | Reference or source | Identifier or catalog number |
|---|---|---|
| Anti-human CD11c (1:50) | Biolegend | 301626 |
| Rabbit anti-human NKG2C (1:100) | Abcam | AB230900 |
| Goat anti-human TRAIL (15 µg/ml) | RyD Systems | AF375 |
| Mouse anti-human p24 (Clone Kal-1) (1:50) | Dako | M0857 |
| Granzyme B antibody (1:100) | Thermo Fisher | 14-8889-80 |
| Donkey anti-rabbit AF488 (1:200) | Thermo Fisher | R37118 |
| Donkey anti-rat AF594 (1:100) | Jackson ImmunoResearch | 712-586-150 |
| Donkey anti-goat AF568 (1:200) | Thermo Fisher | A-11057 |
| Donkey anti-mouse AF647 (1:200) | Thermo Fisher | A-31571 |
| **Oligonucleotides and other sequence-based reagents** | | |
| PCR primers | This study | "Methods" |
| **Chemicals, enzymes and other reagents** | | |
| Polyinosinic-polycytidylic acid sodium salt (poly I:C) | Merck- Sigma | P1530 |
| Brefeldin A | Sigma-Aldrich (Merck) | B7651 |
| Monensin | Sigma-Aldrich (Merck) | M5273 |
| Easy step human NK cell Isolation Kit | Stemcell | 17955 |
| PMA | Sigma-Aldrich (Merck) | P1585 |
| Ionomicin | Sigma-Aldrich (Merck) | I0634-1MG |
| Recombinant human IL-15 | RyD Systems | 247-ILB/CF |
| TransIT-X2 Delivery System | Mirus Bio | MIR 6004 |
| IL-2 | Stem cell | 78036.3 |
| IL-4 | Prepotech | 200-04 |
| GM-CSF | Prepotech | AF-300-03-50UG |
| PHA | Sigma-Aldrich (Merck) | L2769-5MG |
| Romidepsin | Selleck Chemicals | S3020 |
| Raltegravir | Selleck Chemicals | S2005 |
| Human Cytokine Storm Bead Array 1 Kit | RayBiotech | FAH-STRM-1-96 |
| Human Interferon Beta ELISA Kit | Bioss Antibodies | BSKH60184-96T |
| TrueCount | BD | 663028 |
| **Software** | | |
| FlowJo v10.10 | https://www.flowjo.com/flowjo/download | |
| ImageJ | https://imagej.nih.gov/ij/index.html | |
| Gradpad Prism 9.0 | https://www.graphpad.com/features | |
| QIAcuity One 2-plex System | Qiagen | |
| QuantStudio5 system | Thermo Fisher | |

| Reagent/resource | Reference or source | Identifier or catalog number |
|---|---|---|
| Rstudio | https://posit.co/download/rstudio-desktop/ | |
| **Other** | | |
| Illumina NextSeq | San Diego, CA | |
| Leica TCS SP5 confocal | Leica | |
| PT-LINK | Dako | |

## Study participants

A total of $n = 63$ aviremic PWH on ART for a median of 10.5 years (min–max; 1.5–28) and characterized by undetectable plasma viremia (< 20 copies HIV-1 RNA/mL), CD4 + T cell count higher than 500 cell/µl (median 849; min–max, 432–2406) and no co-infection with hepatitis C virus were recruited for the whole study from the Infectious Diseases Unit from Hospital Universitario La Princesa. Median age was 51 (min–max, 25-77) and 96.8% of participants were male. Additional clinical characteristics from the specific PWH cohorts used in the different experimental sets from our study are summarized in Appendix Tables S1–3. $N = 49$ HIV negative donors (HD) were obtained from Buffy coats obtained from the Centro de Transfusiones Comunidad de Madrid and were used as controls for comparison purposes.

## Generation of MDDC and stimulation with Poly I:C-loaded nanoparticles

MDDCs were generated in vitro from circulating adherent Mo from HIV negative donors and from PWH in RPMI complete culture media supplemented with 10% fetal bovine serum (FBS) in the presence of 100 IU/mL recombinant human GM-CSF and 100 IU/mL recombinant human IL-4 (Preprotech) for 6-7 days. Differentiation of MDDC from monocytes was evaluated by FACS confirming high levels of CD1a, CD11c and low expression of CD14 in cultured cells (Fig. EV1A). Subsequently, MDDCs were cultured for additional 16 h in the presence of either 1 µg/ml soluble Poly I:C (Invivogen) (Sol-PIC) or encapsulated in complexes of commercial polymeric nanoparticles (Nano-PIC) (TransIT-X2 Delivery System; Mirus Bionova) prepared using a non-supplemented Opti-MEM media according to the manufacturer´s recommendations for nucleic acid encapsulation and delivery in primary cells (Rajagopal et al, 2021; Del Toro Runzer et al, 2023). In our case, we prepared Nano-PIC complexes in the presence of 1 µg/ml Poly I:C, by incubating at room temperature for 20 min to allow complex assembling and formation. MDDC cultured in the presence of just media or empty nanoparticles, were also included for control purposes. In these experiments, phenotypical analysis of intracellular expression of IL-12 and IFNβ was performed by FACS at 6 h (see "Flow cytometry"), as well as secretion of an additional panel of immunomodulatory cytokines (IL-2, IL-6, IL-12, IL-15, TNFα, IFNγ; Human Cytokine Storm Bead Array 1 Kit) was performed at 16 h. Additional analysis of IFNβ secretion was assessed using the Human Interferon Beta

ELISA Kit (Bioss Antibodies) at 16 h. Surface expression of maturation markers CD86 and CD40 and ligands for NKG2D/A/C receptors MICa/b, ULBP-1 and HLA-E on MDDC from HIV-1 negative donors and PWH cultured under the described conditions was carried out by FACS (see "Flow cytometry").

## Immunomagnetic selection of NK cells, co-culture with Nano-PIC-MDDC and functional assays CD4+ T cells from PWH

NK cells were isolated by negative immunomagnetic selection (STEM cell) from the blood of HIV-1 negative donors and aviremic PWH on ART from our study cohort (purity >90%) and co-cultured with autologous MDDC pre-stimulated with either Sol-PIC or Nano-PIC or their control conditions at a NK:MDDC ratio of 2:1 in the presence of 50 IU/mL IL-2 (Stem Cell). In some experiments, levels of intracellular expression of IFNγ, TNFα, granzyme B and CD107a were addressed in NK from PWH cultured in the mentioned conditions. In addition, dependency of NK cell degranulation on cell-to-cell contact was tested in transwell assays by seeding NK cells in the bottom chamber alone or in combination with Nano or Nano-PIC-MDDC in the same or in an upper well separated by a 0.4 μm pore polycarbonate membrane (Corning Incorporated). Subsequently, primed NK cells from PWH were cultured with autologous CD4+ T cells isolated by negative immunomagnetic selection (DynaBeads Human CD4 + T cell isolation kit, Life technologies) at a MDDC: NK: CD4 + T cell ratio of 1:2:3 (to mimic differences in proportions of NK and T cells in the blood) in the presence of 50 nM Romidepsin (Selleck Chemicals) to induce viral reactivation and 30 μM Raltegavir (Selleck Chemicals) to prevent viral spread. After 16 h, proportions of p24+ HIV-1 infected CD4+ T cells were evaluated by flow cytometry. In some of these assays, analysis of expression of CD56 vs CD16, NKG2C and CD57 defining different subsets of adaptive and effector NK cells, checkpoint receptors (TIGIT, PD-1 and TIM3) and TRAIL within CD56dim CD16+ and CD56lo/-CD16+ populations was performed after stimulation with Nano-PIC-MDDC and associated with functional ability of NK cells to kill HIV-1 infected CD4+ T cells. In some experiments, expression of the TRAIL ligand DR4 and NKG2D/C ligands MICa/b or HLA-E, respectively, was also tested in p24− vs p24+ isolated CD4+ T cells from PWH after reactivation with 1 μg/ml of PHA and 50 IU/ml IL-2. Finally, 1 μg/mL blocking goat anti-TIM3 and 1μg/mL mouse anti-TIGIT were added to cultures of NK with Nano-PIC-MDDC. For control purposes, cells were also treated with a combination of the corresponding of 1 μg/mL goat IgG and 1 μg/mL mouse IgG isotypic mAbs. In some experiments, the impact of these mAbs in the function of individual unstimulated NK cells was also tested.

In a different set of experiments, 5 μg/mL blocking mouse anti-human TRAIL or 5 μg/mL IgG isotypic control mAbs were added during co-culture of unstimulated or Nano-PIC-MDDC treated NK cells with CD4+ T cells in the NK cell-mediated killing of autologous HIV-1-infected CD4 + T cells from PWH described above. In some NK-Nano-PIC-MDDC-CD4 + T cell functional experiments, the effect of a blocking mouse anti-human NKG2C mAb or 5 μg/mL IgG isotypic control was also assessed.

## Natural cytotoxicity and antibody-dependent cellular cytotoxicity in vitro killing assays

To address killing abilities of NK cells stimulated with Nano-PIC-MDDC, we used two in vitro assays using different target cell lines. To address natural cytotoxic function, we co-cultured unstimulated or Nano-PIC-MDDC-primed NK cells with the MHC-I deficient target cell line K562 transfected with GFP at a ratio target:NK 1:2 (NIH AIDS reagent program Ref 11699). Specific killing was determined by flow cytometry quantifying loss of GFP expression and acquisition of a viability dye. For antibody-dependent cellular cytotoxicity (ADCC) we co-cultured NK cells in the presence of the cell line CHO cell line transfected with HIV-1 gp120 selected as specified by the provider (NIH AIDS reagent program Ref 2239) at a ratio target:NK of 1:2 in the absence or after preincubation with a cocktail of three HIV-1-specific broadly neutralizing antibodies (bNAbs) (10 μg/mL VCR01; 5 μg/mL PGT121; 1 μg/mL 3BNC117) with specificities to conserved CD4 and V3 binding regions. Cell death was quantified after 16 h by acquired staining viability dye staining. ADCC activity was calculated after the subtraction of the basal natural cytotoxicity-mediated killing of the CHO target cell line by NK cells in the absence of bNAbs.

## Flow cytometry

Cell viability and differentiation from monocytes of MDDC was confirmed by FACS using Viability Dye and a cocktail of anti-human CD14, CD1a, CD11c mAbs. Maturation of MDDC was addressed 16h using anti-CD86 and in some cases also anti-CD40 mAbs. Surface expression of these molecules was also monitored at 6 h along with intracellular staining of anti-IFNβ, anti-IL-12 in the presence of 0.25 μg/mL Brefeldin A and 0.0025 mM Monensin (SIGMA). At 16h surface expression of ligands for NKG2D and NKG2A/C on MDDC from HD and PWH was tested by FACS using anti-MICa/b, anti-ULBP-1 or anti-HLA-E mAbs.

The analysis of NK cell subpopulations and their ability to secrete cytokines and degranulate was performed using a Ghost Dye Red 780 viability dye, anti-CD56, anti-CD16, in combination with either IFNγ, CD107a, Granzyme B, TNFα mAbs in presence of Phorbol myristate acetate (PMA) and Ionomycin for 1h and Brefeldin A and Monensin after 4h or NKG2D, NKG2A, NKG2C, CD57, TIGIT, TIM3, PD-1 or TRAIL mAbs (see "Reagents and tools table").

In functional assays proportion of infected CD4+ T cells was addressed by intracellular staining of p24 in combination with anti-CD3, anti-CD4 antibodies and in the presence of the mentioned viability dye. A median of 50,000 events of gated CD4+ T cells were considered to ensure detection of intracellular HIV p24 protein.

In humanized mouse experiments, a combination of anti-human CD45, CD3, CD56, CD16, NKG2C, CD57, TIGIT and TIM3 mAbs were used to analyze peripheral blood and spleen samples from transplanted mice. A median of 4800 human CD45+ cell events for humanized BLT mice experiments and 30,000–32,000 live cells for transplanted NSG mice in mVOA experiments were considered for analysis. The absolute numbers of CD45 and CD4+ T cells in blood and spleen were calculated using counting true-count beads (See reagents and tools table).

Samples were analyzed in a BD Canto II and Lyric instruments. Analysis of flow cytometry data was performed using FlowJo v10.6 software (Tree Star).

## Intact proviral DNA analysis

Intact proviral DNA assay (IPDA) was performed as previously described (Bruner et al, 2019) and using primers and probes specific for the Ψ HIV gene (HIV-1 Ψ forward 5′-CAGGACTCGGCTTGCTGAAG-3′, HIV-1 Ψ reverse GCACCCATCTCTCTCCTTCTAGC and probe 5′-6-FAM-TTTTGGCGTACTCACCAGT-MGBNFQ-3′) and env HIV gene (HIV-1 env forward 5′-AGTGGTGCAGAGAGAAAAAAGAGC-3′, HIV-1 env reverse 5′-GTCTGGCCTGTACCGTCAGC-3′, HIV-1 env intact probe 5′-VIC-CCTTGGGTTCTTGGGA-MGBNFQ-3′, and HIV-1 anti-Hypermutant env probe 5′-CCTTAGGTTCTTAGGAGC-MGBNFQ-3′). The hRPP30 gene was used for cell input normalization and to quantify DNA shearing. Samples were analyzed in a QIAcuity One 2-plex System (Qiagen). Proportions of intact HIV-1 sequences from a 100% of total HIV-1 DNA sequences detected including these intact proviruses and defective hypermutated and/or 3' deletion or 5′deleted proviruses were calculated for each experimental condition. Raw IPDA data is provided in Appendix Table S3.

## Analysis of memory NK cells and impact of combined Nano-PIC-MDDC anti-TIGIT immunotherapy in humanized mice in vivo

A total of $n = 11$ NOD/SCID IL2R-y−/− (NSG) mice transplanted with human BM, fetal liver, and thymus (BLT-mouse) and fetal CD34+ HSCs were generated at the Human Immune System Core from the Ragon Institute and Massachusetts General Hospital in collaboration with Dr. Vladimir Vrbanac and Dr. Alejandro Balazs as previously described (Brainard et al, 2009). Mice were housed in ventilated racks and fed autoclaved food and water at a pathogen-free facility. Intraperitoneal injections of 2.5 µg rhIL-15 per mouse considering a 20 g average weight (R&D systems) were performed for 3 weeks to increase reconstitution and maturation of human NK cells in hBLT mice. Human immune reconstitution was monitored for 17 weeks prior to infection with HIV-1 and optimal reconstitution was considered with proportions of human CD45+ lymphocytes over 25%. Viral stocks were generated by transfection of HEK-293 (ATCC® CRL-1573) cells with the plasmid "HIV-1 JR-CSF Infectious Molecular Clone (pYK-JRCSF)" (National Institutes of Health [NIH] AIDS Reagent Program, catalog number 2708) using the X-tremeGENE 9 DNA Transfection Reagent (Roche) according to manufacturer's instructions. 72 h after transfection, supernatants were collected, filtered, concentrated and resuspended in 1× PBS. The JRCFS viral stock was stored at −80 °C for further experiments. The virus TCID$_{50}$ was determined in TZM-bl cells (NIH AIDS Reagent Program, catalog number 8129) by limiting dilution using the Reed and Muench method, as previously described. Dr. Julia García Prado´s group at IRSICaixa, Barcelona, Spain produced and provided the viral stocks for the study. Reconstituted hBLT mice were either intravenously infected ($n = 6$) with 5000 TCID$_{50}$ of JRCSF or left uninfected ($n = 5$). Body weight, HIV-1 plasma viral load, CD4 + T cell counts were monitored for 3 weeks and proportions of memory and effector NK cell subsets in the blood were tested by FACS for 2 weeks.

To address the efficacy of anti-TIGIT mAb modulation of NK cell-Nano-PIC-MDDC immunotherapy eliminating HIV-1 infected CD4 + T cells from PWH in vivo, we performed a mVOA assay by intravenously transplanting $n = 40$ immunodeficient NOD/SCID IL2R-y−/− (NSG) mice with either CD4 + T cells from PWH ($n = 3$ different donors in three independent experiments) alone or in combination with NK cells, Nano-PIC-MDDC from the same PWH at a 1:2:4 (MDDC:NK:CD4) ratio. Mice receiving Nano-PIC-MDDC immunotherapy were split into two groups that were administered intraperitoneally simultaneously with either humanized anti-TIGIT (Tiragolumab, Selleck Chemical) ($n = 14$ from three independent experiments) or humanized IgG1 isotypic (BioXCell) ($n = 14$ from three independent experiments) mAbs. These mAbs were also administered 5 days after the initial immune cell inoculation. As control group, NSG mice were also transplanted only with CD4 + T cells from the same PWH ($n = 12$, from three independent experiments) and injected i.p. with PBS 1×. Proportions of p24+ cells in human CD45 + CD4 + T cells in the peripheral blood and spleen at 7 days post-injection were analyzed by FACS (see "Flow cytometry"). Frequencies of adaptive NKG2C + CD57− and NKG2C + CD57 + NK subsets were also tested in spleen.

## RT-qPCR quantification of Plasma HIV-1 viral load in humanized mouse models

HIV-1 RNA was extracted from plasma samples collected from either HIV-1-infected or uninfected humanized hNSG mice transplanted with CD4+ T cells from PWH in the absence or presence of immunotherapy using Nucleospin™ RNA Virus kit (Cultek, Macherey-Nagel) following the manufacturer's instructions. Reverse transcription of RNA to cDNA was performed with SuperScript™ III Kit (Invitrogen) in accordance with the instructions provided by the manufacturer, and cDNA was quantified by quantitative PCR (qPCR) using primers and probes specific for the 1-LTR region (LTR forward 5′-TTAAGCCTCAATAAAGCTTGCC-3′ and LTR reverse 5′-GTTC GGGCGCCACTGCTAG-3′; LTR probe 5′-CCAGAGTCACACACC AGACGGGCA-3′), and TaqMan™ Gene Expression Master Mix (Thermo Fisher). Samples were analyzed in an Applied Biosystems QuantStudio5 system. Quantification of DNA was performed using a standard curve, and values were normalized to 1 million CD4+ T cells.

## Histological analysis of tissue sections from humanized mice

Spleens were collected from hBLT mice and NSG transplanted mice and paraffin-embedded and segmented in fragments of 2 µm of thickness in a Leica microtome. Tissue sections deparaffinized, hydrated, and target retrieval were performed with a PT-LINK (Dako) previous to staining. For paraffin-preserved tissue, rat anti-human Granzyme B (eBioscience), mouse anti-HIV-1 P24 (Dako), goat anti-human TRAIL (RyD Systems) and rabbit anti-human NKG2C (Abcam) as primary antibodies; and goat anti-rabbit AF488 (Invitrogen), donkey anti-rat AF594 (Jackson ImmunoResearch), donkey anti-goat AF568 (Invitrogen) and donkey anti-mouse AF647 (Thermo Fisher) as secondary antibodies. Images were taken with a Leica TCS SP5 confocal and processed with the LAS AF software. Granzyme B+, TRAI+ and HIV-1 p24+ cell counts and colocalization and distribution of some of these markers

### The paper explained

#### Problem

Access to antiretroviral treatment (ART) has allowed to suppress viral replication and detectable plasma viremia in people living with HIV-1 (PWH), leading to decreased mortality and improved quality of life in this community. However, ART does not eliminate HIV-1 infection and persistent viral reservoirs of long-lived memory CD4 + T cells harboring integrated HIV-1 proviruses represent the main obstacle to achieving a cure. In addition, punctual viral reactivation, viral scape and chronic inflammation in different anatomical sites contribute to immune exhaustion, limiting the ability of immune cells, including natural Killer (NK) cells to eliminate remaining infected cells.

#### Results

This study reports the use of dendritic cells stimulated with nanoparticles loaded with Poly I:C (Nano-PIC-MDDC) as a novel approach to reinvigorate cytotoxic activity of NK cells from PWH against HIV-1-infected cells in vitro and in a preclinical in vivo model. Notably, we identify NKG2C and TRAIL as new biomarkers associated with effective response of a subgroup of PWH to Nano-PIC-MDDC treatment. Furthermore, we identify TIGIT as an additional biomarker for dysfunctional unresponsive NK cells and blockade of this receptor enhances efficacy of Nano-PIC-MDDC stimulation in non-responder PWH.

#### Impact

The presented study identifies new combined strategies based on Nano-PIC-MDDC-primed NK cells and TIGIT blockade that can be used to potentiate elimination of persistent HIV-1 infected cells in a personalized fashion, allowing to improve antiviral immune responses in a broader range of PWH with different immunological characteristics, which could be useful to reduce viral reservoirs and to advance field toward an HIV-1 cure.

were analyzed with ImageJ software. P24+ clusters were defined as areas in tissue defined by a proximity of adjacent p24+ cells of 0.5 microns using GaussianBlur and the watershed functions from the ImageJ software. Size of selected clusters was determined by measuring the median area values for all combined clusters over 800 microns in each group.

## Statistical analysis

A ROC-Curve analysis was used to statistically estimate NKG2C levels defining PWH groups with differential functional ability to eliminate HIV-1-infected CD4 + T cells below (fold change p24+ cells <1) or above (fold change p24+ cells >1) from baseline levels with the support of Dr. Nuria Montes from Unidad Apoyo Metodológica-Instituto Investigación Sanitaria at Hospital de la Princesa. Statistical significance of differences between cells from different or within the same patient cohort under different treatments were assessed using a two-tailed non-parametric Mann–Whitney $U$ or Wilcoxon matched-pairs signed-rank tests, respectively. Multiple comparison corrections were applied using the Bonferroni test or either the One Anova Kruskal–Wallis or Friedman and Dunn´s post-hoc tests for unpaired and paired datasets, respectively. In some analyses, a Grubb´s test was used to identify outliers in datasets and take this information as an exclusion criterion for statistical significance calculation. Chi-square with Yate's correction was used to compare differences in proportions of some parameters within different cell/subject populations. Non-parametric Spearman correlations were used to test both individual correlations and to generate a correlation network. Statistical analyses were performed using GraphPad Prism 9.0 software. No blinding was applied to our data analysis.

## Ethic statement

All subjects participating in the study gave written informed consent, and the study was approved by the Institutional Review Board of Hospital Universitario de La Princesa (Protocol Registration Number 3518) and following the WMA declaration of Helsinki and the Department of Health and Human Services Belmont Report. For in vivo experiments, mice were housed at the BSL3 animal facility from Centro de Biología Molecular Severo Ochoa (CBM) in accordance with the institution's animal care standards. Animal experiments were reviewed and approved by the local ethics committee and were in agreement with the EU Directive 86/609/EEC, Recommendation 2007/526/EC and Real Decreto 53/2013.

## Data availability

This study includes no data deposited in external repositories.

The source data of this paper are collected in the following database record: biostudies:S-SCDT-10_1038-S44321-025-00255-x.

## Peer review information

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

## Acknowledgements

EMG was supported by Ramón y Cajal Program (RYC2018-024374-I), the Spanish Agencia Estatal de Investigación RETOS, Generación de conocimiento and consolidation programs (RTI2018-097485-A-I00; PID2021-127899OB-I00; CNS2023-144841), La Caixa Banking Foundation ETI-CureHIV (HR20-00218), GLD24/00117 grant from Gilead Biosciences and infectious diseases CIBER (CIBERINFEC) from ISCIII (CB21/13/00107). MAL was supported by the Formación de Personal Investigador (FPI) grant PRE2022-104516. IT was supported by FPI UAM fellowship. CDA was supported by Comunidad de Madrid Talento Program (2017-T1/BMD-5396). MCM was supported by La Caixa Foundation ETI-CureHIV (HR20-00218). P2022/BMD7209-INTEGRAMUNE from Comunidad Autónoma de Madrid and La Caixa Health Research Grant LCF/PR/HR23/52430018 to and PID2023-149541OB-I00 FSM also supported the study. ISC was supported by infectious diseases CIBER from ISCIII (CB21/13/00107). JGP was supported by La Caixa Health program ETI-CureHIV project (HR20-00218), the CIBERINFECC from the National Health Institute Carlos III, and the project PI22/01120. MLT was supported by the Spanish Agencia Estatal de Investigación, Ministerio de Ciencia, Innovación y Universidades (PID2022-138880OB-I00 and PDC2021-121238-I00). NMC contributed to ROC Curve analysis.

## Author contributions

**Ildefonso Sánchez-Cerrillo**: Conceptualization; Data curation; Formal analysis; Validation; Investigation; Methodology; Writing—original draft; Writing—review and editing; Developed the research idea and study concept, designed the study and wrote the manuscript. Designed and conducted most experiments. Carried out in vivo experiments. Conducted most histological analyses. **María Agudo-Lera**: Data curation; Formal analysis; Investigation; Methodology; Designed and conducted most experiments. Performed phenotypical analysis from in vitro assays. **Olga Popova**: Resources; Investigation; Methodology; Project administration; Carried out in vivo experiments. Provided technical support. Performed viral load quantification in MVOA and hBLT in vivo experiments. **Ilya Tsukalov**: Validation; Methodology; Performed phenotypical analysis from in vitro assays. Performed viral load quantification in MVOA and hBLT in vivo experiments. **Marta Calvet-Mirabent**: Conceptualization; Data curation; Investigation; Methodology; Developed the research idea and study concept, designed the study and wrote the manuscript. Designed and conducted most experiments. Contributed to histological analyses. **Ignacio de los Santos**: Conceptualization; Investigation; Methodology; Provided samples of HIV negative and PWH donors and access to culture facilities for functional assays, as well as participated in data

discussion. Provided information of clinical parameters during the study. **Lucio García-Fraile**: Data curation; Formal analysis; Validation; Visualization; Provided samples of HIV negative and PWH donors and access to culture facilities for functional assays, as well as participated in data discussion. Provided information of clinical parameters during the study. **Patricia Fuentes**: Validation; Investigation; Methodology; Project administration; Provided NSG immunodeficient mice and contributed to in vivo MVOA experiments. **Cristina Delgado-Arévalo**: Data curation; Investigation; Methodology; Provided technical support. Performed phenotypical analysis from in vitro assays. **Juan Alcain**: Data curation; Investigation; Methodology; Provided NSG immunodeficient mice and contributed to in vivo MVOA experiments. **Nerea Sánchez-Gaona**: Formal analysis; Performed viral load quantification in MVOA and hBLT in vivo experiments. **Judith Grau-Expósito**: Formal analysis; Investigation; Performed the intact HIV DNA analyses. **María Lázaro-Díez**: Methodology; Generated and provided HIV-1 virus stocks for in vivo experiments. **Cecilia Muñoz-Calleja**: Funding acquisition; Investigation; Provided information of clinical parameters during the study. **Arantzazu Alfranca**: Conceptualization; Funding acquisition; Methodology; Developed the research idea and study concept, designed the study and wrote the manuscript. **Meritxell Genescà**: Conceptualization; Funding acquisition; Validation; Visualization; Methodology; Contributed to flow cytometry analysis, provided critical feedback and contributed to the preparation of the manuscript. **Julia G Prado**: Conceptualization; Validation; Investigation; Visualization; Methodology; Generated and provided HIV-1 virus stocks for in vivo experiments. **Vladimir Vrbanac**: Data curation; Investigation; Methodology; Provided humanized BLT mice. **Alejandro Balazs**: Data curation; Investigation; Methodology; Provided humanized BLT mice. **María José Buzón**: Conceptualization; Data curation; Validation; Investigation; Visualization; Methodology; Developed the research idea and study concept, designed the study and wrote the manuscript. Performed the intact HIV DNA analyses. **María L Toribio**: Conceptualization; Data curation; Validation; Investigation; Methodology; Provided NSG immunodeficient mice and contributed to in vivo MVOA experiments. **María A Muñoz-Fernández**: Data curation; Validation; Investigation; Methodology; Provided samples of HIV negative and PWH donors and access to culture facilities for functional assays, as well as participated in data discussion. **Francisco Sánchez-Madrid**: Conceptualization; Data curation; Formal analysis; Supervision; Funding acquisition; Validation; Investigation; Methodology; Writing—original draft; Project administration; Writing—review and editing; Developed the research idea and study concept, designed the study and wrote the manuscript. Provided samples of HIV negative and PWH donors and access to culture facilities for functional assays, as well as participated in data discussion. **Enrique Martín-Gayo**: Conceptualization; Data curation; Formal analysis; Supervision; Funding acquisition; Validation; Investigation; Methodology; Writing—original draft; Project administration; Writing—review and editing; Developed the research idea and study concept, designed the study and wrote the manuscript. Supervised the study.

Source data underlying figure panels in this paper may have individual authorship assigned. Where available, figure panel/source data authorship is listed in the following database record: biostudies:S-SCDT-10_1038-S44321-025-00255-x.

## Disclosure and competing interests statement

The authors declare no competing interests.

# Expanded View Figures

**Figure EV1.  Phenotypical analysis of MDDC from healthy donors and PWH in presence of soluble or nanoparticle-encapsulated Poly I:C.**

(A) Representative flow cytometry gating strategy defining viable big size HLA-DR + , CD14lo/-, CD11c+ and CD1a+ MDDC generated from Mo in the presence of GM-CSF and IL-4. (B) Proportions of live viability dye negative MDDCs from $n = 8$ HD (left panel) and $n = 8$ PWH (right panel). (C) Representative flow cytometry dot plots showing expression of CD86 vs intracellular staining of IFNβ in MDDCs from PWH cultured in media and after soluble (Sol PIC) or Nanoparticle (Nano PIC) -Poly I:C treatments. FMO for IFNβ is shown (left). Analysis of proportions of MDDCs from HD ($n = 8$; left) and PWH ($n = 9$; right) expressing intracellular IFNβ after 6 h of stimulation with soluble (Sol) or nanoparticle encapsulated poly: IC (Nano-PIC). (D) Analysis of IL-2, IL-12, IL-6, Il-15, IFNβ, IFNγ and TNFα, concentrations (pg/ml) present in culture supernatants of MDDCs from HD (left) an PWH (right) at 16 h in the mentioned conditions. (E) Basal mean fluorescence intensity (MFI) of CD86 (left) and CD40 (right) on MDDCs from $n = 15$ or $n = 8$ HD and $n = 13$ or $n = 7$ PWH, respectively. (F–H): MFI levels of CD86 (F) on MDDCs from $n = 15$ HD (left panel) and $n = 17$ PWH (right panel) at 6 h (F) or CD40 after 6 (G) and 16 h (H) of culture from $n = 7$ and $n = 7$ HD or $n = 8$ and $n = 10$ PWH, respectively, in basal conditions and after Sol PIC or nano empty or loaded with PIC treatments. Data in (B–H) are presented in Box and Whiskers plots showing median values and maximum and minimum error bars. Statistical significance between experimental conditions in the same samples were calculated using a two-tailed Wilcoxon tests and Bonferroni correction for multiple comparisons. *$P < 0.05$; **$P < 0.01$; ***$P < 0.001$.

▶

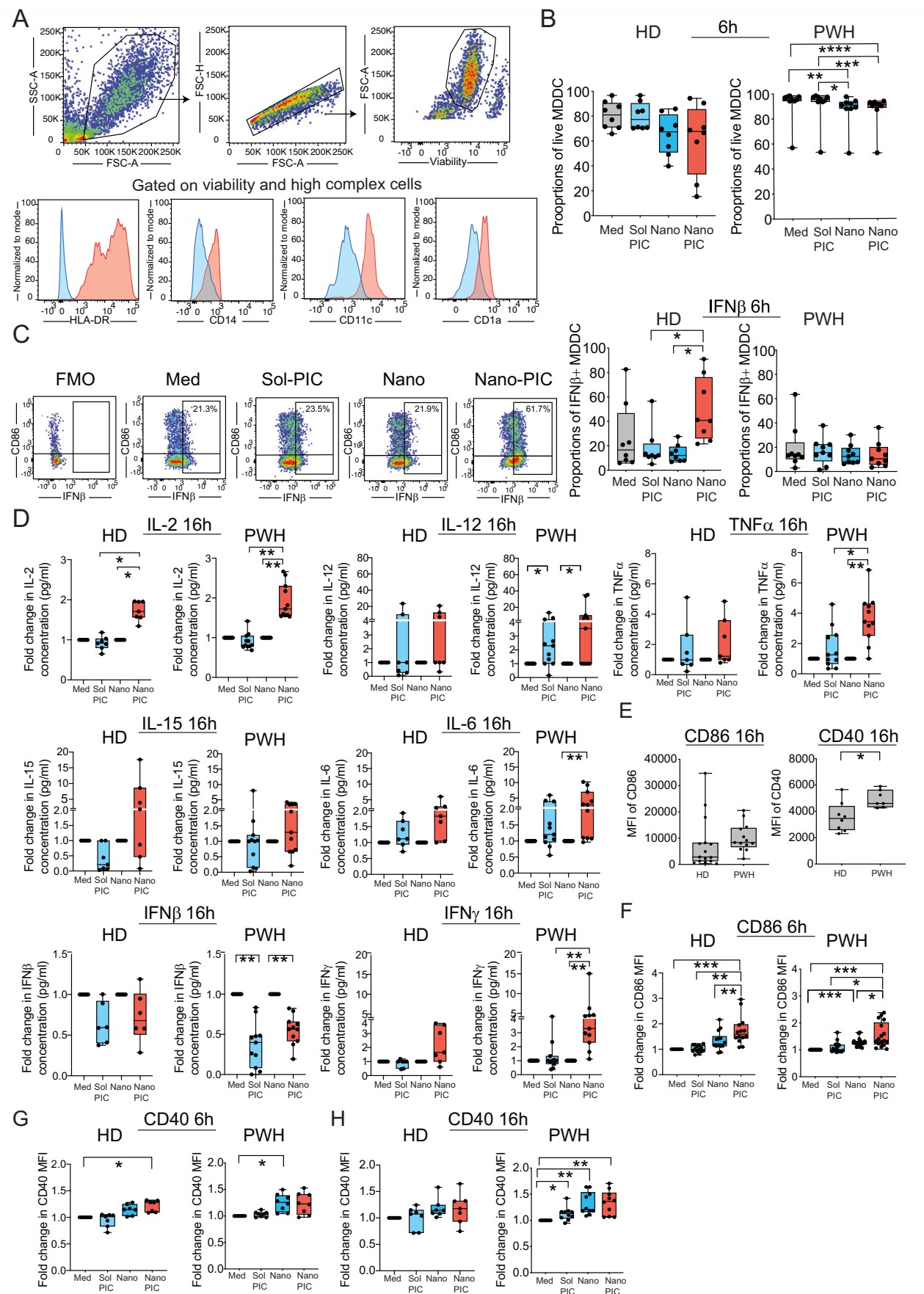

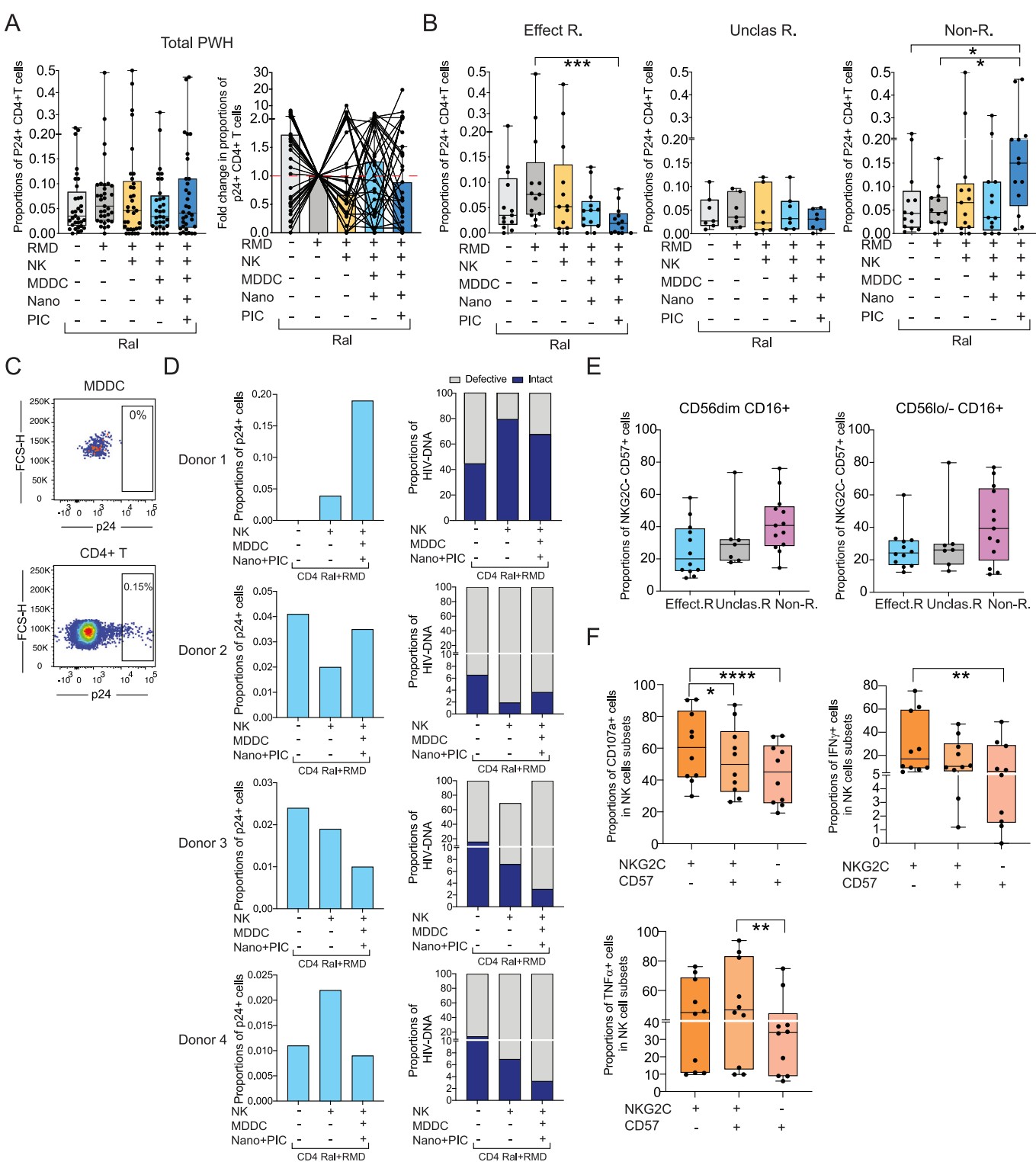

**Figure EV2.   Functional restoration of NK from PWH eliminating HIV-1-infected CD4+ T cells after treatment with Nano-PIC-MDDC.**

(A) Raw data and fold change in proportions of HIV-1 p24+ CD4+ T cells from all $n = 33$ PWH recruited for the study cultured with Romidepsin and Raltegavir in the absence or the presence of autologous NK cells alone or stimulated with Nano-empty or Nano-PIC-MDDC. (B) Proportions of HIV-1 p24+ CD4+ T cells in the presence of NK cells stimulated with Nano-PIC-MDDC from three separate effective responder (Effect. R; $n = 13$), unclassified (Unclas. R. $n = 7$) and non-responder (Non-R.; $n = 13$) PWH groups. (C) Representative flow cytometry dot plots showing intracellular expression of p24 in Nano-PIC-MDDC (upper plot) or co-cultured autologous CD4 + T cells (lower plot) from a tested PWH. (D) Proportions of HIV-1 p24+ cells analyzed by FACS (left plots, light blue) and IPDA analysis of intact (dark blue) or defective (gray) HIV-DNA in CD4+ T cells from $n = 4$ PWH cultured with Romidepsin and Raltegravir in the absence or the presence of autologous NK cells alone or stimulated with Nano-PIC-MDDC. (E) Proportions of NKG2C- CD57+ subset on CD56dim or CD56lo/- CD16 + NK from effective responder ($n = 13$, Effect. R; blue), unclassified ($n = 7$, Unclas. R; gray) and non-responder ($n = 13$, Non-R.; pink) PWH after activation with Nano-PIC-MDDC. (F) Proportions of total CD107a+ (left), total FNγ+ (right) and total TNFα+ (below) included in gated memory NK precursors NKG2C + CD57−, memory differentiated NKG2C + CD57+ and effector NKG2C− CD57+ subsets from $n = 10$ selected effective responder PWH after Nano-PIC-MDDCs in presence of PMA and Ionomycin stimulation for 4 h and Brefeldin A and Monensin. Data from (A, B, F) are presented in Box and Whiskers plots showing median values and maximum and minimum error bars. The data in (D) are represented with bar plots. Statistical Significance was calculated using a Friedman or a Kruskal–Wallis test for multiple comparisons. $*P < 0.05$; $**P < 0.01$; $***P < 0.001$.

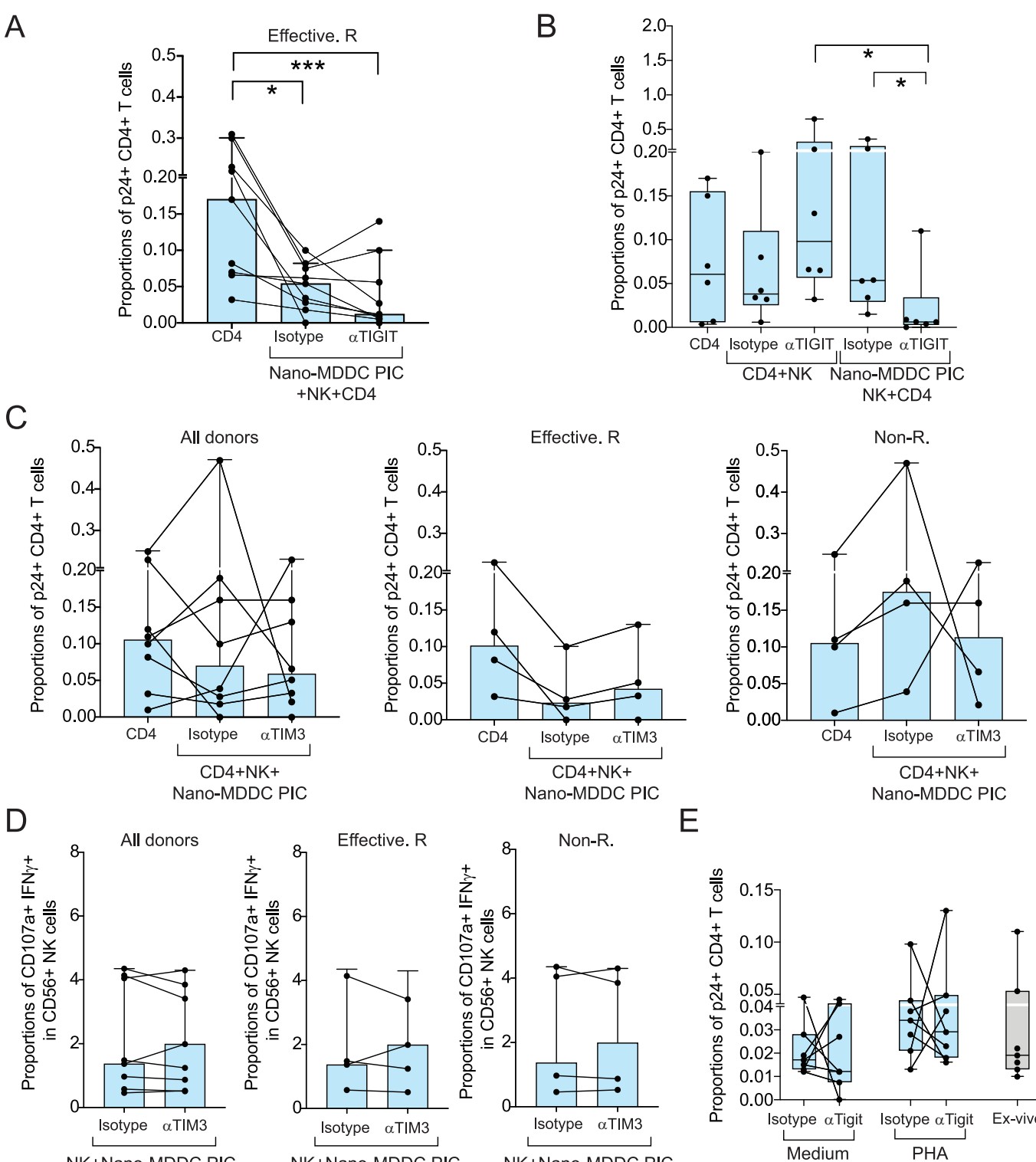

**Figure EV3. Proportions of p24+ CD4+ T cells after TIGIT and TIM3 blocking antibody.**

(A, B) Proportions of CD4+ T cells expressing HIV-1 p24+ cells cultured alone or with NK treated with Nano-PIC-MDDC from $n = 9$ effective responder PWH (A) or just media from $n = 6$ non-responder PWH (B) in the presence of isotypic control or anti-TIGIT mAbs. (C) Analysis of impact of blocking anti-TIM3 mAbs in experiments performed under the same conditions as (A) using $n = 8$ total PWH (left) or stratified based on effective ($n = 4$, middle) and non-responders ($n = 4$, right) donors. (D) Proportions of CD107a+ IFNγ + CD56 + NK cells in the co-culture experimental conditions described in (C) in the three responder PWH groups previously mentioned. (E) Proportions of p24+ cells in CD4+ T cells from $n = 7$ PWH treated with isotypic control or anti-TIGIT mAbs in the presence of media alone (basal) or under reactivation conditions with PHA (PHA). Baseline ex vivo levels of p24+ cells form these PWH are shown in the gray box and whisker bar. Data are presented in Box and Whiskers plots showing median values and maximum and minimum error bars. Statistically significant differences were calculated using a Friedman test for multiple comparisons and in the (B) was calculated using a one tailed Wilcoxon test and Bonferroni correction was applied. $*P < 0.05$; $***P < 0.001$.

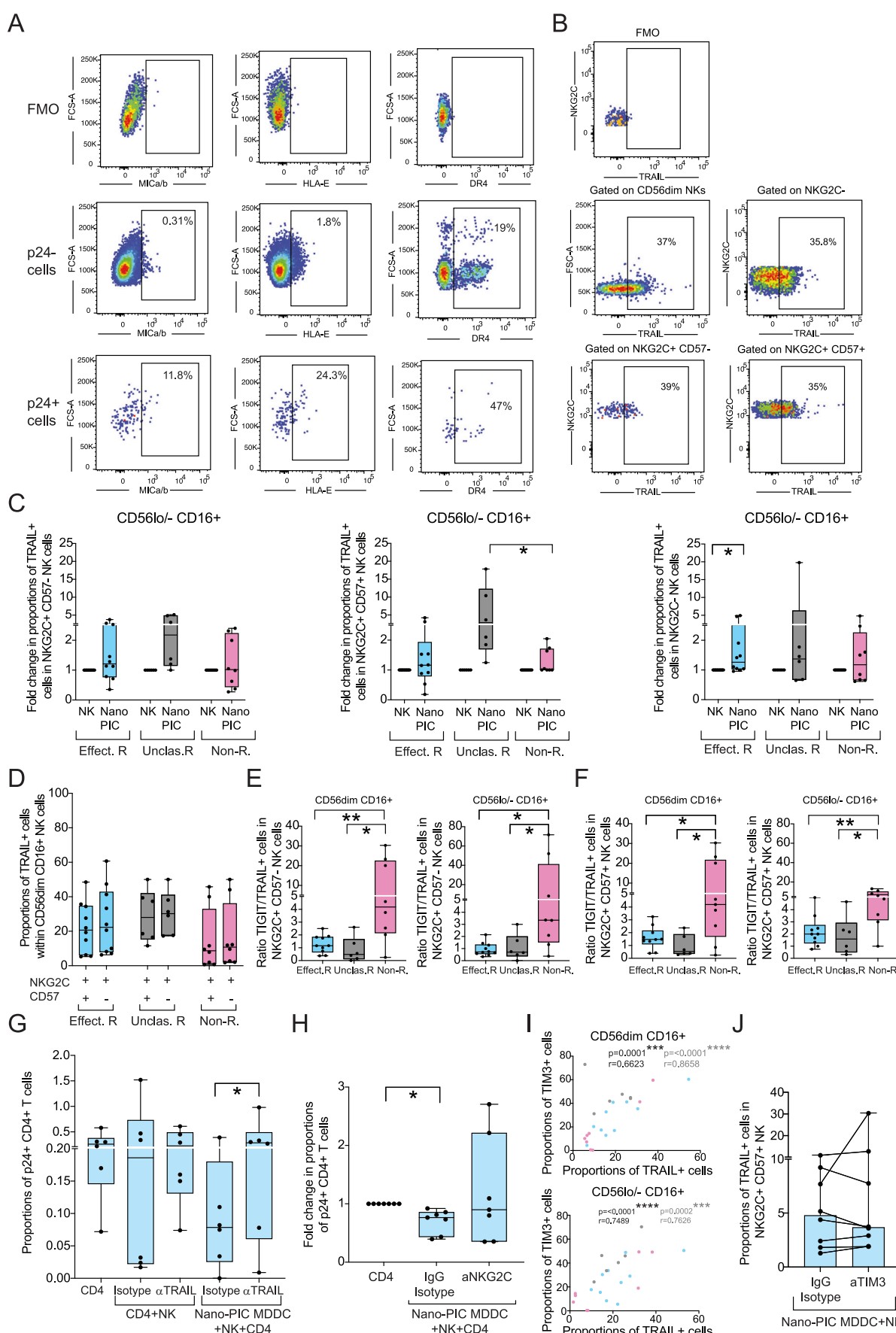

◄

**Figure EV4. Analysis of expression of NK receptor ligands on p24- vs p24+ CD4+ T cells and TRAIL on NK cell subsets from different PWH and its association with functionality.**

(A) Representative flow cytometry dot plots for the NK receptor ligands MIC/ab (NKG2D), HLA-E (NKG2C/A) and DR4 (TRAIL), in gated p24+ or p24- CD4 + T cells from PWH after 16h culture with PHA + IL-2. A FMO control is included to each NK ligand. (B) Representative Flow cytometry dot plot showing expression of TRAIL on total CD56dim CD16+ NK, total NKG2C-, NKG2C+ CD57- and NKG2C+ CD57+ NK from a representative effective responder PWH. (C) Fold change in proportions of TRAIL+ cells in NKG2C- (left) and adaptive NKG2C+ CD57- (middle) and NKG2C+ CD57+ (right) subsets after Nano-PIC-MDDC from effective responder (n = 10; Effect. R; blue), unclassified (n = 6; Unclas. R; gray) and non-responder (n = 8; Non-R.; pink) PWH included on CD56lo/- CD16 + NK. (D) Raw proportions of TRAIL+ cells within adaptive NKG2C+ CD57+ and NKG2C+ CD57− cell subsets in the same responders PWH groups defined in (C) included in CD56dim CD16+ NK subpopulation after stimulation with Nano-PIC-MDDC. Statistical significance was calculated using a Mann–Whitney test and Bonferroni correction. (E, F) Ratio of TIGIT+ versus TRAIL+ cells in precursor NKG2C+ CD57- adaptive (E) and mature NKG2C + CD57+ (F) cells in CD56dim (left) and CD56lo/− (right) CD16 + NK subsets from the same PWH responder groups. Statistical significance was calculated using a one tale Mann–Whitney test and Bonferroni correction. (G, H) Proportions of HIV-p24 + CD4 + T cells from n = 6 (G) and n = 7 (H) PWH previously identified as effective responders treated with Raltegravir and Romidepsin and cultured in the absence or the presence of autologous unstimulated NK cells or NK treated with Nano-PIC-MDDCs in the presence of either IgG Isotypic control or with anti-TRAIL blocking mAb (G) or in the presence of anti-NKG2C mAb (H). Statistical significance was calculated using a one tale Wilcoxon matched pairs test and Bonferroni correction or a Friedman test for multiple comparisons. (I) Spearman correlations between proportions of TIM3 and TRAIL after Nano-PIC-MDDC within CD56dim (upper) and CD56lo/− (bottom) CD16 + NK. Statistical P and R values considering all data (black), without unclassified group (gray). (J) Analysis of proportions of TRAIL+ cells within adaptive NKG2C + CD57 + NK from n = 7 PWH stimulated with Nano-PIC-MDDC in the presence of isotype or anti-TIM3 mAb. Statistical significance was calculated were calculated using a two-tailed Wilcoxon matched pairs. Data from (C–H, J) are presented in Box and Whiskers plots showing median values and maximum and minimum error bars. *P < 0.05; **P < 0.01; ***P < 0.001; ****P < 0.0001.

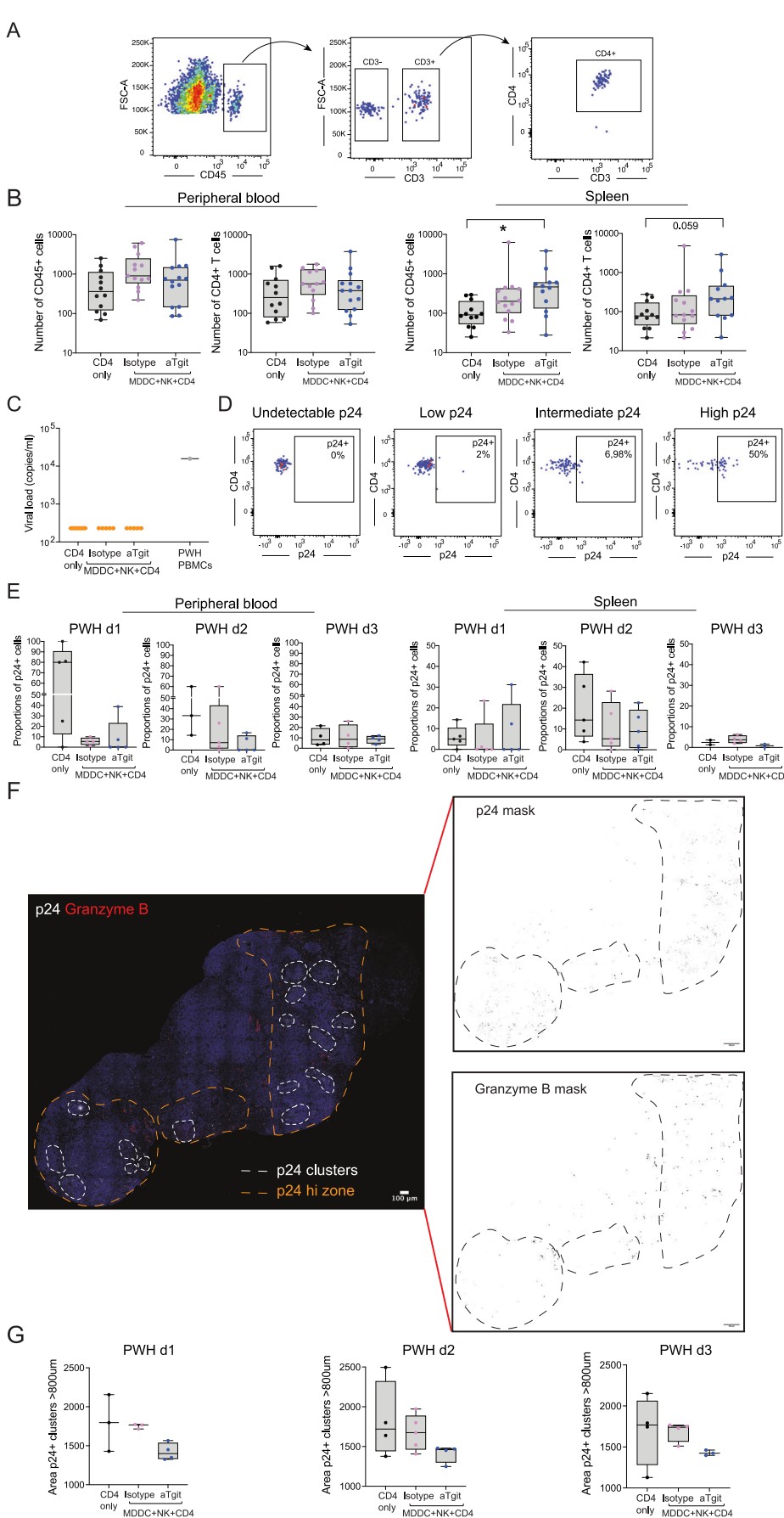

◀

**Figure EV5. Analysis of impact of anti-TIGIT antibodies on HIV-1 reactivation and in reconstitution and histological patterns in humanized NSG mice transplanted with NK cells.**

(A) Representative flow cytometry gating strategy showing identification of human CD45+ cells from PWH transplanted into immunodeficient NSG mice, and the identification of CD4+ CD3+ T cells. (B) Total absolute numbers of human CD45+ (left) and CD4+ T cells (right) in the peripheral blood and spleen from NSG mice transplanted either with CD4 + T cells from PWH alone (black) or in combination with autologous NK and Nano-PIC-MDDC and injected with either Isotypic (violet) or anti-TIGIT (blue). Data from $n = 3$ independent experiments using different PWH donors is shown. (C) Analysis of HIV-1 plasma viral load on plasma of transplanted NSG mice receiving either only CD4 + T cells from PWH or Nano-PIC-MDDC-NK immunotherapy and either Isotypic or anti-TIGIT mAbs. A positive control from cells from PWH is also shown. (D) Representative examples of levels of intracellular HIV-1 p24 on gated circulating human CD4 + CD3+ transplanted into mice and showing undetectable, low, intermediate and high p24 detection. (E) Proportions of HIV-1 p24+ cells within CD4 + T cells in the peripheral blood (left plot) and spleen (right plot) of NSG mice transplanted with CD4 + T cells from PWH alone (black) or in combination with autologous NK cells and Nano-PIC-MDDCs and injected with either Isotypic (violet) or anti-TIGIT (blue) mAbs for each donors 1, 2 and 3 (d1, d2, d3) used on each independent experiment. (F) Representative confocal microscopy images showing immunofluorescence histological analysis of a spleen tissue section from a representative transplanted NSG mouse showing staining of HIV-1 p24 (white) and granzyme B (red) showing areas with areas of high p24 cluster concentration (orange) and individual p24 clusters (white). Quantification of individual cluster cells with p24 (upper right) and granzyme B (lower right) staining masks are shown. Scale bar: 100 μm. (G) Analysis of size of clusters of p24+ cells present in the spleen of NSG mice transplanted with the three different d1, d2, d3 PWH as previously described in (E). In panel (B, E, G) data are presented in Box and Whiskers plots showing median values and maximum and minimum error bars. Statistical significant differences between the different groups of treatment were calculated using a two tail Mann–Whitney test and Bonferroni correction for multiple comparisons. *$P < 0.05$.

