## [Peer Review File · EMBO Molecular Medicine]

COMBINED DENDRITIC CELL AND ANTI-TIGIT IMMUNOTHERAPY POTENTIATES ADAPTIVE NK CELLS AGAINST HIV-1

Ildelfonso Sánchez-Cerrillo, María Agudo-Lera, Olga Popova, Ilya Tsukalov, Marta Calvet-Mirabent, Ignacio de los Santos, Lucio García-Fraile, Patricia Fuentes, Cristina Delgado-Arévalo, Juan Alcain, Nerea Sanchez-Gaona, Judith Grau-Expósito, María Lázaro-Díez, Cecilia Muñoz-Calleja, Arantzazu Alfranca, Meritxell Genescà, Julia García-Prado, Vladimir Vrbanac, Alejandro Balazs, María J Buzon, María Toribio, María Ángeles Muñoz-Fernández, Francisco Sánchez-Madrid, and Enrique Martin-Gayo

Corresponding author: Enrique Martin-Gayo (emgayo@cbm.csic.es)

Review Timeline:

Submission Date:	27th Nov 24
Editorial Decision:	10th Jan 25
Revision Received:	9th Apr 25
Editorial Decision:	6th May 25
Revision Received:	14th May 25
Accepted:	16th May 25

Editor: Zeljko Durdevic

Transaction Report:

10th Jan 2025

Dear Dr. Martin-Gayo,

Thank you for the submission of your manuscript to EMBO Molecular Medicine, and please accept my apologies for the unusual delay in getting back to you. We have now received feedback from the three reviewers who agreed to evaluate your manuscript. All three referees recognize interest of the study but also raise important concerns that should be addressed in a major revision. If you would like to discuss further the points raised by the referees, I am available to do so via email or video. Let me know if you are interested in this option.

We would welcome the submission of a revised version within three months for further consideration. Please let us know if you require longer to complete the revision.

I look forward to receiving your revised manuscript.

Yours sincerely,

Zeljko Durdevic

We require:

- 1) A .docx formatted version of the manuscript text (including legends for main figures, EV figures and tables). Please make sure that the changes are highlighted to be clearly visible.
- 2) Individual production quality figure files as .eps, .tif, .jpg (one file per figure). For guidance, download the 'Figure Guide PDF': (<https://www.embopress.org/page/journal/17574684/authorguide#figureformat>).
- 3) A .docx formatted letter INCLUDING the reviewers' reports and your detailed point-by-point responses to their comments. As part of the EMBO Press transparent editorial process, the point-by-point response is part of the Review Process File (RPF), which will be published alongside your paper.
- 4) A complete author checklist, which you can download from our author guidelines (<https://www.embopress.org/page/journal/17574684/authorguide#submissionofrevisions>). Please insert information in the checklist that is also reflected in the manuscript. The completed author checklist will also be part of the RPF.
- 5) Please note that all corresponding authors are required to supply an ORCID ID for their name upon submission of a revised manuscript.
- 6) It is mandatory to include a 'Data Availability' section after the Materials and Methods. Before submitting your revision, primary

datasets produced in this study need to be deposited in an appropriate public database, and the accession numbers and database listed under 'Data Availability'. Please remember to provide a reviewer password if the datasets are not yet public (see <https://www.embopress.org/page/journal/17574684/authorguide#dataavailability>).

12) Author contributions: You will be asked to provide CRediT (Contributor Role Taxonomy) terms in the submission system. These replace a narrative author contribution section in the manuscript.

13) A Conflict of Interest statement should be provided in the main text.

14) Every published paper now includes a 'Synopsis' to further enhance discoverability. Synopses are displayed on the journal webpage and are freely accessible to all readers. They include a short stand first (maximum of 300 characters, including space) as well as 2-5 one-sentences bullet points that summarizes the paper. Please write the bullet points to summarize the key NEW findings. They should be designed to be complementary to the abstract - i.e. not repeat the same text. We encourage inclusion

of key acronyms and quantitative information (maximum of 30 words / bullet point). Please use the passive voice. Please attach these in a separate file or send them by email, we will incorporate them accordingly.

15) Include a Reagents and Tools Table as part of the Methods section, which can be downloaded from our author guidelines (<https://www.embopress.org/page/journal/17574684/authorguide#structuredmethods>)

**** Reviewer's comments ****

Referee #1 (Comments on Novelty/Model System for Author):

The questions and approaches presented are undoubtedly innovative, offering valuable perspectives from these preclinical models for potential clinical applications. However, conducting additional experiments-particularly to compare effective versus dysfunctional donors or to demonstrate the effectiveness of the therapeutic approach on the viral reservoir-would strengthen the conclusions and enhance the robustness of the data.

Referee #1 (Remarks for Author):

ART effectively suppresses HIV replication but fails to eliminate persistent viral reservoirs in latently infected CD4+ T cells, leading to chronic immune activation and exhaustion of antiviral immune cells like CD8+ T cells and NK cells. Enhancing NK cell activity, particularly memory-like NKG2C+ NK cells, shows promise as they exhibit strong cytotoxicity and are linked to functional HIV remission. DC can enhance NK cell activity through Ag presentation and cytokine signaling. In this study authors aim demonstrating that nanoparticles loaded with Poly I:C (Nano-PIC) enhance DC-mediated activation and cytotoxicity of NK cells against HIV-1-infected cells, especially by expanding memory NKG2C+ TRAIL+ NK cells. Combining TIGIT blockade with Nano-PIC DC immunotherapy further restores NK cell functionality and reduces HIV-1 reservoirs, providing a potential strategy for improved immunotherapy against HIV-1.

Introduction

- Line 80: Consider citing Levy Y et al., Eur J Immun 2014 for "stimulation with activated dendritic cells (DCs) loaded with HIV-1 peptides (19, 26-29)."
- Line 124: Rephrase the sentence for clarity: "Finally, TIGIT blockade restores the cytotoxic activity of dysfunctional NK cells against p24+ cells in vitro following Nano-PIC DC stimulation."

Methods

- Overall, the experiments require more detailed descriptions, including information on doses, time points, p24+ cell isolation methods, and co-culture ratios, or references to previous publications for these protocols.
- Line 159: Provide additional details about the "complexes of polymeric nanoparticles (TRANS-ITX2-Mirus Bionova)" or cite a relevant publication for clarity.
- Trans-well experiments are not described and should be included for completeness.
- Some abbreviations used in the manuscript need clarification, such as "DCs generated in vitro from circulating adherent Mo" (Line 156), ULBP1, PB, etc.
- Certain statistical tests, such as Grubbs' test, are mentioned but not adequately explained.
- The description of box-and-whisker plots (e.g., mean vs. SD) should be clearly specified.

Results

i. in vitro model of DC activation

Authors demonstrate first that DC stimulated with Nano-PIC enhance the activation and cytotoxic function of NK cells, particularly in people with HIV (PWH) on ART. Nano-PIC DCs induce higher expression of maturation markers (CD86), cytokines (IL-12), and ligands for activating NK receptors compared to soluble Poly I:C (Sol-PIC). This activation leads to increased degranulation (CD107a) and IFN γ production in NK cells, improving their ability to eliminate HIV-infected CD4+ T cells.

1. The activation of DCs from HIV+ donors did not yield significant results, which is an important expected outcome, as the authors propose using this system for therapeutic approaches. It would be advisable to first validate MoDC differentiation with a more comprehensive profile (e.g., Sup Fig1A). Maturation profiles are often more clearly represented using MFI rather than the percentage of positive cells.
2. The presentation of data in Figure 1 versus Supplemental Figure 1 is confusing. Data from HD and HIV+ donors should be grouped for better readability.

3. In Figure 1A, it is recommended to show the "Med" condition instead of FMO, while the complete gating strategy with FMO should be moved to Supplemental Materials.

4. Authors tend to overinterpret certain results, which should be adjusted for clarity:

- Line 320: "Treatment of DC with Nano-PIC induced a mild but significant reduction of cell viability" - This is not observed for PWH.

- Line 324: "Nano-PIC more significantly and consistently increased intracellular expression of IL-12 in DC from both HD and PWH" - This is not entirely accurate for PWH; while a trend is visible, it is not significant. The authors should include more donors to achieve statistical significance or perform dose-dependent testing to better demonstrate Nano-PIC DC functionality from PWH, especially in the context of therapeutic applications.

- Line 328: "Upregulated at 6 and 16h in DC primed with Nano-PIC from HD and PWH (Supplemental Figure 1B-1C, right, Figure 1C, Supplemental Figure 2A-2B)" - i) There is no significant response in DC from PWH. ii) Clarify in the methods section why Figure 1B shows 16h of stimulation for HD, while Sup Fig1 shows 6h. iii) In Sup Fig2, define "sample1," ensure gates are consistent across all dot plots, and label the percentages. Also, include non-stimulated controls.

5. MoDCs from HIV- and HIV+ donors appear to differ at baseline (e.g., CD86 expression). The authors should provide a more detailed phenotyping of MoDCs, including additional maturation markers (e.g., CD40, as described by Marta Calvet-Mirabent in EBioMedicine 2022), and discuss the reduced responsiveness of DCs from PWH to Nano-PIC.

6. Line 331: "MICa/b and ULBP1 as well as HLA-E, which is the ligand for both the activating and inhibitory receptors NKG2C or NKG2A, respectively (Supplemental Figure 2A; Figure 1D)." Supplemental Figure 2A refers to PWH, while Figure 1D refers to HD. Please clarify. Additionally, HLA-E in Figure 1D is borderline significant and should be noted accordingly.

7. Line 333: "The increase of these ligands in DCs from PWH stimulated with Nano-PIC was confirmed." - No data is presented comparing Sol-PIC versus Nano-PIC in DCs from PWH.

8. Statistical analysis for Figures 1 and 2, as well as the Supplemental Figures, relies on the Wilcoxon paired t-test, which is not appropriate for multiple group comparisons. ANOVA or Friedman tests should be used instead.

9. Line 336: "Nano-PIC DCs may display increased abilities to activate NK cells." - Specify this is based solely on CD86 expression, as the percentage of CD86+ cells in PWH remains unchanged.

ii. NK function with DC Nano-Pic

Effective responders among PWH showed elevated memory NKG2C+ NK cells with increased TRAIL expression, a key mediator of cytotoxicity, while dysfunctional responders exhibited immune exhaustion linked to TIGIT overexpression. Blocking TIGIT restored NK cytotoxicity in dysfunctional responders, highlighting potential therapeutic avenues using Nano-PIC DCs to bolster immune responses and target HIV reservoirs.

1. Provide an experimental scheme in Supplemental Figure for clarity.

2. Line 339: "Expression of IFN γ and CD107a on CD56dim CD16+ cells was analyzed (Supplemental Figure 3A)." The representative dot plots are not convincing, as the number of dots appears too low in the double-positive populations. Include comparative dot plots with associated groups in a single plot, and provide the exact number of events for accuracy.

3. Line 340: "The expression of the degranulation marker CD107a (Figure 1E, left panel) and proportions of CD107a+ IFN γ + (Figure 1E, right panel) cells within the CD56dim CD16+ NK subset were observed in response to Nano-PIC DCs." Clarify whether this is in comparison to NK cells alone. Additionally, differences between DC Nano-PIC and DC Nano groups are not adequately described in Figure 1E. Ensure it is specified that these data are from HD.

4. Line 342: "The induction of these degranulation parameters was confirmed in DCs from PWH exposed to Nano-PIC (Supplemental Figure 3B)." No significant results are presented. This statement should be adjusted accordingly.

5. Line 344: "This effect was dependent on cell-to-cell contact of NK cells with Nano-PIC DCs in a transwell assay compared to DCs treated with empty nanoparticles (Figure 1F)." The transwell experiment is not described, and no data for DCs from PWH are included.

6. Line 349: "NK cells primed with Nano-PIC DCs were able to kill significantly higher proportions of K562-GFP cells, compared to untreated NK or cells." Specify donor groups and ensure the experiment includes comparisons for both HD and PWH.

7. For the ADCC test on gp120/CHO cells, include a control without neutralizing antibodies (NeutAb). The choice of the cocktail used should be justified within this model. Discuss the rationale for using CHO-HXB2 as the model, as opposed to infection models like JRCSF, which are presented in subsequent figures.

8. In Figure 2B, the baseline is not shown, despite being mentioned in the legend. Ensure that the baseline is included for clarity.

9. The model presented in Figure 2A measures p24+ cells. It would be important to demonstrate that the DC Nano-PIC system impacts cells producing infectious particles (reservoir cells) rather than defective virus.

10. The legend for Supplemental Figure 4 mentions the use of DC Nano empty, which appears to be the most appropriate control. However, this control is not presented in Figure 2 and is missing.

11. The use of the CD57 marker in Figure 2C should be clarified, including its relevance and interpretation in this context.

12. For Figure 2, the Wilcoxon test is not suitable, as the three groups are independent. An appropriate statistical test, such as ANOVA or Kruskal-Wallis, should be used instead.

13. In Figure 2K and Supplemental Figure 4D, clarify whether the 8 PWH are grouped as effectors. Did the authors observe this functional difference in KKG2C CD57- cells only within the effector group? This distinction should be clearly described.

iii. Immunomodulation through TIGIT expression

1. It is regrettable that the baseline for HDs is not included, especially as the study aims to explore immune exhaustion in PWH. FACS histograms in Supplemental Figure 5A should display baseline levels from representative donors in all three groups to provide a clearer comparison.
2. The rationale behind performing correlation analyses on "all groups" (black) versus "all without dysfunctional" (red) is unclear, especially when subgroup correlations are clearly established in Figure 4D. The coding, color scheme, and legend for these analyses need to be clarified to enhance understanding.
3. Line 429: "In contrast, induction of TIM3 expression in NK was significantly correlated with lower proportions of infected p24+ CD4+ T cells in our functional assays (Supplemental Figure 5E) and higher frequencies of NKG2C+ CD57- memory precursor NK cells and lower percentages of NKG2C- CD57+ cells within the CD56dim (left) and CD56lo/- (right) CD16+ subpopulations (Supplemental Figure 5F-5G)." The authors overinterpret their results here. This statement should be rephrased to better reflect the data's actual significance and limitations.
4. Line 452: "Thus, our data demonstrate that blockade of TIGIT can improve functional restoration of NK cells from dysfunctional responder PWH in combination with Nano-PIC DCs." To strengthen this conclusion, it would have been more compelling to directly compare the effects of these treatments across responder and non-responder groups. This approach would validate the dichotomy between these groups and reinforce the study's findings.

iv. Mediation by TRAIL

1. Supplemental Figure 6A: The percentage of p24+ cells is low, making it essential to include the number of events acquired in the methods section to validate the percentages shown in Figure 4 and Supplemental Figure 6. This addition would enhance the credibility and reproducibility of the data.
2. In general, it would be helpful to include a reference line at Y=1 in fold-change graphs to clearly distinguish between groups showing an increase versus a decrease. This visual aid would make the data interpretation more straightforward.
3. Line 470: "None of the NK cell subsets from dysfunctional...Supplemental Figure 6B_C" is inconsistent with the data shown in Supplemental Figure 6C (right). The statement should be revised to accurately reflect the results presented in the figure.
4. Line 478: "Therefore, our results indicate a generalized defect in upregulation of TRAIL in both NKG2C+ and NKG2C- NK from dysfunctional responder PWH that is linked to TIGIT upregulation." The term "linked" does not seem appropriate here, as it implies a causative relationship. A more neutral term, such as "associated with," would be more accurate and better supported by the data.

v. In vivo models

In a humanized BLT mouse model of HIV-1 infection, TIGIT expression increases on NKG2C+ memory NK precursors during their maturation, correlating with viral loads. Combined Nano-PIC DC and NK cell immunotherapy with TIGIT blockade effectively reduces HIV-1-infected CD4+ T cells in both peripheral blood and lymphoid tissues of NSG mice, as evidenced by reduced p24+ cell proportions and cluster sizes. This treatment enhances cytotoxic NK cell recruitment and granzyme B expression in high viral replication zones, with a concurrent preservation of TRAIL-expressing NKG2C+ memory NK precursors in lymphoid tissues.

1. Some figures are not correctly mentioned (e.g., Supplemental Figure 8): Ensure all references to figures throughout the text are accurate and consistent with their labeling. Correctly referencing figures will improve clarity and avoid confusion.
2. Line 553: The phrase "we detected a considerable increase of up to a median of 23.35% of p24+ cells in human circulating CD4+ T cells from the transplanted NSG mice that did not receive any immunotherapy" should be rephrased for precision. Replace "considerable" with a more objective term such as "notable" or "measurable," or simply describe the increase quantitatively without qualitative language.
3. Line 557: The statement "treatment of mice with Nano-PIC-DCs and NK cells reduced proportions of p24+ cells within circulating CD4+ T cells, suggesting that immunotherapy by itself was able to reduce expansion of infected cells in PB" is misleading since the difference is not statistically significant compared to the isotype control. Clarify this limitation in the text.
4. Supplemental Figure 8B: Clearly specify how "events of cells" were calculated. Providing methodological details will ensure the data's transparency and reproducibility.
5. Supplemental Figure 8D: Define what "example 1 to 3" represent. Explicitly stating their meaning will avoid ambiguity for readers.
6. Figure 6C: Draw and label the clusters on the figure to enhance interpretability and provide a clear visual representation of the data.
7. Supplemental Figure 9: The number of events appears insufficient to determine the percentages of these populations reliably. Include the total number of events analyzed in the methods section or the figure legend to validate these findings.
8. Line 595: The phrase "We observed a significant enrichment of NKG2C+ CD57- precursors within the total pool of memory NK cells in the spleen after anti-TIGIT mAb treatment..." would benefit from including pre-treatment (pre-mOVA) data in Figure 7A for comparison. This would provide a more comprehensive view of the treatment's impact.
9. Supplemental Table 3: Clarify whether the Effective versus Dysfunctional status of the three donors is known. If not, conducting experiments with defined donors (Effective vs. Dysfunctional) would strengthen the conclusions drawn and make the data more robust.

Discussion

This study explores immunotherapy strategies for HIV-1 remission, focusing on enhancing NK cell activity through DC immunotherapy and checkpoint blockade. Nano-PIC DCs improved NK cell cytotoxicity, particularly by promoting NKG2C+ memory NK cells, which showed increased functionality against HIV-1 infected CD4+ T cells. The research also identifies TIGIT as a key inhibitory receptor associated with NK cell dysfunction in HIV-1 infection, and its blockade improved NK responses. Additionally, TRAIL was found to play a role in NK cell-mediated cytotoxicity. These findings suggest that modulating NKG2C+ NK cells and targeting TIGIT could improve NK-based immunotherapies for HIV-1.

Suggested rephrased limitations to mention in the Ms:

1. Specificity of the Therapy: The proportion of p24+ cells is relatively low in the in vitro system (~0.1%). The authors could discuss the potential effects of the therapy on uninfected cell populations, as this might provide insights into its broader impact and specificity.
2. Potential infection of Nano-PIC activated DCs: A potential limitation of the therapy is whether dendritic cells activated by Nano-PIC might themselves become infected and contribute to viral spread. This risk should be acknowledged and discussed as a potential caveat of the approach.
3. Cytokines important for NK Activity: The study mentions that no significant levels of IL-15, a cytokine crucial for NK cell activity, were detected in treated DC. These results should be explicitly presented in the manuscript. Additionally, the authors could discuss the potential use of humanized mouse models with Flt3-L, which allow better reconstitution of both NK cells and DCs, to overcome this limitation and improve NK-DC interactions.

These points would add depth to the discussion, acknowledging the study's limitations and offering potential avenues for further research.

Referee #2 (Remarks for Author):

The manuscript by Sánchez-Cerrillo I et.al. explored the ways to enhance the cytotoxicity of NK cells to eliminate HIV-1-infected cells. They reported that combination of Nano-PIC stimulated DCs and anti-TIGIT antibodies might be a promising strategy to increase the efficacy of immunotherapies for HIV. While this is an important study, several concerns should be addressed before it can be accepted for publication.

Concerns:

1. The title "NANOPARTICLE-ENHANCED DENDRITIC CELL" is some kind misleading. It will be better to use "Nano-PIC" or other words?
2. The DCs used in the study were generated in vitro from circulating adherent monocytes, but not primary DCs isolated from blood. This should be highlighted in the abstract and results part clearly.
3. It's known that NK cells interact with the HLA-E molecule via both the inhibitory receptor NKG2A and the activating receptor NKG2C. Did the authors check the expression of NKG2A on NK cells in the experiment systems?
4. The manuscript needs extensive proofreading to make sure all the statements are correct, and all the labels are clear. For example, it's hard to guess whether Figure 1B was generated from HD or PWH. The other example: in the sentence "In fact, higher ratios of cells expressing TIGIT ... and NKG2C+ CD57+ present in this group (Supplemental Figure E-F)", it's not clear which figure the authors was talking about.

Referee #3 (Comments on Novelty/Model System for Author):

The question is trendy and interesting (NK immunotherapy/impact on HIV-1 reservoir/replication).
The experiments are carried out with care in relevant and difficult models.
The ease of using such a method in patients is questionable

Referee #3 (Remarks for Author):

HIV cure remains a challenge despite decades of intensive research. Sánchez-Cerrillo et al. Present a nice study ranging from

in vitro evidence to a preclinical model for the proof-of-concept of a novel immunotherapy aiming to restore NK-cell function and activate their cytotoxic activity towards HIV-1 infected cells. Such a strategy could be a game changer in the field. Experiments are carried out with an exceptional level of effort (autologous primary cell co-culture) and result in high-quality data. However, the interpretation is sometimes confusing, mostly due to some unnecessary speculation.

Major comments:

- Figure 1 shows that DC from HD and PLWH react differently to stimulation. This intriguing finding warrants further exploration, as it could significantly impact the efficacy of immunotherapy in the target group. As the lack of IFN β response is striking, a Luminex may be performed to determine what cytokine profile of HD and PLWH stimulated DC. Despite this very interesting observation, the author focuses all subsequent analysis on NK cells to explain the responder/non-responder status. The DC capacity to respond to the stimulus may deserve attention.
- The authors stated very clearly that the immune environment of PWH is complex and could lead to dysfunctionality in the immune response. It would be great to add data on the phenotype of NK from dysfunctional or responsive PWH. For example, the authors show that TIGIT expression level by NK is inversely correlated with the response to immunotherapy. Is it already the case in circulating NK from the so-called dysfunctional PWH?
- How PLWH were classified into responders/dysfunctional/unclassified responders in figure 2 is questioning. Some responders have complete suppression with NK only, which then impedes the capacity to consider them as responders or not, the other way around, some unclassified responders show a decrease in the proportion of p24 cells. This creates confusion. Objective thresholds must be used to define the groups.
- In this vein, why use "dysfunctional"? It sounds subjective. Why not use "non-responders"?
- The validity of a ROC curve performed on selected, but not all individuals should be confirmed by an expert on biostatistics.
- Also, data support that DC therapy and TIGIT blockade have an impact on NK cell functions, but some controls are missing to determine the extent of their synergy. (i) The authors should demonstrate that the enhancement of cytotoxic activity due to TIGIT blockade is specific to Nano PIC DC-exposed NK cells. (ii) The same question applies to TRAIL blockade. (iii) Showing that TIGIT blockade increases TRAIL would also add to their hypothesis. Finally (iii) What is the impact of NK alone on HIV-1 infection in the humanized mice model? This control is missing to confirm the activity of the DC immunotherapy in this model.
- The authors tend to overinterpret the significance of the result in many instances:
 - o P15: "As shown in Figure 1B and Supplemental Figure 1D, Nano-PIC more significantly and consistently increased intracellular expression of IL-12 in DC from both HD and PWH at 6 h compared to cells treated with Sol-PIC" Supplemental Fig1D show no statistical difference between conditions for PWH
 - o P16 : "The induction of these degranulation parameters was confirmed in DC from PWH exposed to Nano-PIC (Supplemental Figure 3B)". no statistical difference is shown
 - o P19: "Together, these findings suggest that effective response of NK cells from PWH to Nano-PIC DC immunotherapy is associated with the induction of precursor and mature memory NKG2C+ NK cells with different functional profiles, which may have improved capabilities to target HIV-infected CD4+ T cells." Authors should be careful, even though differentiation is most probable, evidence only show that immunotherapy induces NKG2C expression on NK cells. Furthermore, it is known that the different NK subsets present distinct functionality.
 - o P21 : "In contrast, none of the NK cell subsets from dysfunctional and unclassified responders were able to further increase expression of this molecule (Figure 4B; Supplemental Figure 6 B-C)" sentence is true but statistical significance is not met for any of the groups in NKG2C+ CD57- NK population
 - o P21: "Of note, resulting raw levels of TRAIL were significantly higher in mature NKG2C+ CD57+ from effective responders compared to the low amount of these cells present in dysfunctional responders in the most mature CD56lo/- CD16+ NKbut not the CD56dim CD16+ NK cells subset (Figure 4C; Supplemental Figure 6D)" none of the figures show statistically higher levels of TRAIL in Effective PWH
 - o FigS9 : authors should be very careful analyzing under 100 events to state finding.
 - o P23 In addition, reduced ratios of TIM3+/TIGIT+ cells in these were observed from 1 to 2 weeks p.i. suggesting a loss in TIM3 expression and progressive enrichment of TIGIT in memory NK cell precursors from HIV-1- infected mice (Figure 5D). and should be replaced by and/or because authors shown none of it.
 - o P25 : "Consistently, numbers of hCD4+ T cells were also significantly higher in the spleen in animals treated with Nano-PIC DCs and NK cells in combination with anti-TIGIT mAbs (Supplemental Figure 7B, right plot)" authors show only a significant increase for hCD45 count under TIGIT inhibition but no significant effect on CD4+ T cell count.
 - o P25 : "However, a higher proportion of animals with undetectable p24 expression within CD4+ T cells from PB was present in the anti-TIGIT group and differences in this group were more significant compared to the control mice transplanted only with CD4+ T cells (Figure 6B, left panel)." Authors do not show animals with undetectable infected circulating CD4+ T-cells but the frequencies of this population for each animals. The conclusion is an overstatement plus there is no difference between isotype and anti-TIGIT group. Reader has to wait until the discussion to see the authors discuss that which is clearly misleading.

- Authors should discuss their choice of Mo-DC differentiation protocol, as many others exist and result in different functionalities of DC. This is even more important, since previous work carried out by the authors was done with cDC1, cDC2 and monocytes.

- In the cytotoxic assay with A562-GFP cells and ADCC assay, the authors did not mention whether NK cells are purified or if coculture of NK-DC is used. This is very crucial, since educated DC secrete IFN- γ , which could sensitize K562 cells to cell death mediated by NK cells. If NK cells are not purified, the possibility of a combined effect of IFN- γ and NK-mediated toxicity should be discussed.

- In Fig2 and Suppl Fig6 as for all quantification by flow cytometry of small versus large populations (in the case of Fig2 p24- and p24+ CD4+ T-cells) authors should use FlowJo strength to compare equal numbers of cells to avoid any bias due to the funder effect.

- P18: authors say, "Finally, at a functional level, NKG2C+ CD57- memory NK cell precursors were intrinsically characterized by higher expression levels of CD107a degranulation marker (Supplemental Figure 4D, left) and higher co-expression with IFN (left) and granzyme B(right), compared to effector NKG2C- CD57+ NK cells (Figure 2K)". Without mentioning in which population this is observed. The caption of the figure is written PWH but does not mention whether the group is responder, dysfunctional or mixed. This needs to be clarified.

- P20: the authors inhibit TIM3 on NK cells of dysfunctional PWH and observe no change in p24 reduction and conclude that TIM3 is not involved. However, given that TIM3 is higher in effect. Group and positively correlated with p24 reduction, it is in this last group that the authors should have inhibited it and see whether it diminished p24 reduction levels.

- Authors should clarify why anti-CD4 antibody increases p24 levels.

- Suppl Fig9B, given the very low number of events (under 30), authors should reconsider using words as "significant" in the result section, since 3 cells represent around 10% variation.

- In the discussion p27-28 "In fact, IL-15 secreted by DC has been involved in the activation of NK cells (81), but in our model the expression of this cytokine was not detected" however, the authors never measured IL-15 secretion by educated DC and should reevaluate their statement or add the data.

Minor points:

- The manuscript needs to be checked for English, mostly for the phrasing.

- It would have been of great help for the reviewing process to show line numbering, as it helps to point elements.

- ADCC incubation time should be described in the method section as well as the ratio of NK to target cells.

- "Immunomagnetic selection of NK cells, co-culture with Nano-PIC-DC and functional assays CD4+ T cells from PWH." Section in methods should be one section above. Or at least NK-cells isolation method should be described before their first depicted use in cytotoxicity assays.

- "Mo" as used by the authors is not described

- Nano-PIC or S-Pic as described in the method section should be used accordingly

- As all reagents are listed in Fig Supl 2, then state it once in the method section and avoid adding the manufacturer to the method section then. For instance, some antibodies have associated companies, while others do not.

- Concentration and origin of PHA and IL-2 used for CD4+ T-cell reactivation should be mentioned.

- Origin and production method (if applicable) of bNAbs used should be mentioned

- Regarding NSG mouse experiments, page 12 authors say that mice were injected at a "1:2:4 (DC: NK:CD4) ratio". What is the relevance of such a ratio?

- Supplemental Fig3.D: are those data obtained with HD or PWH cells?

- P23 : We then compared the evolution of NKG2C+ CD57- precursor and NKG2C+ CD57+ mature memory NK cells in hBLT

mice infected with HIV-1 compared with an uninfected group of animals (Supplemental Figure 7C). Suppl Fig7 does not refer to anything in this sentence.

Referee #1 (Remarks for Author):

ART effectively suppresses HIV replication but fails to eliminate persistent viral reservoirs in latently infected CD4+ T cells, leading to chronic immune activation and exhaustion of antiviral immune cells like CD8+ T cells and NK cells. Enhancing NK cell activity, particularly memory-like NKG2C+ NK cells, shows promise as they exhibit strong cytotoxicity and are linked to functional HIV remission. DC can enhance NK cell activity through Ag presentation and cytokine signaling.

In this study authors aim demonstrating that nanoparticles loaded with Poly I:C (Nano-PIC) enhance DC-mediated activation and cytotoxicity of NK cells against HIV-1-infected cells, especially by expanding memory NKG2C+ TRAIL+ NK cells. Combining TIGIT blockade with Nano-PIC DC immunotherapy further restores NK cell functionality and reduces HIV-1 reservoirs, providing a potential strategy for improved immunotherapy against HIV-1.

Introduction

- Line 80: Consider citing Levy Y et al., Eur J Immun 2014 for "stimulation with activated dendritic cells (DCs) loaded with HIV-1 peptides (19, 26-29)."

As requested by the reviewer we have added the mentioned reference in the revised manuscript.

- Line 124: Rephrase the sentence for clarity: "Finally, TIGIT blockade restores the cytotoxic activity of dysfunctional NK cells against p24+ cells in vitro following Nano-PIC DC stimulation."

We have now modified the sentence using the modification suggested by the reviewer.

Methods

- Overall, the experiments require more detailed descriptions, including information on doses, time points, p24+ cell isolation methods, and co-culture ratios, or references to previous publications for these protocols.

Following the reviewer's comment we have now provided a more detailed description of experimental procedures and protocols used in the method section from our revised manuscript.

- Line 159: Provide additional details about the "complexes of polymeric nanoparticles (TRANS-ITX2-Mirus Bionova)" or cite a relevant publication for clarity.

We have now included the protocol of polymeric nanoparticle complexes using the commercial TransIT-X2 delivery reagent and we have added relevant references, as requested.

- Trans-well experiments are not described and should be included for completeness.

We apologize for omitting this information. We have now included the missing information for the transwell assay in the method section of the revised manuscript.

- Some abbreviations used in the manuscript need clarification, such as "DCs generated in vitro from circulating adherent Mo" (Line 156), ULBP1, PB, etc.

We have now modified the abbreviation of DC generated from Mo to MDDC and clarified the other remaining abbreviations in our revised manuscript.

- Certain statistical tests, such as Grubbs' test, are mentioned but not adequately explained.

We have improved the description of the statistical tests used including the Grubb's test

- The description of box-and-whisker plots (e.g., mean vs. SD) should be clearly specified.

We have now specified the description of box and whisker with median and maximum and minimum values represented for each presented plot in the figure legends from the revised manuscript.

Results

i. in vitro model of DC activation

Authors demonstrate first that DC stimulated with Nano-PIC enhance the activation and cytotoxic function of NK cells, particularly in people with HIV (PWH) on ART. Nano-PIC DCs induce higher expression of maturation markers (CD86), cytokines (IL-12), and ligands for activating NK receptors compared to soluble Poly I:C (Sol-PIC). This activation leads to increased degranulation (CD107a) and IFN γ production in NK cells, improving their ability to eliminate HIV-infected CD4⁺ T cells.

1. The activation of DCs from HIV⁺ donors did not yield significant results, which is an important expected outcome, as the authors propose using this system for therapeutic approaches. It would be advisable to first validate MoDC differentiation with a more comprehensive profile (e.g., Sup Fig1A). Maturation profiles are often more clearly represented using MFI rather than the percentage of positive cells.

We thank the reviewer for the suggestion. As requested, we have now increased the number of experiments including higher number of PWH donors used to generate MDDCs (see new main Figures 1, 2 and supplemental figures 1A from the revised manuscript). We have also included a more detailed description of MDDC phenotype generated in vitro in our new supplemental figure 1, and as requested by the reviewer we have also represented expression of maturation markers in MFI rather than percentages.

2. The presentation of data in Figure 1 versus Supplemental Figure 1 is confusing. Data from HD and HIV⁺ donors should be grouped for better readability.

We have followed the reviewer's request and we have grouped the extended data on MDDC and PWH and HD in the same main Figure 1,2 and supplemental figures 1-3, including some of the additional information requested by the reviewers. We hope the improved data representation is now more clear.

3. In Figure 1A, it is recommended to show the "Med" condition instead of FMO, while the complete gating strategy with FMO should be moved to Supplemental Materials.

As requested, we have included the Med condition in Figure 1A and we have kept the FMO in the same figure to facilitate visualization of the representative staining.

4. Authors tend to overinterpret certain results, which should be adjusted for clarity:

We thank the reviewer for pointing this out. We have now clarified and modified the statements throughout the manuscript to avoid overclaiming.

- Line 320: "Treatment of DC with Nano-PIC induced a mild but significant reduction of cell viability" - This is not observed for PWH.

We have now clarified that cell viability is more affected by Nano-PIC treatment in MDDC from HD but not in PWH. We have mentioned in the text and discussed that potential differences in cell viability may be due to obtaining Mo from fresh (PWH) versus from buffy coat (HD) samples.

- Line 324: "Nano-PIC more significantly and consistently increased intracellular expression of IL-12 in DC from both HD and PWH" - This is not entirely accurate for PWH; while a trend is visible, it is not significant. The authors should include more donors to achieve statistical significance or perform dose-dependent testing to better demonstrate Nano-PIC DC functionality from PWH, especially in the context of therapeutic applications.

We have followed the reviewer's recommendation and we have increased the number of PWH donors to confirm significant increase of IL-12 in MDDCs activated with nano-PIC compared to sol-PIC. The new dataset was added to our new Figure 1B of the revised manuscript.

- Line 328: "Upregulated at 6 and 16h in DC primed with Nano-PIC from HD and PWH (Supplemental Figure 1B-1C, right, Figure 1C, Supplemental Figure 2A-2B)" - i) There is no significant response in DC from PWH. ii) Clarify in the methods section why Figure 1B shows 16h of stimulation for HD, while Sup Fig1 shows 6h.

We thank the reviewer for the comment. We have clarified that we showed levels of CD86 at 16h in our original Figure 1B (now shown in new Figure 1C) which represents the final maturation state reached by MDDC, but we have also monitored the expression of this molecule at earlier time points such as 6h of stimulation (Supplemental Figure 1F). We have clarified this information in the methods and the results sections from the revised manuscript.

iii) In Sup Fig2, define "sample1," ensure gates are consistent across all dot plots, and label the percentages. Also, include non-stimulated controls.

We have now revised the gating strategy as well as included representative plots from non-stimulated, soluble PIC, empty Nano and Nano-PIC MDDC.

5. MoDCs from HIV- and HIV+ donors appear to differ at baseline (e.g., CD86 expression). The authors should provide a more detailed phenotyping of MoDCs, including additional maturation markers (e.g., CD40, as described by Marta Calvet-Mirabent in EBioMedicine 2022), and discuss the reduced responsiveness of DCs from PWH to Nano-PIC.

As the reviewer pointed out, MDDC from PWH express higher basal levels of CD86 as previously described (Barron et al,2003; Smed-Sørensen et al, 2004 now included in the discussion), for this reason we have normalized data to baseline levels to better determine increased MFI levels in expression of these molecule in response to Nano-PIC stimulation. In addition, following the reviewer's request we included also CD40 expression in the new dataset of MDDCs from PWH and baseline levels of CD86 and CD40 in PWH compared to HD (new supplemental figure 1E-F-G). These observations have been further discussed in the context of the literature as suggested.

6. Line 331: "MICa/b and ULBP1 as well as HLA-E, which is the ligand for both the activating and inhibitory receptors NKG2C or NKG2A, respectively (Supplemental Figure 2A; Figure 1D)." Supplemental Figure 2A refers to PWH, while Figure 1D refers to HD. Please clarify. Additionally, HLA-E in Figure 1D is borderline significant and should be noted accordingly.

We have modified the mentioned sentence for clarification purposes to:

MICa/b and ULBP1 (ligands for NKG2D) as well as HLA-E, (ligand for both the activating and inhibitory receptors NKG2C or NKG2A) (Supplemental Figure 2A; Figure 1D)."

Also, the number of PWH donors has been increased to confirm significant induction of HLA-E expression is induced by Nano-PIC on MDDC from HIV+ individuals.

7. Line 333: "The increase of these ligands in DCs from PWH stimulated with Nano-PIC was confirmed." – No data is presented comparing Sol-PIC versus Nano-PIC in DCs from PWH.

Although we originally analyzed the expression of ligands for activating NK receptors specifically in MDDC treated with Nano-PIC, we have now included a new dataset comparing levels of these ligands in soluble versus Nano-PIC conditions in parallel (see new Figure 1D from the revised manuscript).

8. Statistical analysis for Figures 1 and 2, as well as the Supplemental Figures, relies on the Wilcoxon paired t-test, which is not appropriate for multiple group comparisons. ANOVA or Friedman tests should be used instead.

Multiple comparison correction using either a Bonferroni or Friedman or Kruskal wallis ANOVA tests has been applied when necessary in our revised manuscript (see materials and methods and figure legends).

9. Line 336: “Nano-PIC DCs may display increased abilities to activate NK cells.” – Specify this is based solely on CD86 expression, as the percentage of CD86+ cells in PWH remains unchanged.

We have now clarified the hypothesis based on the previous observations of increased early secretion of IL-12 and also NK ligands and CD86 at 16h are increased after treatment of MDDC with Nano-PIC.

ii. NK function with DC Nano-Pic

Effective responders among PWH showed elevated memory NKG2C+ NK cells with increased TRAIL expression, a key mediator of cytotoxicity, while dysfunctional responders exhibited immune exhaustion linked to TIGIT overexpression. Blocking TIGIT restored NK cytotoxicity in dysfunctional responders, highlighting potential therapeutic avenues using Nano-PIC DCs to bolster immune responses and target HIV reservoirs.

1. Provide an experimental scheme in Supplemental Figure for clarity.

We already had provided a scheme of the experimental procedure used for the functional assay in the main figure 2A from our original manuscript. We have now further expanded the scheme in order to increase clarity which is shown in the new figure 3A.

2. Line 339: “Expression of IFN γ and CD107a on CD56dim CD16+ cells was analyzed (Supplemental Figure 3A).” The representative dot plots are not convincing, as the number of dots appears too low in the double-positive populations. Include comparative dot plots with associated groups in a single plot, and provide the exact number of events for accuracy.

As requested, we have included a new representative plot in Supplemental Figure 3A with higher number of events of double positive cells for IFN γ and CD107a, in NK cells in the absence or the presence of empty Nano and Nano-PIC MDDC stimulation in a PWH.

3. Line 340: “The expression of the degranulation marker CD107a (Figure 1E, left panel) and proportions of CD107a+ IFN γ + (Figure 1E, right panel) cells within the CD56dim CD16+ NK subset were observed in response to Nano-PIC DCs.” Clarify whether this is in comparison to NK cells alone. Additionally, differences between DC Nano-PIC and DC Nano groups are not adequately described in Figure 1E. Ensure it is specified that these data are from HD.

We have now included also degranulation data from PWH and increased the number of donors for both HD and PWH in a new figure 2A, allowing us to detect significant differences not only compared to NK alone but also with MDDC loaded with empty nanoparticles. We have modified the figure and the sentence from our revised manuscript accordingly.

4. Line 342: “The induction of these degranulation parameters was confirmed in

DCs from PWH exposed to Nano-PIC (Supplemental Figure 3B).” No significant results are presented. This statement should be adjusted accordingly.

We have increased the number of experiments with nano-PIC MDDCs culture with NK cells using PWH to have enough sample size to appreciate significant differences and we have updated the new figure 2A (both bottom and upper panels) and text from the revised manuscript to reflect these changes.

5. Line 344: “This effect was dependent on cell-to-cell contact of NK cells with Nano-PIC DCs in a transwell assay compared to DCs treated with empty nanoparticles (Figure 1F).” The transwell experiment is not described, and no data for DCs from PWH are included.

We have described the transwell experiment in methods section and also added new transwell data using MDDC and NK cells from PWH for confirmation purposes in the new figure 2B.

6. Line 349: “NK cells primed with Nano-PIC DCs were able to kill significantly higher proportions of K562-GFP cells, compared to untreated NK or cells.” Specify donor groups and ensure the experiment includes comparisons for both HD and PWH.

We have now added a new dataset of killing assays using the target K562-GFP with both NKs from HD and PWH (New figure 2C).

7. For the ADCC test on gp120/CHO cells, include a control without neutralizing antibodies (NeutAb). The choice of the cocktail used should be justified within this model. Discuss the rationale for using CHO-HXB2 as the model, as opposed to infection models like JRCSF, which are presented in subsequent figures.

Our ADCC data shown in supplemental figure 3D represents values of cell death from CHO cells expressing HIV-1 -gp120 cells killed in the presence of NK cells and the bNAb cocktail after subtraction of baseline killing of the cell line cultured just in the presence of NK, to reflect specific ADCC effect induced by the bNAbs. Attached to this letter we are including the values from the individual conditions. We used the CHO model since these cells are transfected the HIV-1 gp120 from the consensus HXB2 strain in their membrane and allow to distinguish between natural and ADCC NK cell mediated cytotoxicity in the absence or presence of bNAbs directed to conserved env regions that should be similarly susceptible than other viral strains such as the JRCSF. Regarding the choice of bNAbs, to facilitate recognition of conserved epitopes, we chose bNAbs directed to conserved HIV-1 envelope sites such as VCR01; 3BNC117 (CD4 binding site); PGT121 (V3 glycan site) that have been previously shown to be effective neutralizing most HIV-1 strains in combination (*Thavarajah et al, 2024; Edupuganti et al, 2025; Frattari et al, 2023*). This information has been included in the methods section from the revised manuscript.

Figure for reviewers removed.

8. In Figure 2B, the baseline is not shown, despite being mentioned in the legend. Ensure that the baseline is included for clarity.

In Figure 2B, to avoid overgrowth of viral particles we have used CD4+ T cells treated with Raltegravir (Ral) alone as our baseline condition, which is included in the gating strategy. We have also clarified this in the text of our revised manuscript and also included the data for this condition in the normalized plots from the main Figure 3G and Supplemental Figure 4A-B.

9. The model presented in Figure 2A measures p24+ cells. It would be important to demonstrate that the DC Nano-PIC system impacts cells producing infectious particles (reservoir cells) rather than defective virus.

To address this comment, we have measured the amount of intact and defective HIV provirus sequences by IPDA as a redout of replication competent virus in CD4+ T cells from PWH after culture NK cells and nano-PIC MDDCs and compared with the tendency in intracellular p24 expression analyzed by flow cytometry in parallel for the same samples. We have confirmed a decrease in proportion of intact HIV sequences from total HIV DNA detected in these assays. The new dataset has been included in our new supplemental figure 4D.

10. The legend for Supplemental Figure 4 mentions the use of DC Nano empty, which appears to be the most appropriate control. However, this control is not presented in Figure 2 and is missing.

We have included the empty nano-MDDC condition in the Supplemental figure 4 A-B and figure 3G from our revised manuscript.

11. The use of the CD57 marker in Figure 2C should be clarified, including its relevance and interpretation in this context.

We have further justified the use of CD57 as a maturation marker for adaptive NK cells (Kobyzeva *et al*, 2020; Kared *et al*, 2016).

12. For Figure 2, the Wilcoxon test is not suitable, as the three groups are independent. An appropriate statistical test, such as ANOVA or Kruskal-Wallis, should be used instead.

For independent samples in panels H,I, J we had originally used a Mann witney test, we apologize. We have now included this information and improved the analysis adding a multiple comparison correction using Kruskal-Wallis test for these comparisons in figure 2.

13. In Figure 2K and Supplemental Figure 4D, clarify whether the 8 PWH are grouped as effectors. Did the authors observe this functional difference in KKG2C CD57- cells only within the effector group? This distinction should be clearly described.

For the original analysis we used all samples of PWH available. To address the comment made by the reviewer, we have specifically shown the phenotypical changes in the effective responder PWH in our new figure 3K and supplemental figure 4F)

iii. Immunomodulation through TIGIT expression

1. It is regrettable that the baseline for HDs is not included, especially as the study aims to explore immune exhaustion in PWH. FACS histograms in Supplemental Figure 5A should display baseline levels from representative donors in all three groups to provide a clearer comparison.

We have now included baseline levels of PD-1, TIGIT and TIM3 for NK cells from HD and also compared with the different PWH groups (new supplemental Figure 5B-C) as well as comparative FACS histograms of representative donors from the three effective, unclassified and non-responder PWH groups (new supplemental figure 5C).

2. The rationale behind performing correlation analyses on "all groups" (black) versus "all without dysfunctional" (red) is unclear, especially when subgroup correlations are clearly established in Figure 4D. The coding, color scheme, and legend for these analyses need to be clarified to enhance understanding.

We have now clarified in the figure legends and in the text the correlation analyses considering all PWH or excluding unclassified but maintaining effective or non responder groups, and we have changed the color of the significant values for these particular comparisons in our new figure 4-5 and supplemental figure 5,7. We apologize for previous confusions. We also have explained highlighted outliers and criteria of exclusion using the Gubb's test in our methods section and figure legends.

3. Line 429: "In contrast, induction of TIM3 expression in NK was significantly

correlated with lower proportions of infected p24+ CD4+ T cells in our functional assays (Supplemental Figure 5E) and higher frequencies of NKG2C+ CD57- memory precursor NK cells and lower percentages of NKG2C- CD57+ cells within the CD56dim (left) and CD56lo/- (right) CD16+ subpopulations (Supplemental Figure 5F-5G)." The authors overinterpret their results here. This statement should be rephrased to better reflect the data's actual significance and limitations.

We have softened the sentence with regards to the mentioned associations of TIM-3 to avoid overclaiming: "levels of TIM3 expression in NK appeared to be associated with lower proportions of infected p24+ CD4+ T cells in our functional assays (supplemental Figure 5G) and with higher frequencies of NKG2C+ CD57- memory precursor NK cells and lower percentages of NKG2C- CD57+ cells more significantly in the CD56lo/- (right) CD16+ subpopulation..."

4. Line 452: "Thus, our data demonstrate that blockade of TIGIT can improve functional restoration of NK cells from dysfunctional responder PWH in combination with Nano-PIC DCs." To strengthen this conclusion, it would have been more compelling to directly compare the effects of these treatments across responder and non-responder groups. This approach would validate the dichotomy between these groups and reinforce the study's findings.

We have now shown separate effect of TIGIT (Figure 4D; supplemental Figure 6B) and TIM3 (Supplemental Figure 6C) blockade in separate responder and non-responder groups and we have highlighted this information in the revised manuscript.

iv. Mediation by TRAIL

1. Supplemental Figure 6A: The percentage of p24+ cells is low, making it essential to include the number of events acquired in the methods section to validate the percentages shown in Figure 4 and Supplemental Figure 6. This addition would enhance the credibility and reproducibility of the data.

We have now included in the methods section the number of CD4 T cell events acquired allowing us to confirm low percentages of p24 detected in analyses shown in figures 4 and supplemental figure 6.

2. In general, it would be helpful to include a reference line at Y=1 in fold-change graphs to clearly distinguish between groups showing an increase versus a decrease. This visual aid would make the data interpretation more straightforward.

As suggested by the reviewer we have now included a reference line in all the normalized data at Y=1 fold-change in our revised manuscript.

3. Line 470: "None of the NK cell subsets from dysfunctional...Supplemental Figure 6B_C" is inconsistent with the data shown in Supplemental Figure 6C (right). The statement should be revised to accurately reflect the results presented in the figure.

We have now clarified that the plots show trail levels in CD56dim CD16+ NK cells for effective responders in the main figure and highlighted differences for the other groups.

4. Line 478: "Therefore, our results indicate a generalized defect in upregulation of TRAIL in both NKG2C+ and NKG2C- NK from dysfunctional responder PWH that is linked to TIGIT upregulation." The term "linked" does not seem appropriate here, as it implies a causative relationship. A more neutral term, such as "associated with," would be more accurate and better supported by the data.

We have now changed the term linked by "associated with" in the mentioned sentence.

v. In vivo models
In a humanized BLT mouse model of HIV-1 infection, TIGIT expression increases on NKG2C+ memory NK precursors during their maturation, correlating with viral loads. Combined Nano-PIC DC and NK cell immunotherapy with TIGIT blockade effectively reduces HIV-1-infected CD4+ T cells in both peripheral blood and lymphoid tissues of NSG mice, as evidenced by reduced p24+ cell proportions and cluster sizes. This treatment enhances cytotoxic NK cell recruitment and granzyme B expression in high viral replication zones, with a concurrent preservation of TRAIL-expressing NKG2C+ memory NK precursors in lymphoid tissues.

1. Some figures are not correctly mentioned (e.g., Supplemental Figure 8): Ensure all references to figures throughout the text are accurate and consistent with their labeling. Correctly referencing figures will improve clarity and avoid confusion.

We apologize for the mistake. We have now corrected the mention of figures in the correct order in our revised manuscript.

2. Line 553: The phrase "we detected a considerable increase of up to a median of 23.35% of p24+ cells in human circulating CD4+ T cells from the transplanted NSG mice that did not receive any immunotherapy" should be rephrased for precision. Replace "considerable" with a more objective term such as "notable" or "measurable," or simply describe the increase quantitatively without qualitative language.

We have followed the reviewer's recommendation and substituted the "considerable" for the "notable" term in the mentioned sentence.

3. Line 557: The statement "treatment of mice with Nano-PIC-DCs and NK cells reduced proportions of p24+ cells within circulating CD4+ T cells, suggesting that immunotherapy by itself was able to reduce expansion of infected cells in PB" is misleading since the difference is not statistically significant compared to the isotype control. Clarify this limitation in the text.

As indicated by the reviewer, we have now specifically referred to the non-significant difference of the MDDC-Nk immunotherapy in the presence of Isotypic control mAb.

4. Supplemental Figure 8B: Clearly specify how "events of cells" were

calculated. Providing methodological details will ensure the data's transparency and reproducibility.

We have now clarified how absolute numbers (instead of referring to events) were calculated in peripheral blood and spleen from the in vivo experiments shown in Supplemental Figure 8.

5. Supplemental Figure 8D: Define what "example 1 to 3" represent. Explicitly stating their meaning will avoid ambiguity for readers.

We have now explained that the dot plots shown are representative images of three different examples showing either high, intermediate, low or undetectable proportions of p24+ cells within gated hCD45+ CD4+ T cells present in the humanized mice.

6. Figure 6C: Draw and label the clusters on the figure to enhance interpretability and provide a clear visual representation of the data.

Following the reviewer's recommendation we have now shown a representative image showing the delineated quantified p24+ cell cluster (see new figure 7C) .

7. Supplemental Figure 9: The number of events appears insufficient to determine the percentages of these populations reliably. Include the total number of events analyzed in the methods section or the figure legend to validate these findings.

We have now shown representative dot plots displaying higher number of events using combined data from each animal group to facilitate visualization of cell populations and we have specified the number of hCD45+ events analyzed in methods section.

8. Line 595: The phrase "We observed a significant enrichment of NKG2C+ CD57- precursors within the total pool of memory NK cells in the spleen after anti-TIGIT mAb treatment..." would benefit from including pre-treatment (pre-mOVA) data in Figure 7A for comparison. This would provide a more comprehensive view of the treatment's impact.

We apologize for the confusion in the mentioned sentence we referred to intrinsic enrichment present in the group of mice receiving anti-TIGIT versus IgG Isotypic control mAbs. We now have included baseline levels of different NKG2C+ CD57- and NKG2C+ CD57 in the PWH donors prior to inoculation, comparing the evolution in these populations in humanized mice treated with Isotype or aTIGIT mAbs. We have now clarified this information in the methods and main text from the revised manuscript Include in the Pie comparison with pre-mOVA levels of NKG2C+ CD57- and NKG2C+ CD57+ (new figure 8 A).

9. Supplemental Table 3: Clarify whether the Effective versus Dysfunctional status of the three donors is known. If not, conducting experiments with defined donors (Effective vs. Dysfunctional) would strengthen the conclusions drawn and make the data more robust.

We have clarified the responder group of the three effective PWH used to in vivo experiments, by including an additional table 5 with their clinical data and the baseline proportions of NKG2C prior to mVOA experiments. We have observed that donor 3 with the lowest basal NKG2C levels, is the patient that displays lower level of viral control in the absence of TIGIT blockade (new

supplemental figure 9). Also, the data suggests that anti-TIGIT treatment can further also improve the in vivo viral control for the other two d1 and d2 PWH (see new supplemental Figure 9) in line with new in vitro data provided of the effect of anti-TIGIT mAb in effective responder PWH in new Supplemental Figure 6. Therefore, we think this is an evidence for efficacy on different types of PWH donors and we have not performed further in vivo experiments.

Discussion

This study explores immunotherapy strategies for HIV-1 remission, focusing on enhancing NK cell activity through DC immunotherapy and checkpoint blockade. Nano-PIC DCs improved NK cell cytotoxicity, particularly by promoting NKG2C+ memory NK cells, which showed increased functionality against HIV-1 infected CD4+ T cells. The research also identifies TIGIT as a key inhibitory receptor associated with NK cell dysfunction in HIV-1 infection, and its blockade improved NK responses. Additionally, TRAIL was found to play a role in NK cell-mediated cytotoxicity. These findings suggest that modulating NKG2C+ NK cells and targeting TIGIT could improve NK-based immunotherapies for HIV-1.

Suggested rephrased limitations to mention in the Ms:

1. Specificity of the Therapy: The proportion of p24+ cells is relatively low in the in vitro system (~0.1%). The authors could discuss the potential effects of the therapy on uninfected cell populations, as this might provide insights into its broader impact and specificity.

We have further discussed that we cannot rule out an impact of the Nano-PIC MDDC treatment in the non-infected populations in our revised manuscript.

2. Potential infection of Nano-PIC activated DCs: A potential limitation of the therapy is whether dendritic cells activated by Nano-PIC might themselves become infected and contribute to viral spread. This risk should be acknowledged and discussed as a potential caveat of the approach.

We have now provided levels of intracellular p24 on NanoPIC-DC and CD4 T cells from a representative PWH donor (see new supplemental figure 4C) and we have discussed the possibility that MDDC may contribute to viral spread in vitro.

3. Cytokines important for NK Activity: The study mentions that no significant levels of IL-15, a cytokine crucial for NK cell activity, were detected in treated DC. These results should be explicitly presented in the manuscript. Additionally, the authors could discuss the potential use of humanized mouse models with Flt3-L, which allow better reconstitution of both NK cells and DCs, to overcome this limitation and improve NK-DC interactions. We have included secreted levels of IL-15 in supernatants from Nano-PIC MDDC cultures as well as we have discussed the possibility of using FTL3-L treated mice as a potential means to improve NK and DC reconstitution.

These points would add depth to the discussion, acknowledging the study's limitations and offering potential avenues for further research.

Referee #2 (Remarks for Author):

The manuscript by Sánchez-Cerrillo I et.al. explored the ways to enhance the cytotoxicity of NK cells to eliminate HIV-1-infected cells. They reported that combination of Nano-PIC stimulated DCs and anti-TIGIT antibodies might be a promising strategy to increase the efficacy of immunotherapies for HIV. While this is an important study, several concerns should be addressed before it can be accepted for publication.

Concerns:

1. The title "NANOPARTICLE-ENHANCED DENDRITIC CELL" is some kind misleading. It will be better to use "Nano-PIC" or other words?

To avoid confusion and fit the journal character limit we have modified the manuscript title to "Combined Dendritic Cell and Anti-Tigit Immunotherapy Potentiates Adaptive Nk Cells Against HIV-1".

2. The DCs used in the study were generated in vitro from circulating adherent monocytes, but not primary DCs isolated from blood. This should be highlighted in the abstract and results part clearly.

We have highlighted the MDDC were derived from Mo in the abstract and the main text.

3. It's known that NK cells interact with the HLA-E molecule via both the inhibitory receptor NKG2A and the activating receptor NKG2C. Did the authors check the expression of NKG2A on NK cells in the experiment systems?

We have included NKG2A expression data in NKs from PWH cultured with nano-PIC MDDCs. We observed a decrease in the expression of this receptor under these conditions, and this information has been included in the supplemental figure 3B and the text from the revised manuscript.

4. The manuscript needs extensive proofreading to make sure all the statements are correct, and all the labels are clear. For example, it's hard to guess whether Figure 1B was generated from HD or PWH.

We have improved the statements and proofreading of our manuscript, and have explicitly referred to the data generated in HD and in PWH throughout the manuscript.

The other example: in the sentence "In fact, higher ratios of cells expressing TIGIT ... and NKG2C+ CD57+ present in this group (Supplemental Figure E-F)", it's not clear which figure the authors was talking about.

We apologize, we have revised the manuscript to avoid errors in the citation of figures as well.

Referee #3 (Comments on Novelty/Model System for Author):

The question is trendy and interesting (NK immunotherapy/impact on HIV-1 reservoir/replication).

The experiments are carried out with care in relevant and difficult models. The ease of using such a method in patients is questionable

We thank the reviewer for appreciating the findings and the experimental systems used in our preclinical study.

Referee #3 (Remarks for Author):

HIV cure remains a challenge despite decades of intensive research. Sánchez-Cerrillo et al. Present a nice study ranging from in vitro evidence to a preclinical model for the proof-of-concept of a novel immunotherapy aiming to restore NK-cell function and activate their cytotoxic activity towards HIV-1 infected cells. Such a strategy could be a game changer in the field. Experiments are carried out with an exceptional level of effort (autologous primary cell co-culture) and result in high-quality data. However, the interpretation is sometimes confusing, mostly due to some unnecessary speculation.

We appreciate the comments about the relevance of the study and we have addressed all the comments to avoid unnecessary speculation.

Major comments:

- Figure 1 shows that DC from HD and PLWH react differently to stimulation. This intriguing finding warrants further exploration, as it could significantly impact the efficacy of immunotherapy in the target group. As the lack of IFN β response is striking, a Luminex may be performed to determine what cytokine profile of HD and PLWH stimulated DC. Despite this very interesting observation, the author focuses all subsequent analysis on NK cells to explain the responder/non-responder status. The DC capacity to respond to the stimulus may deserve attention. We have now confirmed a significant reduction in the ability of MDDC from PWH to secrete IFN β using an ELISA and we have discussed the different efficacy to MDDCs from HD and PWH to induce secretion of IFN β . Interestingly, we have also observed that type II IFN such as IFN γ is more efficiently and significantly produced by MDDC from PWH after Nano-PIC treatment. These new data have been included in the new supplemental figure 1D from our revised manuscript.

- The authors stated very clearly that the immune environment of PWH is

complex and could lead to dysfunctionality in the immune response. It would be great to add data on the phenotype of NK from dysfunctional or responsive PWH. For example, the authors show that TIGIT expression level by NK is inversely correlated with the response to immunotherapy. Is it already the case in circulating NK from the so-called dysfunctional PWH? As requested by this reviewer and the other, we have compared baseline levels of TIGIT and other checkpoint receptors in NK from PWH and HD, as well as provided representative FACS data of TIGIT, TIM3 and PD-1 for NK cells from effective responder and non-responder PWH showing that there is no obvious differences in the absence of Nano-PIC MDDC stimulation (new supplemental figure 5A-C).

- How PLWH were classified into responders/dysfunctional/unclassified responders in figure 2 is questioning. Some responders have complete suppression with NK only, which then impedes the capacity to consider them as responders or not, the other way around, some unclassified responders show a decrease in the proportion of p24 cells. This creates confusion. Objective thresholds must be used to define the groups.

We have now specified that responder and non-responder phenotype was assigned based on the increase or decrease in p24+ cell proportions following Nano-PIC DC treatment compared to LRA-reactivated CD4+ T cells as one of our objective criteria together with levels of NKG2C defined in our ROC curve analysis. However, we have discussed the complexity of intrinsic differences in killing abilities of individual NK cells from these groups that may be affected by MDDC treatment.

- In this vein, why use "dysfunctional"? It sounds subjective. Why not use "non-responders"?

We have changed "dysfunctional" by "non-responders" throughout the manuscript.

- The validity of a ROC curve performed on selected, but not all individuals should be confirmed by an expert on biostatistics.

We have specified that we used all the PWH data for the ROC curve analysis which predicted 80% of cases, we did not select patients for this analysis. Unclassified PWH were the remaining 20% that could not be predicted by our ROC curve criteria and this has been highlighted in the text of our revised manuscript.

- Also, data support that DC therapy and TIGIT blockade have an impact on NK cell functions, but some controls are missing to determine the extent of their synergy. (i) The authors should demonstrate that the enhancement of cytotoxic activity due to TIGIT blockade is specific to Nano PIC DC-exposed NK cells.

We have now included the control with individual NKs with aTIGIT antibody in the absence of nano-PIC-MDDC in the supplemental data (see new supplemental figure 6B).

(ii) The same question applies to TRAIL blockade.

We also have included in supplemental material an additional control in which we show that NK cells cultured with aTRAIL antibody without nano-PIC MDDC are not affected (see new supplemental figure 7G).

(iii) Showing that TIGIT blockade increases TRAIL would also add to their hypothesis.

We have now included an additional data set of experiments where we show we show that blockade of TIGIT led to an increase in TRAIL expression in mature adaptive NKG2C+ CD57+ NK cells from PWH cultured with Nano-PIC MDDC (see new Figure 5E from the revised manuscript).

Finally (iii) What is the impact of NK alone on HIV-1 infection in the humanized mice model? This control is missing to confirm the activity of the DC immunotherapy in this model. although we agree that an individual NK cell control would have been important, we had to prioritize the number of conditions and animals used for these experiments. We used as a criteria our previous observations (Figure1-Figure 4) showing that NanoPIC-MDDC and NK combination was more effective inducing NK activation and targeting infected cells than individual NK cells. Also, our new *in vitro* data also shows that the blockade of TIGIT and TRAIL in individual NK cells does not exert the same effect as with NK-Nano-PICDC combination suggesting a synergic effect. Therefore, we would not expect a consistent effect of injected individual NK cells in the humanized mouse model.

- The authors tend to overinterpret the significance of the result in many instances:

o P15: "As shown in Figure 1B and Supplemental Figure 1D, Nano-PIC more significantly and consistently increased intracellular expression of IL-12 in DC from both HD and PWH at 6 h compared to cells treated with Sol-PIC" Supplemental Fig1D show no statistical difference between conditions for PWH
o P16 : "The induction of these degranulation parameters was confirmed in DC from PWH exposed to Nano-PIC (Supplemental Figure 3B)". no statistical difference is shown

We have increased the number of PWH donors and confirmed significant increase in IL-12 expression experiments and in NK degranulation experiments following nano-PIC MDDC treatment.

o P19: "Together, these findings suggest that effective response of NK cells from PWH to Nano-PIC DC immunotherapy is associated with the induction of precursor and mature memory NKG2C+ NK cells with different functional profiles, which may have improved capabilities to target HIV-infected CD4+ T cells." Authors should be careful, even though differentiation is most probable, evidence only show that immunotherapy induces NKG2C expression on NK cells. Furthermore, it is known that the different NK subsets present distinct functionality.

We have now softened the conclusion sentence to avoid overclaiming as suggested by the reviewer.

o P21 : "In contrast, none of the NK cell subsets from dysfunctional and unclassified responders were able to further increase expression of this molecule (Figure 4B; Supplemental Figure 6 B-C)" sentence is true but statistical significance is not met for any of the groups in NKG2C+ CD57- NK population

We have better described the mentioned section to avoid errors.

o P21: "Of note, resulting raw levels of TRAIL were significantly higher in mature NKG2C+ CD57+ from effective responders compared to the low amount of these cells present in dysfunctional responders in the most mature CD56lo/- CD16+ NKbut not the CD56dim CD16+ NK cells subset (Figure 4C; Supplemental Figure 6D)" none of the figures show statistically higher levels of TRAIL in Effective PWH

Statistically higher levels of TRAIL induced in NK from effective PWH responders are shown in our new Figure 5C.

o FigS9 : authors should be very careful analyzing under 100 events to state finding.

We have specified the minimum number of human CD45+ cell events considered for some of the humanized mouse experiments and recognized low events for minor populations in this model.

o P23 In addition, reduced ratios of TIM3+/TIGIT+ cells in these were observed from 1 to 2 weeks p.i. suggesting a loss in TIM3 expression and progressive enrichment of TIGIT in memory NK cell precursors from HIV-1- infected mice (Figure 5D). and should be replaced by and/or because authors shown none of it.

We have now included in the supplemental figure 8G the evolution of percentages of TIM3+ and TIGIT+ cells in NK from HIV-1-infected BLT mice.

o P25 : "Consistently, numbers of hCD4+ T cells were also significantly higher in the spleen in animals treated with Nano-PIC DCs and NK cells in combination with anti-TIGIT mAbs (Supplemental Figure 7B, right plot)" authors show only a significant increase for hCD45 count under TIGIT inhibition but no significant effect on CD4+ T cell count.

We have corrected the reference to supplemental figure 9B for a tendency of increase in CD4+ T cell absolute numbers in the spleen.

o P25 : "However, a higher proportion of animals with undetectable p24 expression within CD4+ T cells from PB was present in the anti-TIGIT group and differences in this group were more significant compared to the control mice transplanted only with CD4+ T cells (Figure 6B, left panel)." Authors do not show animals with undetectable infected circulating CD4+ T-cells but the frequencies of this population for each animals. The conclusion is an overstatement plus there is no difference between isotype and anti-TIGIT group. Reader has to wait until the discussion to see the authors discuss that which is clearly misleading.

We have replaced the sentence to highlight the percentage of mice with

undetectable levels of p24 in circulation and have included a representative FACS dot plot in the new Supplemental Figure 9D.

- Authors should discuss their choice of Mo-DC differentiation protocol, as many others exist and result in different functionalities of DC. This is even more important, since previous work carried out by the authors was done with cDC1, cDC2 and monocytes. As suggested by the reviewer, we have now further discussed the choice of MDDC for immunotherapy instead of primary cDC1 or cDC2.

- In the cytotoxic assay with A562-GFP cells and ADCC assay, the authors did not mention whether NK cells are purified or if coculture of NK-DC is used. This is very crucial, since educated DC secrete IFN- γ , which could sensitize K562 cells to cell death mediated by NK cells. If NK cells are not purified, the possibility of a combined effect of IFN- γ and NK-mediated toxicity should be discussed. We have clarified that NK cells were not reisolated for the functional killing assays, we have specified the type of HD or PWH donor used for the functional assays. We have also generated a new dataset of killing assays with K562 cells using NK cells from PWH. We have also included the control of NK cells in the presence of NanoPIC MDDC or individually.

- In Fig2 and Suppl Fig6 as for all quantification by flow cytometry of small versus large populations (in the case of Fig2 p24⁻ and p24⁺ CD4⁺ T-cells) authors should use FlowJo strength to compare equal numbers of cells to avoid any bias due to the funder effect. We have also considered similar reduced number of events in p24⁻ cell population to the minimum to 1% of events allowed by FlowJo and obtained similar results comparing to the p24⁺ population (see dot plots in figure 2 for the reviewers, attached to this letter).

Figure for reviewers removed.

- P18: authors say, "Finally, at a functional level, NKG2C+ CD57- memory NK cell precursors were intrinsically characterized by higher expression levels of CD107a degranulation marker (Supplemental Figure 4D, left) and higher co-expression with IFN γ (left) and granzyme B(right), compared to effector NKG2C- CD57+ NK cells (Figure 2K)". Without mentioning in which population this is observed. The caption of the figure is written PWH but does not mention whether the group is responder, dysfunctional or mixed. This needs to be clarified.

We have now shown the data from effective PWH responder (since few non-responder PWH were included in the original analysis) to avoid confusion for the mentioned dataset. Nevertheless, similar results although less significant with the few non-responder PWH are attached to this letter (See attached figure 3 for reviewers).

Figure for reviewers removed.

- P20: the authors inhibit TIM3 on NK cells of dysfunctional PWH and observe no change in p24 reduction and conclude that TIM3 is not involved. However, given that TIM3 is higher in effect. Group and positively correlated with p24

reduction, it is in this last group that the authors should have inhibited it and see whether it diminished p24 reduction levels.

We have now shown the impact of anti-TIM3 mAb in effective and non-responder PWH and we did not observe significant results or obvious trends after treatment with this blocking Ab (see supplemental figure 6C-6D). Similarly, blockade of TIM3 does not seem to have an effect on TRAIL expression, in contrast to TIGIT (Supplemental Figure 7I). We also have discussed these findings in our revised manuscript.

- Authors should clarify why anti-CD4 antibody increases p24 levels. We did not observe any upregulation of p24 after combination with anti-CD4 antibody.

- Suppl Fig9B, given the very low number of events (under 30), authors should reconsider using words as "significant" in the result section, since 3 cells represent around 10% variation.

We have discussed the number of events in gated populations from a limited amount of human cells in the humanized in vivo models. Also, to facilitate detection of subsets in the representative FACS data provided in Supplemental figures 9 and 10 we have also combined data from animals from the each separate group to maximize number of events and data visualization.

- In the discussion p27-28 "In fact, IL-15 secreted by DC has been involved in the activation of NK cells (81), but in our model the expression of this cytokine was not detected" however, the authors never measured IL-15 secretion by educated DC and should reevaluate their statement or add the data. We have now included a new data set analyzing secretion of IL-15 and we have further discussed the importance of this cytokine (see new supplemental figure 1D).

Minor points:

- The manuscript needs to be checked for English, mostly for the phrasing. We have now significantly improved the English grammar and phrasing in our revised manuscript.

- It would have been of great help for the reviewing process to show line numbering, as it helps to point elements. We have included number lines in our revised manuscript.

- ADCC incubation time should be described in the method section as well as the ratio of NK to target cells. We have now improved our description of ADCC assay in our methods section.

- "Immunomagnetic selection of NK cells, co-culture with Nano-PIC-DC and functional assays CD4+ T cells from PWH." Section in methods should be one section above. Or at least NK-cells isolation method should be described before

their first depicted use in cytotoxicity assays. As suggested by the reviewer, we have modified the order to allow to mention the NK cell isolation first in the Method section.

- "Mo" as used by the authors is not described. We have now defined de monocyte abbreviation in our revised manuscript.

- Nano-PIC or S-Pic as described in the method section should be used accordingly

Nomenclature of Nano-PIC and Sol PIC has been kept consistently in the figure, methods and text sections from our revised manuscript.

- As all reagents are listed in Fig Supl 2, then state it once in the method section and avoid adding the manufacturer to the method section then. For instance, some antibodies have associated companies, while others do not. We have detailed commercial and manufacturer's information consistently in the reagent's description throughout the revised manuscript.

- Concentration and origin of PHA and IL-2 used for CD4+ T-cell reactivation should be mentioned.

We now have detailed concentration of PHA and IL-2 in the Methods section.

- Origin and production method (if applicable) of bNAbs used should be mentioned

We have now described in the methods and discussion section details regarding the rationale of bNAbs used as well as concentrations, etc.

- Regarding NSG mouse experiments, page 12 authors say that mice were injected at a "1:2:4 (DC: NK:CD4) ratio". What is the relevance of such a ratio? We have highlighted that we were trying to mimic the cell ratios used in our in vitro experiments.

- Supplemental Fig3.D: are those data obtained with HD or PWH cells? We have now detailed the specific data obtained from PWH and HD in all materials from the manuscript

- P23 : We then compared the evolution of NKG2C+ CD57- precursor and NKG2C+ CD57+ mature memory NK cells in hBLT mice infected with HIV-1 compared with an uninfected group of animals (Supplemental Figure 7C). Suppl Fig7 does not refer to anything in this sentence. The reference has been corrected in the main text from the revised manuscript.

6th May 2025

Dear Dr. Martin-Gayo,

Thank you for the submission of your revised manuscript to EMBO Molecular Medicine. I am pleased to inform you that we will be able to accept your manuscript pending the following final amendments:

- 1) Please address the referee #3 remaining concerns and implement referee #1 and referee #3 suggestions. Please pay particular attention to the grammar and syntax and consider running the article by a native English speaker.
- 2) Authors: There are name discrepancies for several authors in our system and in the manuscript. Judith Grau-Exposito in the manuscript vs. Judit Grau-Expósito in our system; Arantzazu Alfranca in the manuscript vs. Arancha Alfranca in our system; Vladimir Vbrnac in the manuscript vs. Vladimir Vrbanac in our system. Please correct.
- 3) Figures:
 - We note that you have 10 supplementary figures. Several of them (e.g. 5) should be made to EV figures and the rest of the supplementary figures with their legends should be compiled together with supplementary tables in the Appendix. Appendix file should be uploaded as a PDF with the table of content with page numbers on the title page. Appendix figures and tables should be renamed to "Appendix Figure S1" etc. and "Appendix Table S1" etc. and their callouts in the main manuscript text should be updated. Please check "Author Guidelines" for more information:
<https://www.embopress.org/page/journal/17574684/authorguide#expandedview>
 - Main figures and EV figures should be uploaded as individual high-resolution files. EV figures legends should be placed after main figure legends in the main manuscript file under the heading "Expanded View Figure Legends". Please updated EV figure callouts in the main manuscript file. Please check "Author Guidelines" for more information:
<https://www.embopress.org/page/journal/17574684/authorguide#figureformat>
<https://www.embopress.org/page/journal/17574684/authorguide#expandedview>
- 4) Abstract: I have gone through your text and revised it (see below). Please review it, amend as you see fit:

Natural Killer (NK) cells are promising candidates for targeting persistently infected CD4⁺ T cells in people with HIV-1 (PWH). However, chronic HIV-1 infection impairs NK cell functionality, necessitating additional strategies to enhance their cytotoxic activity. In this study, we demonstrate that dendritic cells primed with nanoparticles containing Poly I:C (Nano-PIC-MDDCs) enhance the natural cytotoxic function of NK cells from effective responder PWH. These NK cells exhibit increased proportions of NKG2C⁺ subsets capable of eliminating HIV-1-infected CD4⁺ T cells via the TRAIL receptor pathway. In contrast, in non-responder PWH, elevated expression of the inhibitory receptor TIGIT is associated with reduced frequencies of NKG2C⁺ NK cells and diminished TRAIL expression. TIGIT blockade restores functional capacities of NK cells from non-responder PWH by upregulating TRAIL and enhancing cytotoxicity against HIV-1-infected targets in vitro. Furthermore, combining Nano-PIC-MDDC-primed NK cells with anti-TIGIT immunotherapy in humanized NSG mice reduces the expansion of HIV-1-infected cells, preserves NKG2C⁺ NK cell precursors, and increases TRAIL expression. Collectively, these findings support the combined use of Nano-PIC-MDDCs and TIGIT blockade as a promising immunotherapeutic strategy toward an HIV-1 cure.

- 5) In the main manuscript file, please do the following:
 - Please address all comments suggested by our data editors listed below:
 - o Figure legends:
 1. Please note that the exact p values are not provided in the legends of figures 1B-D; 2A-C; 3G-K; 4A, D; 5A, B, C, E, F, G; 6B, D; 7B, C, D; 8A, C.
 2. Please note that the box plots need to be defined in terms of bounds of box and whiskers, and percentile in the legends of figures 1B-D; 2A-C; 3H-K; 4A, 5A, B, C, F, G; 6A, B, D; 7B, D; 8A, C.
 3. Please note that information related to n is missing in the legend of figure 5E.
 4. Please note that the error bars are not defined in the legends of figures 3G, 4D, E; 5E, G.
 - Author contributions: Please remove it from the manuscript and specify author contributions in our submission system. CRediT has replaced the traditional author contributions section because it offers a systematic machine-readable author contributions format that allows for more effective research assessment. You are encouraged to use the free text boxes beneath each contributing author's name to add specific details on the author's contribution. More information is available in our guide to authors:
<https://www.embopress.org/page/journal/17574684/authorguide#authorshipguidelines>
 - Indicate in legends exact n and exact p values, not a range, along with the statistical test used. To keep the figures "clear" some authors found providing an Appendix table Sx with all exact p-values preferable. You are welcome to do this if you want to.
 - Rename "Material and Methods" to "Methods".
 - In Methods, provide the antibody dilutions that were used for each antibody.
 - Please remove Supplemental Table 6 and include structured Methods section that includes a Reagents and Tools Table (should be uploaded as a separate file) followed by a Methods and Protocols section. More information on how to adhere to this format as well as downloadable templates (.docx) for the Reagents and Tools Table can be found in our author guidelines:
<https://www.embopress.org/page/journal/17574684/authorguide#structuredmethods>

An example of a paper with Structured Methods can be found here:

<https://www.embopress.org/doi/full/10.1038/s44320-024-00037-6#sec-4>

6) Funding: Please make sure that information about all sources of funding are complete in both our submission system and in the manuscript. Currently, Formación de Personal Investigador (FPI) grant PRE2022-104516 and UAM fellowship are missing in our system and CB21/13/00107 is missing in the manuscript. Please correct.

7) The Paper Explained: The Paper Explained: Please provide "The Paper Explained" and add it to the main manuscript text. Please check "Author Guidelines" for more information.

<https://www.embopress.org/page/journal/17574684/authorguide#researcharticleguide>

8) Synopsis:

- Synopsis image: Please resize the image to 550 px-wide x (300-600)-px high and upload it as a high-resolution jpeg file.

- Synopsis text: Please provide a short standfirst (maximum of 300 characters, including space) as well as 2-5 one sentence bullet points that summarise the paper as a .doc file. Please write the bullet points to summarise the key NEW findings. They should be designed to be complementary to the abstract - i.e. not repeat the same text. We encourage inclusion of key acronyms and quantitative information (maximum of 30 words / bullet point). Please use the passive voice.

9) As part of the EMBO Publications transparent editorial process initiative (see our Editorial at

<http://embomolmed.embopress.org/content/2/9/329>), EMBO Molecular Medicine will publish online a Review Process File (RPF) to accompany accepted manuscripts. This file will be published in conjunction with your paper and will include the anonymous referee reports, your point-by-point response and all pertinent correspondence relating to the manuscript. Let us know whether you agree with the publication of the RPF and as here, if you want to remove or not any figures from it prior to publication. Please note that the Authors checklist will be published at the end of the RPF.

10) Please provide a point-by-point letter INCLUDING my comments as well as the reviewer's reports and your detailed responses (as Word file).

I look forward to reading a new revised version of your manuscript as soon as possible.

Yours sincerely,

Zeljko Durdevic

Zeljko Durdevic
Senior Editor
EMBO Molecular Medicine

*** Instructions to submit your revised manuscript ***

To submit your manuscript, please follow this link:

<https://embomolmed.msubmit.net/cgi-bin/main.plex>

1) a .docx formatted version of the manuscript text (including Figure legends and tables)

2) Separate figure files*

3) supplemental information as Expanded View and/or Appendix. Please carefully check the authors guidelines for formatting Expanded view and Appendix figures and tables at

<https://www.embopress.org/page/journal/17574684/authorguide#expandedview>

4) a letter INCLUDING the reviewer's reports and your detailed responses to their comments (as Word file).

5) The paper explained: EMBO Molecular Medicine articles are accompanied by a summary of the articles to emphasize the major findings in the paper and their medical implications for the non-specialist reader. Please provide a draft summary of your article highlighting

This may be edited to ensure that readers understand the significance and context of the research.

Please refer to any of our published articles for an example.

6) Author contributions: the contribution of every author must be detailed in a separate section.

7) EMBO Molecular Medicine now requires a complete author checklist

(<https://www.embopress.org/page/journal/17574684/authorguide>) to be submitted with all revised manuscripts. Please use the checklist as guideline for the sort of information we need WITHIN the manuscript. The checklist should only be filled with page numbers where the information can be found. This is particularly important for animal reporting, antibody dilutions (missing) and exact values and n that should be indicated instead of a range.

8) Every published paper now includes a 'Synopsis' to further enhance discoverability. Synopses are displayed on the journal webpage and are freely accessible to all readers. They include a short stand first (maximum of 300 characters, including space) as well as 2-5 one sentence bullet points that summarise the paper. Please write the bullet points to summarise the key NEW findings. They should be designed to be complementary to the abstract - i.e. not repeat the same text. We encourage inclusion of key acronyms and quantitative information (maximum of 30 words / bullet point). Please use the passive voice. Please attach these in a separate file or send them by email, we will incorporate them accordingly.

You are also welcome to suggest a striking image or visual abstract to illustrate your article. If you do please provide a jpeg file 550 px-wide x 300-600px high.

9) A Conflict of Interest statement should be provided in the main text

10) Please note that we now mandate that all corresponding authors list an ORCID digital identifier. This takes <90 seconds to complete. We encourage all authors to supply an ORCID identifier, which will be linked to their name for unambiguous name identification.

Currently, our records indicate that the ORCID for your account is 0000-0002-2619-6086.

Link Not Available

11) Include a Reagents and Tools Table as part of the Methods section, which can be downloaded from our author guidelines (<https://www.embopress.org/page/journal/17574684/authorguide#structuredmethods>)

Photos 400-800 DPI

*Additional important information regarding figures and illustrations can be found at

<https://bit.ly/EMBOPressFigurePreparationGuideline>. See also figure legend preparation guidelines:

<https://www.embopress.org/page/journal/17574684/authorguide#figureformat>

***** Reviewer's comments *****

Referee #1 (Remarks for Author):

I thank the authors for addressing the suggestions by updating the figures and text accordingly. They have also moderated some of their conclusions based on the statistical analysis. Additionally, previously missing details have now been included.

Minor remarks:

1. The abbreviation "Mo" is still not explained in the Discussion section.
2. Lines 499 and 515: The value $p = 0.06$ could be indicated on the figures.
3. Line 518: The proportions could be stated directly in the text.
4. Lines 554-555: The sentence "two of the three... used in" is still unclear to me and could be rephrased for clarity.

Referee #2 (Remarks for Author):

The author have addressed all my concerns.

Referee #3 (Remarks for Author):

The authors only partially addressed my concerns.

I am still confused about how responders and non-responders are classified. The answer of the authors to my initial comment is: « We have now specified that responder and non-responder phenotype was assigned based on the increase or decrease in p24+ cell proportions following Nano-PIC DC treatment compared to LRA-reactivated CD4+ T cells as one of our objective criteria together with levels of NKG2C defined in our ROC curve analysis. »

In the text; they say (lines 261-263) : « Based on these associations, we then asked if levels of NKG2C could predict different PWH groups whose NK differentially responded to Nano-PIC MDDC stimulation. »

It means that prior to the ROC analysis, two groups of PWH were created, with the criteria « whose NK differentially responded to Nano-PIC MDDC stimulation. » This must be objectively defined and depicted.

Their answer to my first comment is also inaccurate. I asked whether the difference in moDC responses across PWH may explain the responder/non-responder status. The authors answered by comparing DC from HD and PLWH.

Finally, spelling must be further improved. For instance, the first result sentence of the new abstract is long and difficult to understand :

« Here, we show that dendritic cells primed with nanoparticles containing Poly I:C (Nano-PIC-MDDC) improve natural cytotoxic function in NK cells from effective responder PWH which display increased proportions of NKG2C+ cells that eliminate HIV-1 infected CD4+ T cells through the TRAIL receptor. »

I apologize for bringing this up again, as I am not a native English speaker myself. However, it does make the reading uneasy.

Reviewer comments:

Referee #1 (Remarks for Author):

I thank the authors for addressing the suggestions by updating the figures and text accordingly. They have also moderated some of their conclusions based on the statistical analysis. Additionally, previously missing details have now been included. We would like to thank the reviewer for acknowledging the improvement of the revised manuscript.

Minor remarks:

1. The abbreviation "Mo" is still not explained in the Discussion section. The "Mo" abbreviation from monocyte has been explained in the Discussion section, as requested.
2. Lines 499 and 515: The value $p = 0.06$ could be indicated on the figures. p values have been specified in the corresponding Fig EV5 and Fig7 as requested and removed from the text.
3. Line 518: The proportions could be stated directly in the text. The mentioned proportions have been indicated in the main text from manuscript, as suggested.
4. Lines 554-555: The sentence "two of the three... used in" is still unclear to me and could be rephrased for clarity. The sentence has been rephrased and simplified for clarity.

Referee #2 (Remarks for Author):

The author have addressed all my concerns.

We would like to thanks the referee 2 for all the useful comments that have helped us to improve the manuscript.

Referee #3 (Remarks for Author):

The authors only partially addressed my concerns.

We respectfully believe we addressed the most relevant questions initially raised by the reviewer, as specified in our previous point by point response letter. Nevertheless, we apologize if the answer to the remaining questions was insufficient. We now have addressed again these two pending concerns.

I am still confused about how responders and non-responders are classified. The answer of the authors to my initial comment is: « We have now specified that responder and non-responder phenotype was assigned based on the increase or decrease in p24+ cell proportions following Nano-PIC DC treatment compared to LRA-reactivated CD4+ T cells as one of our objective criteria together with levels of NKG2C defined in our ROC curve analysis. »

In the text; they say (lines 261-263) : « Based on these associations, we then asked if levels of NKG2C could predict different PWH groups whose NK differentially responded to Nano-PIC MDDC stimulation. »

It means that prior to the ROC analysis, two groups of PWH were created, with the criteria « whose NK differentially responded to Nano-PIC MDDC stimulation. » This must be objectively defined and depicted.

We apologize by the confusion. We hypothesized that levels of NKG2C on NK cells may define different groups with different abilities to reduce or increase p24 proportions, since we observed a significant correlation between these two parameters using data from all PWH used in functional assays (as shown in correlation network in Fig. 3D). Therefore, there was no previous stratification prior to this observation. The ROC curve was used to estimate levels of NKG2C allowing to discriminate individuals capable of reducing (Fold Change <1) or increasing (fold change >1) proportions of p24+ cells in the functional assays. This was the only functional criteria used for the ROC analysis to define best cut off for NKG2C to define two-responder effective and non-responder groups. This information has now been included in the methods and highlighted in the main text.

Their answer to my first comment is also inaccurate. I asked whether the difference

in moDC responses across PWH may explain the responder/non-responder status. The authors answered by comparing DC from HD and PLWH.

In addition to the previous answer comparing responses in MDDC from PWH and HD provided in the previous review, we have now attached to this letter a figure 1 showing the levels of different functional markers including the maturation marker CD86, the NKG2D ligands HLA-E and ULBP1 and NKG2C/A ligand HLA-E and supernatant levels of IL-12, IFN β and IFN γ in MDDCs from ER and NR PWH in which we conducted phenotypical characterization after activation with Nano-PIC. We did not observe obvious or significant differences in these parameters in cells from the two groups, suggesting that the NK cell response is not due to altered MDDC stimulation. Nevertheless, we have previously acknowledged in the discussion from the revised manuscript that further studies are required to elucidate the role of individual cytokines or receptor-ligand interactions in the abilities of MDDCs from PWH to stimulate NK cells, which includes both groups.

Figure 1

Figure 1. Analysis of phenotypical maturation markers in MDDCs from effective responders and non-responders PWH stimulated with Nano-PIC

treatment. (A, B): Fold change in MFI of CD86 and CD40 (A) and in proportions of ligands for NKG2C (MICA/b and ULBP-1) and the NKG2C/A ligand (HLA-E) (B) in MDDC from effective responders and non-responders PWH after nano-PIC activation normalized to baseline levels present in cells treated with empty nanoparticles. (C): Fold change in secreted IFN γ , IFN β and IL-12 tested in culture supernatants of Nano-PIC-MDDCs from effective responder and non-responder at 16h.

Finally, spelling must be further improved. For instance, the first result sentence of the new abstract is long and difficult to understand: « Here, we show that dendritic cells primed with nanoparticles containing Poly I:C (Nano-PIC-MDDC) improve natural cytotoxic function in NK cells from effective responder PWH which display increased proportions of NKG2C+ cells that eliminate HIV-1 infected CD4+ T cells through the TRAIL receptor. » I apologize for bringing this up again, as I am not a native English speaker myself. However, it does make the reading uneasy.

We appreciate this suggestion and abstract has been adapted to avoid confusion.

16th May 2025

Dear Dr. Martin-Gayo,

We are pleased to inform you that your manuscript is accepted for publication and is now being sent to our publisher to be included in the next available issue of EMBO Molecular Medicine.

Zeljko Durdevic
Senior Editor
EMBO Molecular Medicine
